# A Case Study of Low Ranked Self-Expressive Structures in Neural Network Representations

Uday Singh Saini[1],[*] William Shiao[1], Yahya Sattar[1], Yogesh Dahiya[2], Samet Oymak[3], Evangelos E. Papalexakis[1]

[1]University of California, Riverside, [2]The Institute of Mathematical Sciences, Chennai, [3] University of Michigan, Ann-Arbor

usain001@ucr.edu, wshia002@ucr.edu, ysatt001@ucr.edu, yogeshd2612@gmail.com, oymak@umich.edu, epapalex@cs.ucr.edu

Understanding neural networks by studying their underlying geometry can help us understand their embedded inductive priors and representation capacity. Prior representation analysis tools like (Linear) Centered Kernel Alignment (CKA) offer a lens to probe those structures via a kernel similarity framework. In this work we approach the problem of understanding the underlying geometry via the lens of subspace clustering, where each input is represented as a linear combination of other inputs. Such structures are called self-expressive structures. In this work we analyze their evolution and gauge their usefulness with the help of linear probes. We also demonstrate a close relationship between subspace clustering and linear CKA and demonstrate its utility to act as a more sensitive similarity measure of representations when compared with linear CKA. We do so by comparing the sensitivities of both measures to changes in representation across their singular value spectrum, by analyzing the evolution of self-expressive structures in networks trained to generalize and memorize and via a comparison of networks trained with different optimization objectives. This analysis helps us ground the utility of subspace clustering based approaches to analyze neural representations and motivate future work on exploring the utility of enforcing similarity between self-expressive structures as a means of training neural networks.

## 1. Introduction

Analysing structures in representations of trained Neural Networks has been the subject of interest for many post-hoc interpretability methods [1]. [2] propose a Centered Kernel Alignment (CKA) [3] based similarity measure between linear kernels of network activations (Linear-CKA) that been used to compare deep and wide neural networks in [4], analysing Vision Transformers [5] vs ResNets [6] in [7], comparing effects of loss functions [8], differences between self-supervised and supervised methods [9] and differences between self-supervised objectives for Vision Transformer representations [10].

Recently, works like [11] and [12] have demonstrated that Linear-CKA [2] similarity is usually dominated by similarity between singular vectors of neural activations possessing the largest singular values, thereby rendering it insensitive to differences in singular vectors with smaller singular values. [11] propose a sensitivity test to rigorously evaluate similarity measures by observing the effects of changes in internal representations of a network on a linear classifier's performance on those representations. Taking into account the observations made in [11] about the spectral behaviour of Linear-CKA we motivate a Low Ranked Subspace Clustering ($LRSC$) [13] based pairwise affinity measure in conjunction with CKA and show its relationship to Linear-CKA [2]. We demonstrate how this choice ameliorates some issues raised by [11] regarding Linear-CKA while also offering a more extensive comparison between the two in Section 5.

Since an LRSC Kernel over neural activations highlights *self-expressive structures* [14] in neural rep-

---

[*]Corresponding Author.

resentations, the combination of LRSC with CKA compares the similarity between self-expressive structures of 2 neural representations. In Section 6 we demonstrate that *self-expressive structures* become more class-concentrated as measured by its subspace representation reconstruction (sub. recon.) [15, 16] as we go deeper in the network's layers. Furthermore, this reconstruction based accuracy strongly correlates with a linear probe's [17] performance on the same internal representations, thereby serving as a tool to understand intermediate representations of neural networks via computing just its singular vectors. Additionally in Section 6.2 we analyse networks which generalise well and compare them to networks which memorise parts of their training set and observe that for most of the layers of these 2 networks the learnt representations are similar and the dissimilarities between them only appear in the last few layers where each network learns markedly different representations. These observations are in alignment with results from [18–20].

In Appendix D we explore the limits of representations analysis using tools that approximate linear subspaces. In this setup we use rational activations [21] based ResNets and compare them with ReLU based ResNets under settings of generalisation and memorisation. We test the efficacy of LRSC-CKA and Linear-CKA to discern differences between rational networks with varying generalisation performance and demonstrate deficiencies in their ability to discover meaningful differences between networks trained in different regimes. We then take another prominent approach for representation analysis called Mean-Field Theoretic Manifold Analysis (MFTMA) [19] and demonstrate similar deficiencies its ability for the same task.

Finally in Appendix E to understand the emergence of self-expressive structures in networks trained on Cross-Entropy loss we compare these networks with networks trained on Maximum Coding Rate Reduction(MCRR)[22] Loss. MCRR Loss encourages the model to separate out data points from different classes into different subspaces, thereby encouraging the development of *self-expressive structures*. In doing such a comparison we find that final layers of cross entropy trained networks indeed share similarity with networks trained on MCRR loss, thereby indicating formation of *self-expressive structures*.

## 2. Related Work

Understanding neural networks via comparing the similarity of their internal representations has been the subject of various lines of research with many different similarity measures. To begin with, [23] propose a Canonical Correlation Analysis (CCA) [24] based tool called SVCCA, which uses an SVD over the representations of the network to remove noise before proceeding to compare them using Canonical Correlational Analysis. Building upon SVCCA, [18] propose a different weighting of canonical correlations, thereby calling their methodology Projection Weighted Canonical Correlation Analysis (PWCCA). Subsequently [2] which utilises Centered Kernel Alignment (CKA) [3] to measure similarities between kernels derived out of layerwise activations demonstrates some limitations of CCA in terms of its inability to discover architecturally identical layers in networks trained with different initialisations. Similar limitations of CCA based methods are also demonstrated in [11]. [2] predominantly utilises linear kernels for measuring similarity between networks and therefore we shall refer it to as Linear-CKA.

Other representation similarity based approaches like [25] perform representation similarity analysis by computing correlations between representation similarity matrices based on various distance measures. [26] compares the similarity of representations by considering the distances of positive semi-definite kernels on the Riemannian manifold. AGTIC [27] proposes an adaptive similarity criteria that ignores extreme values of similarity in the representations. [28] utilises Normalised Bures Similarity [29] to study similarity of neural networks with respect to layerwise gradients.

Beyond just utilising representations, works like Representation Topology Divergence [30] learn a graph based on embeddings and then computing similarity based on various connected components in the graph. Works like [31] try to use cosine information to compute an adjacency matrix and study the modularity [32] of the resulting graph. Similar approach was also taken in [33] which computes a graph based on sparse subspace representation [14] and analyses the modularity of such graphs, along with using CKA [2] to compute the similarity between graphs.

# 3. Background

In this section we establish the foundation for the tools used and procedures adopted in our Subspace based analysis of Neural Network Representations. For a reading of related work, please refer to Section 2. We begin by laying the background on Low Ranked Subspace Clustering (LRSC) [13] and provide justification for its use in Section 3.1. Then in Section 3.2 we describe Centered Kernel Alignment or CKA [3, 34], a well known technique for representation similarity comparison. We then combine LRSC with CKA and the resultant approach is described in Section 4.1.

## 3.1. Low Rank Subspace Clustering

Given a Matrix $\mathbf{X} \in \mathbf{R}^{d \times N}$ of $N$ data points which in the context of this study will be activations of hidden layers of a Neural Network. Low Rank Subspace Clustering or LRSC [13] tries to uncover the underlying structure of the data in this union of subspaces. LRSC accomplishes this by trying to find a low rank representation of each point subject to *Self-Expressiveness* [35] constraint, where each point is expressed as a linear combination of other points in the subspace. More concretely, given a low rank matrix $\mathbf{X} = [\mathbf{x}_1, \ldots, \mathbf{x}_N]$ where $\mathbf{x}_i \in \mathbf{R}^d \, \forall \, i$. The goal of LRSC is to learn an affinity matrix $\mathbf{C} = [\mathbf{c}_1, \ldots, \mathbf{c}_N] \in \mathbf{R}^{N \times N}$ where each column $\mathbf{c}_i \in \mathbf{R}^N$ is the representation of $\mathbf{x}_i$ as a linear combination of other data points $\mathbf{x}_j$'s $\forall \, j$. More specifically each entry $\mathbf{C}_{ij}$ in the matrix $\mathbf{C}$ denotes the weight of $\mathbf{x}_j$ in the self-expressive reconstruction of $\mathbf{x}_i$.
A noiseless version of LRSC [13], henceforth called LRSC-Noiseless, aims to solve the objective in Equation 1.

$$\min_C \text{ rank(C) s.t. } X = XC. \implies C^* = V_1 V_1^T \text{ where } X = U_1 \Sigma_1 V_1^T$$

$$\text{Where } U_1 \Sigma_1 V_1^T \text{ is the Truncated SVD of X} \tag{1}$$

Our goal in utilising LRSC is to analyse and compare internal activations of neural networks over a set of $N$ data points in an architecture agnostic manner. Therefore, we utilise the noise-robust version of LRSC, as proposed in [13] and also shown in Equation 2. Utilising subspace clustering helps us learn a pairwise affinity kernel or a graph between $N$ data points. Doing so helps us represent every layer of a neural network as an $\mathbf{R}^{N \times N}$ matrix, which is architecture agnostic, thereby facilitating analysis and comparisons of different layers of same and different networks.

$$\min_C \|C\|_* + \frac{\tau}{2} \|X - XC\|_F^2 \quad \text{s.t.} \quad C = C^T. \implies C^* = V \mathcal{P}_{\frac{1}{\sqrt{\tau}}}(\Sigma) V^T = V_1 (I - \frac{1}{\tau} \Sigma_1^{-2}) V_1^T$$

$$\mathcal{P}_\epsilon(\sigma) = \begin{cases} 1 - \frac{\epsilon^2}{\sigma^2}, & \sigma > \epsilon \\ 0, & \sigma \leq \epsilon \end{cases} \text{ \& } X = U \Sigma V^T \text{ is the Full SVD}$$

$$\|C\|_* = \sum_i \sigma_i(C) \text{ - where } \sigma_i \text{ denotes the } i^{th} \text{ singular value of } C \tag{2}$$

$$\text{Note that } V_1 \text{ and } \Sigma_1 \text{ denote the truncated } V \text{ and } \Sigma \text{ based on diagonal 0's in } \mathcal{P}_{\frac{1}{\sqrt{\tau}}}(\Sigma)$$

## 3.2. Centered Kernel Alignment

Starting with [2], Centered Kernel Alignment or CKA ([3],[34]) has emerged has a key tool to analyse representations of Neural Networks ([7],[36],[37],[38]). Given 2 neural activation matrices of layer $i$ and $j$, namely $\mathbf{X} \in \mathbf{R}^{d_i \times N}$ and $\mathbf{Y} \in \mathbf{R}^{d_j \times N}$, Linear-CKA [2] computes their respective $\mathbf{R}^{N \times N}$ inner product kernels $\mathbf{K} = \mathbf{X}^T \mathbf{X}$ and $\mathbf{L} = \mathbf{Y}^T \mathbf{Y}$. It then utilises CKA to compute a similarity between two general kernels as shown in the Equation 3, where the equality on the left computes the CKA similarity between any pairwise similarity matrices $K$ and $L$. Similarly, the equality on the right, also called $CKA_{Lin}$, is a derived form of CKA for linear kernels $\mathbf{X}^T \mathbf{X}$ and $\mathbf{Y}^T \mathbf{Y}$ where $\lambda_X^i, \lambda_Y^j$ are the $i^{th}$ and $j^{th}$ squared singular vectors and $v_X^i, v_Y^j$ are the $i^{th}$ and $j^{th}$ right singular vectors of activation matrices $X$ and $Y$ respectively. Note that, HSIC or Hilbert Space Independence Criterion [39] used in Equation 3 is a way to compute the similarity between two $\mathbf{R}^{N \times N}$ kernel matrices and

serves as the backbone of CKA.

$$CKA(\mathbf{K}, \mathbf{L}) = \frac{HSIC(\mathbf{K}, \mathbf{L})}{\sqrt{HSIC(\mathbf{K}, \mathbf{K})HSIC(\mathbf{L}, \mathbf{L})}}, CKA_{Lin}(\mathbf{K}, \mathbf{L}) = \frac{\sum_{i=1}^{r_1} \sum_{j=1}^{r_2} \lambda_X^i \lambda_Y^j \langle v_X^i, v_Y^j \rangle^2}{\sqrt{\sum_{i=1}^{r_1} \left(\lambda_X^i\right)^2} \sqrt{\sum_{j=1}^{r_2} \left(\lambda_Y^j\right)^2}}$$

$$\text{where } HSIC(\mathbf{K}, \mathbf{L}) = \frac{tr(\mathbf{HKHHLH})}{(N-1)^2} \text{ and } \mathbf{H} = \mathbf{I} - \frac{1}{\mathbf{N}}\mathbf{11^T}$$

(3)

While for the purposes of this work we do refer to [2] as Linear-CKA, the authors of [2] also experiment with other Kernels like the Radial Basis Functions and demonstrate their effectiveness. [38] is another study in the line of analysing neural representations that studies the application of general non linear kernels to analyse neural representations with CKA.

# 4. Method

We now describe the methodologies used in this study to analyse neural networks. We begin by describing how we use LRSC based affinity matrices to compute LRSC-CKA and a Subspace Representation based classifier in Section 4.1 and Section 4.2, respectively. Then as a counter-part to Section 4.2 and analogous to the methodology adopted in [33] we define a Linear-CKA based classifier scheme in Section A.1. Lastly, in Section A.2 we describe the configurations and protocols for training followed in subsequent sections.

## 4.1. LRSC-CKA

---

**Algorithm 1:** All pairs CKA

**Data:** Activation Matrices: $[X_1, \ldots, X_l]$
**Result:** Affinity Matrices: $\mathbf{C} = [C_1, \ldots, C_l]$
Pairwise LRSC-CKA: $\mathbf{S} \in \mathbb{R}_+^{l \times l}$.

1 **initialization:** $\mathbf{C} = [\,]$;
2 $\mathbf{S} = \mathbf{0}$;
3 **for** $i \leftarrow 1$ **to** $l$ **do**
4     Given $X_i$, Compute $C_i$ based on Equation 1 or Equation 2 for LRSC-CKA or let $C_i = X_i^T X_i$ for Linear-CKA;
5     $\mathbf{C}$.append($C_i$);
6 **end**
7 **for** $i \leftarrow 1$ **to** $l$ **do**
8     **for** $j \leftarrow 1$ **to** $l$ **do**
9         $S_{ij}$ = CKA($C_i$,$C_j$) - Equation 4;
10         Use Equation 3 for Linear-CKA.
11     **end**
12 **end**

---

$$CKA_{LRSC}(\mathbf{C_X}, \mathbf{C_Y}) = \frac{\sum_{i=1}^{\tau_1} \sum_{j=1}^{\tau_2} \langle v_X^i, v_Y^j \rangle^2}{\sqrt{\tau_1}\sqrt{\tau_2}}$$

(4)

Based on discussions in Section 3 we frame LRSC-CKA as a spectral variant of Linear-CKA, an experimental analysis for establishing that is conducted in Section 5.1. Please note for all the results used throughout the paper we use Equation 2 to compute the LRSC Affinity Matrices, but for simplicity, let's consider a noiseless version of the problem described in Equation 1. Given neural activation matrices for layer $i$ and $j$ as $\mathbf{X} \in \mathbf{R}^{d_i \times N}$ and $\mathbf{Y} \in \mathbf{R}^{d_j \times N}$, we first compute their respective LRSC Affinity matrices denoted $\mathbf{C_X}$ and $\mathbf{C_Y}$ based on Equation 1. Based on the formula for Linear-CKA utilising the Singular Value Decomposition of activation matrices $\mathbf{X}$ and $\mathbf{Y}$ as shown in Equation 3, we write an analogous formula for LRSC-CKA in Equation 4 for low rank approximations of $\mathbf{X}$ with rank $\tau_1$ and $\mathbf{Y}$ with rank $\tau_2$. Unless otherwise stated, for all LRSC-CKA computations in this study we select the low rank $\tau$ as the number of components which explain 80% of the variance in the matrix. Using the noiseless variant of LRSC from Equation 1 allows us to more easily demonstrate that LRSC-CKA is a uniformly weighted sum of pairwise cosine similarities of top $\tau$ right singular

vectors of $\mathbf{X}$ and $\mathbf{Y}$. In contrast to Linear-CKA from Equation 3 this uniformity over a set of $\tau$ singular vectors ensures that LRSC-CKA is sensitive to changes beyond the dominant singular vectors, an issue that plagues Linear-CKA [11],[12] and algorithm 1 describes the process for computing LRSC-CKA for a given neural network.

## 4.2. Subspace representation based classification

Next, we describe subspace representation reconstruction (sub. recon.) based classification from [15],[16]. Given a point $\mathbf{x}_i \in \mathbb{R}^d$ and its self expressive encoding $\mathbf{c}_i \in \mathbb{R}^N$ learned via LRSC a per-class, reconstruction residual as defined in Equation 5. Once $\mathbf{r}_i^{(k)}$ for all classes have been computed, $\mathbf{x}_i$ is then assigned to the class, c, with the smallest residual norm $\|\mathbf{r}_i^{(c)}\|_2$. A higher value in this metric indicates a higher degree of co-planarity of a data point with respect to other points of the same class among different classes. Since LRSC encodes the degree of co-planarity between data points, layerwise LRSC-CKA is essentially a metric of similarity based upon co-planarity of data point $x_i$'s across various layers of a network. Computation of a subspace reconstruction based class label only requires an SVD of activations $X_l$ of a set of inputs for a given layer, which is obtained as a consequence of computing LRSC-CKA between any 2 layers. It doesn't require any additional training of linear classifiers for that layer's activations, thus making it a viable probe to evaluate linear structures in the activation space of a network.

The computation of subspace reconstruction based classification for every layer of the network is performed as follows - (1) Using algorithm 1 for LRSC computation we obtain the set of layerwise LRSC matrices $\{C_l\}$. (2) Given each $C_l \in \mathbb{R}^{N \times N}$ encodes the subspace representations for at network layer $l$ for inputs $x_1, \ldots, x_N$. For each input $x_i$ we compute the class-wise subspace residual $\mathbf{r}_i^{(k)}$ as defined in Equation 5 over all classes and assign it the label $c = \arg\min_k \|\mathbf{r}_i^{(k)}\|$ and do so for all inputs $i$ over all layers $l$.

# 5. Comparing Low Rank Subspace Clustering based CKA and Linear-CKA

Our goal is to analyse the role played by the singular value spectrum of activations of a given neural network and how different functions over the spectrum yield different interpretations. More specifically, as shown in Section 4.1 LRSC imposes a shrinkage operator like step-function over the singular values. Singular values below a certain rank are 0 and the rest are given an equal weight. Whereas by contrast as shown in Section 3.2, Linear-CKA squares the singular values of the representation matrices, which causes it to be more sensitive to singular vectors with high singular values, as shown in Section 5.1,[11] and [12]. A more analytical analysis of this fact is performed in Appendix F.

## 5.1. Spectral analysis of LRSC-CKA and Linear-CKA

When computing the similarity between neural activation matrices $X$ and $Y$, Linear-CKA computes a weighted average over the cosine similarities of left singular vectors of $X$ and $Y$ as shown in Equation 3 and (Noiseless) LRSC-CKA computes a uniformly weighted average of those components up to a certain rank. Recent works ([11],[38],[12]) have shown that Linear-CKA is mostly sensitive to changes in directions of topmost principal components and not sensitive to lower principal component deletion. We demonstrate that by the virtue of uniformly weighting cosine similarity's of principal components (PC), LRSC-CKA is sensitive to changes with greater uniformity. Similar to the protocol followed in [11] we describe the principal component (PC) sensitivity tests and present the results in Table 1.

Given the original neural activation matrix $X$ for a given layer and a set of its low rank representations $S$, we perform a spectral sensitivity analysis comparing LRSC-CKA and Linear-CKA along the lines of [11]. For the Top PC Addition Test in Table 1 the set $S$ consists of low rank representations starting with the first PC and going up to a representation that contains the top 50% PCs. The bottom PC Deletion Test starts with top 80% Principal Components and removes them down to top 30% PCs, the lowest 20% PCs are not used to maintain a parity for comparison. For the purpose of experimental validation we perform this analysis on the last 5 layers, just like in [11] and report the

average for each network. Given Low Rank Representations $S = \{X_\tau\}_{\tau_1}^{\tau_2}$, where $\tau_1$ and $\tau_2$ denote the start and end for number of principal components in the low rank representation. The Principal Component Sensitivity Test for a given layer is performed as follows -

1. Given the layer's neural activation matrix $X$, compute the linear probe accuracy, denoted $f(X)$, LRSC affinity matrix based on Equation 2 denoted by $C_X$ and Linear Kernel $K_X = X^T X$.
2. For each low rank representation $X_\tau \in S$:
   - Compute $f(X_\tau)$, $C_{X_\tau}$ and $K_{X_\tau}$ - The linear probe accuracy, LRSC Affinity and Linear Kernel of the said low rank representation.
   - Compute $|f(X) - f(X_\tau)|$, the difference in linear probe accuracies between the original representation and the low rank representation.
   - Compute $CKA(C_X, C_{X_\tau})$ and $CKA(K_X, K_{X_\tau})$, the LRSC-CKA and the Linear-CKA between the original and low rank representation.
3. Compute the Pearson's Correlation Coefficient $\rho$ between $|f(X) - f(X_\tau)|$ and $CKA(C_X, C_{X_\tau})$ or $CKA(K_X, K_{X_\tau})$ to compute the sensitivity for LRSC-CKA and Linear-CKA respectively. Please note that as 2 representations become similar, their CKA score will increase and the linear probe's accuracy difference between them will decrease, therefore we expect $\rho$ to be more negative in case of higher sensitivity.

We present the results of this procedure over 5 different random seeds of ResNet20 on CIFAR10 and CIFAR100 in Table 1. For each network we perform the Principal Component Sensitivity Test on the last 5 layers and compute Pearson's Correlation Coefficient for LRSC-CKA and Linear-CKA for each layer and show the mean and standard deviation. We observe that for Top PC Addition Test both LRSC-CKA and Linear-CKA are sensitive to changes in the Top most Principal Components. But for changes in lower principal components as demonstrated by the Bottom PC Deletion Test we observe that LRSC-CKA is much more sensitive than Linear-CKA. Therefore, LRSC-CKA has a higher sensitivity to change throughout the spectrum of an activation matrix as opposed to Linear-CKA, which is sensitive only to changes in the topmpost PCs [11],[38],[12]. A theoretical analysis of this phenomena is further presented in Appendix F.

| Top Principal Component Addition Sensitivity Test | | | | | | | | | | | |
|---|---|---|---|---|---|---|---|---|---|---|---|
| Setup $\rightarrow$ | | CIFAR10 Network R20 | | | | | CIFAR100 Network R20 | | | | |
| CKA $\downarrow$ | | V1 | V2 | V3 | V4 | V5 | V1 | V2 | V3 | V4 | V5 |
| $\rho$ - **LRSC** | $\mu$ | -0.88 | -0.9 | -0.88 | -0.9 | -0.89 | **-0.98** | **-0.98** | **-0.99** | **-0.98** | **-0.98** |
| | $\sigma$ | 0.07 | 0.05 | 0.06 | 0.05 | 0.07 | 0.009 | 0.004 | 0.005 | 0.007 | 0.005 |
| $\rho$ - **Linear** | $\mu$ | **-0.96** | **-0.96** | **-0.97** | **-0.97** | **-0.95** | -0.85 | -0.85 | -0.85 | -0.84 | -0.85 |
| | $\sigma$ | 0.04 | 0.03 | 0.02 | 0.04 | 0.05 | 0.14 | 0.13 | 0.15 | 0.15 | 0.14 |
| Bottom Principal Component Deletion Sensitivity Test | | | | | | | | | | | |
| Setup $\rightarrow$ | | CIFAR10 Network R20 | | | | | CIFAR100 Network R20 | | | | |
| CKA $\downarrow$ | | V1 | V2 | V3 | V4 | V5 | V1 | V2 | V3 | V4 | V5 |
| $\rho$ - **LRSC** | $\mu$ | **-0.93** | **-0.95** | **-0.93** | **-0.94** | **-0.93** | **-0.94** | **-0.95** | **-0.96** | **-0.95** | **-0.96** |
| | $\sigma$ | 0.02 | 0.01 | 0.01 | 0.02 | 0.02 | 0.03 | 0.02 | 0.01 | 0.02 | 0.009 |
| $\rho$ - **Linear** | $\mu$ | -0.51 | -0.53 | -0.62 | -0.44 | -0.45 | -0.53 | -0.55 | -0.55 | -0.57 | -0.56 |
| | $\sigma$ | 0.74 | 0.63 | 0.45 | 0.75 | 0.68 | 0.79 | 0.8 | 0.8 | 0.8 | 0.79 |

Table 1: Avg. Pearson Correlation Coefficients $\rho$ for Principal Components Additional and Deletion Tests for Linear-CKA and LRSC-CKA. 5 Networks with different initialisations used for each dataset, denoted V1-V5. This shall be the norm for using this notation for subsequent experiments unless otherwise stated.

As show above, the main advantage an LRSC framing of CKA instead of a Linear Kernel framing is that it helps us unlock more sensitivity to help detect changes in representations across a wider spectrum of their singular values, thus highlighting the main advantage of LRSC-CKA over Linear-CKA.

# 6. Subspace Analysis of Networks with LRSC-CKA

Having established similarities and differences between Linear-CKA and LRSC-CKA in Section 5 we now undertake a more detailed analysis of networks with LRSC-CKA. As a background, in Section 6.1, we establish the evolution of self-expressive structures throughout the network and measure the correlation of their functional performance, as defined in Section 4.2, to that of a respective linear-classifier. In Section 6.2 we further investigate the self-expressive structures present in network trained to memorise parts of their training input and corroborate the results established in Section 6.1. We also show that networks which memorise are similar to networks that generalise in all but the last few layers, indicating that changes as a result of noisy training needed to achieve memorisation might manifest more strongly in the latter layers of the network.

## 6.1. Self-Expressive Structures in the Latent Representations of Neural Networks

The LRSC Affinity matrix for each layer of a neural network encodes each input activation in terms of the Self-Expressive structures present in that representation. Computing the similarity between LRSC affinity matrices of any two layers using CKA thus provides a measure of similarity between the self-expressive structures encoded by the two representations. To supplement this analysis, we also explore the class label homogeneity/cohesiveness in composition of the self-expressive structures via a reconstruction classification as described in Equation 5, and we discuss those results next with additional results included in Appendix G.

In Figure 1 we plot the pairwise LRSC-CKA Heatmaps and layerwise subspace reconstruction based classification for ResNets trained on CIFAR10 and CIFAR100. We observe that as we go deeper in the network, subspace reconstruction based classification accuracy (blue line) rises rapidly, indicating that as we go deeper, a network separates out classes into separate subspaces. The subspace reconstruction based accuracy (blue line) also strongly correlates with the accuracy of a linear-classifier (orange line) trained on the representations of that layer, as shown in Table 2. That is, as we go deeper in the network, the data in addition to being more linearly separable w.r.t. class labels also becomes more self-expressive w.r.t. class labels. Such a high correlation demonstrates the predictive classification power encoded in self-expressive structures despite not necessarily being linearly-separable [14] and not being enforced during the training of the network.

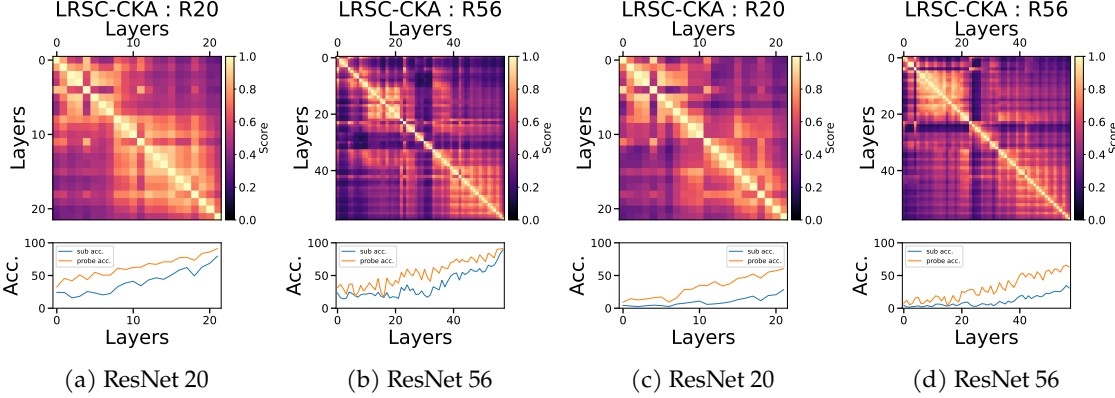

| (a) ResNet 20 | (b) ResNet 56 | (c) ResNet 20 | (d) ResNet 56 |

Figure 1: Comparison between layer-wise linear probe accuracy vs subspace reconstruction based classification accuracy on various networks trained on CIFAR 10 (Figure 1a-Figure 1b) and CIFAR100 (Figure 1c-Figure 1d). We observe that as we go deeper in the networks, the subspace reconstruction accuracy increases proportionally with the linear probe accuracy for those representations with correlations quantified in Table 2.

In Appendix E, we further explore the emergence of self-expressive structures in neural networks by comparing networks trained on the cross-entropy loss vs those trained on Maximum Coding Rate Reduction (MCRR) loss [22], a measure that ensures that features of data from different classes

| Pearson's Correlation $\rho$ - layer wise linear probe accuracy vs metrics | | | | | | | | |
|---|---|---|---|---|---|---|---|---|
| Datasets$\longrightarrow$ | CIFAR 10 | | | | CIFAR 100 | | | |
| Metric$\downarrow$ | R20 | R56 | R101 | R164 | R20 | R56 | R101 | R164 |
| **LRSC recon. acc.** | **0.940** | **0.909** | **0.920** | **0.883** | **0.905** | **0.931** | **0.942** | **0.888** |
| LRSC coeff. acc. | 0.933 | 0.897 | 0.897 | 0.851 | 0.902 | 0.917 | 0.918 | 0.871 |
| LinCKA coeff. acc. | 0.903 | 0.865 | 0.864 | 0.820 | 0.845 | 0.888 | 0.897 | 0.816 |

Table 2: Pearson's Correlation Coefficient $\rho$ between layer wise linear probe accuracies and LRSC-CKA and Linear-CKA metrics based accuracy for networks with different depths trained on CIFAR10 and CIFAR100.

belong to different linear subspaces, thereby encouraging the model to learn self-expressive structures.

Next in Section 6.2, a similar analysis on behaviour of self-expressive structures in networks which memorise a part of their training data is presented in Section 6.2. One of the goals in doing so is to establish that the performance correlation between linear probe accuracy and subspace reconstruction accuracy shown in Table 2 is not dependent on a network's generalisation ability, but a more robust phenomena.

## 6.2. Structure of Networks with data memorisation

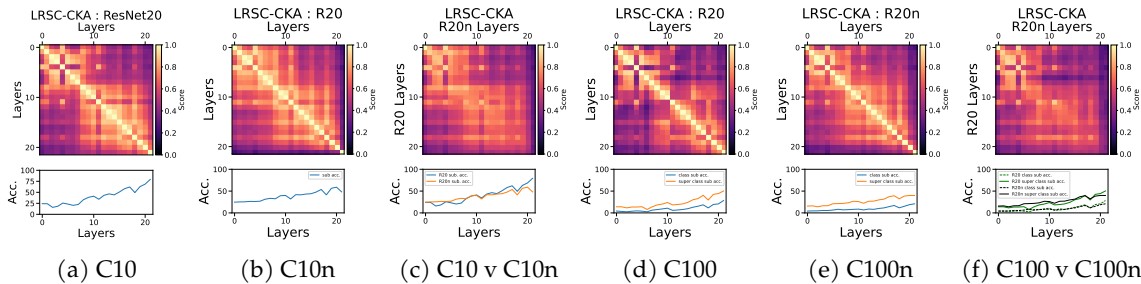

|   (a) C10   |   (b) C10n   |   (c) C10 v C10n   |   (d) C100   |   (e) C100n   |   (f) C100 v C100n   |

Figure 2: LRSC based comparison of ResNet20 when trained on CIFAR10 and CIFAR100 vs their Corrupted Versions. Figure 2a is the LRSC-CKA map for a normally trained ResNet on CIFAR10 and Figure 2b is the map for a network that memorises. Figure 2c offers a direct pairwise comparison of layers of the two networks with the normal network on the vertical axis and the over fitted network on the horizontal axis, here we clearly observe that the final few layers of the over fitted network share almost no similarity with any layers of the normally trained network. This is also accompanied by a dip in the subspace reconstruction based accuracy for those layers. An similar analysis using class and super class labels is done with CIFAR100 in Figure 2d - Figure 2f.

In this set of experiments we use LRSC-CKA to investigate the behaviour of neural networks when they have memorised parts of the training set and compare them with networks that offer good generalisation performance. To do so, we follow the experimental setup in [40],[41], and for the purpose of our study train networks with 50% of the training data labelled uniformly at random.

It is well known that neural networks can easily fit a random labeling of training data[42]. Given this observation our goal in conducting this experiments is to use LRSC-CKA similarity to understand differences between networks trained on randomly corrupted vs regular data. Our goal is to understand how such overfitting affects the manifold of the learned representations as measured by LRSC and how different are layers of a network which memorises w.r.t. a network that generalises well. Using subspace reconstruction based accuracy then further offers us another tool to observe the prediction dynamics and their differences in the internal layers of the network in each training regime.

We observe that networks with memorisation, Figure 2b, tend to learn similar representations when compared to networks with good generalisation, Figure 2a, for most of the network depth and differ

substantially in their later layers as shown in Figure 2c, while [43] demonstrate that memorisation is confined to a set of neurons rather than layers, observations similar to ours were also made in [18], [19], [20]. This phenomena is also highlighted by a decrease in the class-label homogeneity of the self-expressive structures, as shown in Figure 2c, of the 2 networks as they offer similar reconstruction based accuracy performance for all except the last few layers. Figure 2d - Figure 2f show a similar set of conclusions between networks which generalise and memorise on the CIFAR100 dataset. Figure 18 shows a similar analysis using Linear-CKA which also demonstrates that the major changes between the two types of networks appear towards the end of the networks, but a Linear-Kernel coefficient based classification methodology as described in Section A.1 isn't a reliable indicator of performance shift. A more comprehensive set of results demonstrating the differences between networks offering strong generalisation and memorisation performance while establishing their independence from network depth along with experimental setup details are described in Appendix H.

| Pearson's Correlation $\rho$ - layer wise linear probe accuracy vs metrics | | | | | | | | | | |
|---|---|---|---|---|---|---|---|---|---|---|
| Datasets→ | CIFAR 10 | | | | | CIFAR 100 | | | | |
| Metric ↓ | N1 | N2 | N3 | N4 | N5 | N1 | N2 | N3 | N4 | N5 |
| **LRSC recon. acc.** | 0.94 | **0.96** | **0.93** | **0.91** | **0.96** | 0.91 | **0.90** | 0.86 | **0.93** | **0.89** |
| LRSC coeff. acc. | **0.95** | 0.93 | **0.93** | 0.90 | **0.96** | **0.93** | 0.89 | **0.90** | 0.91 | **0.89** |
| LinCKA coeff. acc. | 0.83 | 0.86 | 0.88 | 0.78 | 0.87 | 0.80 | 0.77 | 0.76 | 0.78 | 0.76 |

Table 3: Pearson's Correlation Coefficient $\rho$ between layer wise linear probe accuracies and LRSC-CKA and Linear-CKA metrics based accuracy for networks trained on noisy datasets

Next, along the lines of Section 6.1, we establish the robustness of subspace reconstruction based classification as defined in Section 4.2 by correlating its performance with that of a linear classifier trained on intermediate layers of over fitted neural networks. We train different ResNets on CIFAR10 and CIFAR100 with 50% of the data randomly labelled, see Figure 3 and Figure 19, and measure the correlation of our metric with the accuracy of a linear classifier and present the results in Table 3.

The goal in doing so is to establish that the layer-wise correlations observed earlier in Section 6.1 are not dependent on an inherently well performing model. As shown in Table 3 the subspace reconstruction based label assignment, denoted by LRSC recon. acc., performs better than Linear-CKA coefficient based label assignment, indicating that the class cohesiveness of the self-expressive structures offers more insights into the generalisation performance than dot-products of activations from the same class. This establishes the subspace reconstruction approach as a valuable alternative to learning a linear classifier which first requires a computational overhead of training a classifier for all layers of the network as the subspace reconstruction based accuracy can be readily computed for any set of input activations. Additional results are presented in Appendix C, Appendix I, Appendix J, Appendix K and Appendix L .

## 7. Conclusion and Discussion

In this work we demonstrate that the use of self-expressive structures to understand the underlying geometry in representations of hidden layers of a neural network and its relation to previously well established methods. In doing so we use Low Rank Subspace Clustering (LRSC) on the activations of hidden layers of neural networks to encode each layer as a self-expressive affinity matrix which is architecture agnostic. We then use Centered Kernel Alignment(CKA) to compare affinity matrices of various layers of a network and across networks, and in doing so demonstrate that :

- We demonstrate that the combination of LRSC with CKA is an alternate spectral formulation of Linear-CKA which makes the similarity measure more sensitive to changes over a broader spectrum of principal components of the representations. Such a connection was lacking in prior working utilising subspace clustering [33, 44] to analyse representations .

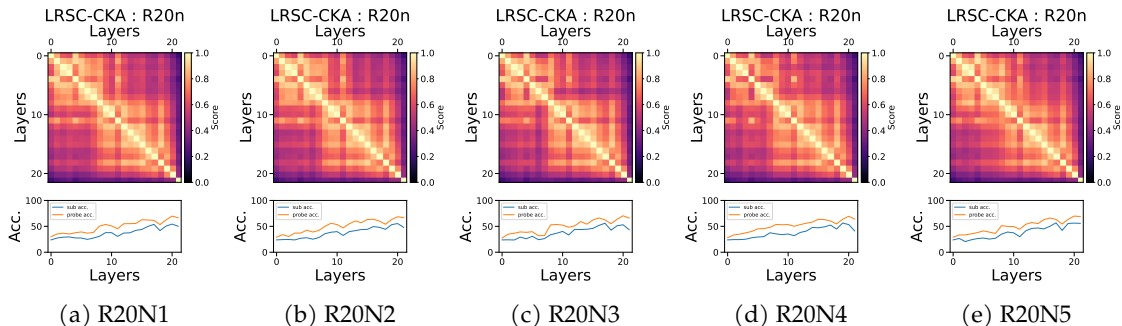

|           |           |           |           |           |
|-----------|-----------|-----------|-----------|-----------|
| (a) R20N1 | (b) R20N2 | (c) R20N3 | (d) R20N4 | (e) R20N5 |

Figure 3: Along with Table 3 here we establish a correlation between layer-wise LRSC subspace reconstruction accuracy vs linear probe based classification accuracy on the internal layers of networks trained on Noisy CIFAR 10 across different initialisations. The subspace reconstruction and linear probe accuracy for these over fitted networks continues to rise till the final few layers where it starts dropping along with the LRSC-CKA scores of those layers.

- Using LRSC-CKA we then demonstrate when compared to well trained networks, the networks which memorise parts of their training data tend to demonstrate significant differences in their final layer representations. Taking this further we also demonstrate that this phenomena tends to diminish upon the use of non linear activations like rational polynomials.
- We also demonstrate show that the predictive performance encoded in self-expressive structures strongly correlates which performance of a linear classifier trained on the same representations, irrespective of the networks generalisation ability.
- Finally we compare cross entropy objective based networks to networks trained on a coding rate objective, which encourage the separation of data into different subspaces. This is done to demonstrate that self-expressive structures emerge in networks trained with a cross-entropy objective, even when such constraints were not explicitly enforced.

# Acknowledgments

Research was supported by the National Science Foundation under CAREER grant no. IIS 2046086, grant no. CNS 2106982, and CREST Center for Multidisciplinary Research Excellence in CyberPhysical Infrastructure Systems (MECIS) grant no. 2112650.

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

# Appendix

## Table of Contents

# A. Method - Addendum

## A.1. Linear CKA Coefficient based classification

To devise an approximate analogue of subspace representation reconstruction from Section 4.2 for Linear-CKA, we follow affinity coefficient based classification as defined in [33, 44] to construct a linear kernel coefficient based classification heuristic for Linear-CKA. Unlike subspace reconstruction based metric defined in Section 4.2 which measures co-planarity of data points, a high Linear-CKA coefficient based classification accuracy only provides information about the cosine similarity of a given data point $x_i$ in terms of its class coherence. A high accuracy here indicates that most members of a class share a high cosine similarity, which is more limiting than subspace reconstruction based accuracy of Section 4.2 as high co-planarity doesn't need high cosine similarity. In Section 6 we test both subspace reconstruction and Linear-CKA Coefficient based classification schemes in terms of their correlation to a linear classifier and demonstrate the efficacy of subspace reconstruction based measure. The computations involved for Linear-CKA Coefficient based classification is as follows - (1) First similar to Section 4.2 we use algorithm 1 to compute Linear-CKA for a network and simultaneously obtain a layerwise set of Linear Kernels $\{K_i\}$. (2) Given a Linear-Kernel $K \in \mathbb{R}_{++}^{N \times N}$ of activation matrix $\mathbf{X} = [\mathbf{x}_1, \dots, \mathbf{x}_N]$ we compute the affinity of $x_i$ to class $k$ based on the formula in Equation 6 for inputs $i$ over all classes $k$ and assign $x_i$ the label $c = \arg\max_k \mathbf{a}_i^{(k)}$.

$$\mathbf{r}_i^{(k)} = \mathbf{x}_i - \sum_{\mathbf{x}_j \in \mathbf{X}^{(k)}} c_{ij}\mathbf{x_j}$$

$\mathbf{x}_j \in \mathbf{X}^{(k)}$ - the set of examples in class k.

$$\mathbf{a}_i^{(k)} = \frac{\sum_{\mathbf{x}_j \in \mathbf{X}^{(k)}} \langle \mathbf{x_i}, \mathbf{x_j} \rangle}{\sum_m \langle \mathbf{x_i}, \mathbf{x_m} \rangle}$$

(5)    $\mathbf{x}_j \in \mathbf{X}^{(k)}$ - the set of examples in class k.

(6)

## A.2. Experimental Setup

For the purpose of experiments conducted in this work, we train ResNets [6] on CIFAR10 and CIFAR100[2] using the code available here[3]. For ResNets trained on CIFAR10 and CIFAR100 with correct class labels we use a learning rate of 0.1 with a weight decay of 0.0001 trained for 164 epochs with a learning rate step size change milestones at epoch 81 and 122 with a gamma of 0.1. For the same ResNets but with a training regime around data memorisation or noisy labels as explained later in Section 6.2, we use a learning rate of 0.1 with a weight decay of 0.0005 and training time of 200 epochs with a cosine annealing based learning rate scheduler, similar to the setup followed in [40], both for CIFAR10 and CIFAR100. For Rational Neural Network [21] experiments on CIFAR10 in Appendix D we use the same parameters for correctly trained ResNets and for noisy training of Rational Neural Networks, we correspondingly use the same parameters as noisy ResNets as described earlier. For LRSC-CKA computations, unless otherwise stated we use a variance threshold of 80%. For training Linear classifiers on internal representations of networks the classifier size is the same as ambient dimension size and the learning rate is set to 0.001 with training for 164 epochs. For analysing ResNets in the Maximum Coding Rate Reduction (MCRR) [22] in Appendix E we follow the network, hyper-parameter and dataset setup provided by authors in their paper.

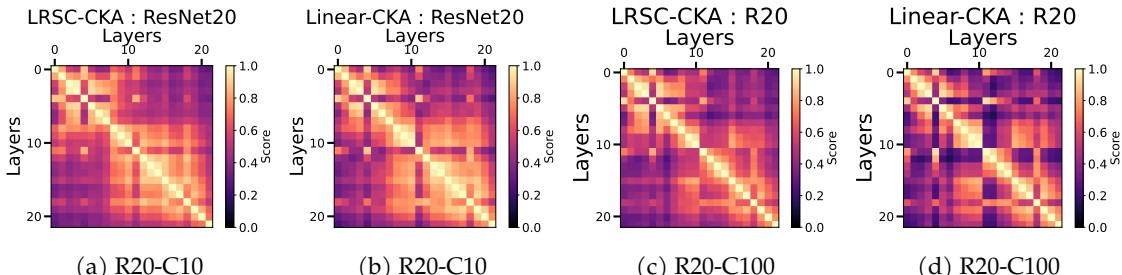

| (a) R20-C10 | (b) R20-C10 | (c) R20-C100 | (d) R20-C100 |

Figure 4: LRSC-CKA with varying levels of network depth to compare subspace evolution. Figure 4a and Figure 4b - CIFAR 10. Figure 4c and Figure 4d - CIFAR100. LRSC-CKA and Linear CKA discover a common set of high similarity layers in the network with a reduction in consensus arising for layer which share lower similarity.

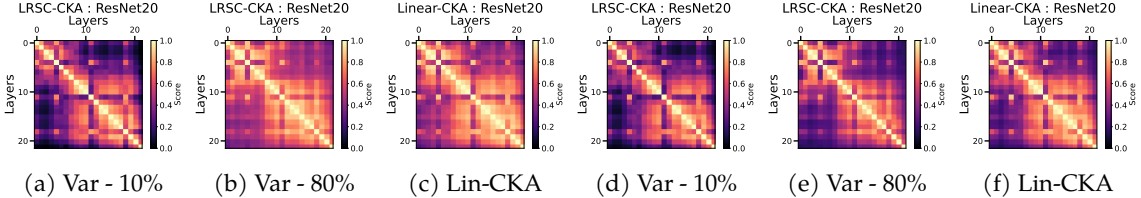

| (a) Var - 10% | (b) Var - 80% | (c) Lin-CKA | (d) Var - 10% | (e) Var - 80% | (f) Lin-CKA |

Figure 5: Figure 5a - Figure 5c shows LRSC-CKA with varying levels of variance preserved vs Linear-CKA on a CIFAR-10 set of 500 images. Figure 5d - Figure 5f shows LRSC-CKA with varying levels of variance preserved vs Linear-CKA on a CIFAR-10 set of 2500 images. Network ResNet20.

## B. Comparing Low Rank Subspace Clustering based CKA and Linear-CKA - Addendum

### B.1. A Simple comparison between LRSC-CKA and Linear-CKA

To begin, we show LRSC-CKA computations and corresponding Linear-CKA computations on ResNets trained on CIFAR10 and CIFAR100 and demonstrate the results in Figure 4, where Figure 4a is the LRSC-CKA of a network on CIFAR10 and Figure 4b is the corresponding Linear-CKA. Figure 4c and Figure 4d contain the corresponding CIFAR100 analysis. We observe that both LRSC-CKA and Linear-CKA discover largely similar similarity patterns with some differences about which regions of the network have a lower similarity. Next in Figure 5 we compare LRSC-CKA in a high and low variance setting across 2 mutually exclusive sample sizes on CIFAR10 with Linear-CKA. For the rest of the experiments we use the smaller probing set unless otherwise stated. We observe that when comparing a low variance LRSC-CKA setup as shown in Figure 5a and Figure 5d to Linear-CKA in Figure 5c and Figure 5f both setups indicate same regions of the network as one sharing a high similarity but slightly differ in their degree of dissimilarity for regions with lower CKA scores, with LRSC-CKA at 10% variance find virtually no similarity between initial and final layer and Linear-CKA finding modest similarity. Given the block-structure similarity between Linear-CKA and LRSC-CKA at such a low variance, it becomes clear that Linear-CKA pays a lot of attention to the topmost singular vectors, and any difference between this instantiation of LRSC-CKA and Linear-CKA is mostly along the rest of the singular vectors of the 2 representations, which is captured by a higher variance instantiation of LRSC-CKA as shown in Figure 5b and Figure 5e. An analogous analysis for CIFAR100 is performed in Figure 6.

---

[2]https://www.cs.toronto.edu/ kriz/cifar.html
[3]https://github.com/bearpaw/pytorch-classification

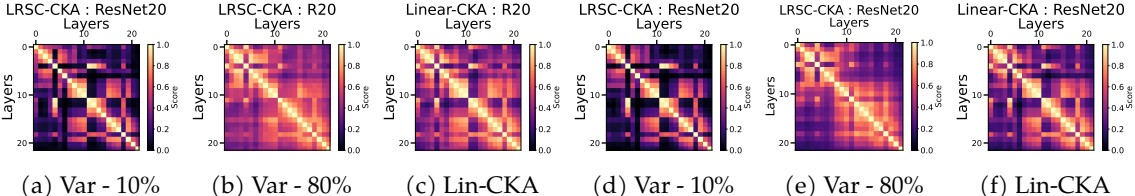

| (a) Var - 10% | (b) Var - 80% | (c) Lin-CKA | (d) Var - 10% | (e) Var - 80% | (f) Lin-CKA |

Figure 6: Figure 6a - Figure 6c shows LRSC-CKA with varying levels of variance preserved vs Linear-CKA on a CIFAR-100 set of 600 images. Figure 6d - Figure 6f shows LRSC-CKA with varying levels of variance preserved vs Linear-CKA on a CIFAR-100 set of 3000 images. Network ResNet20.

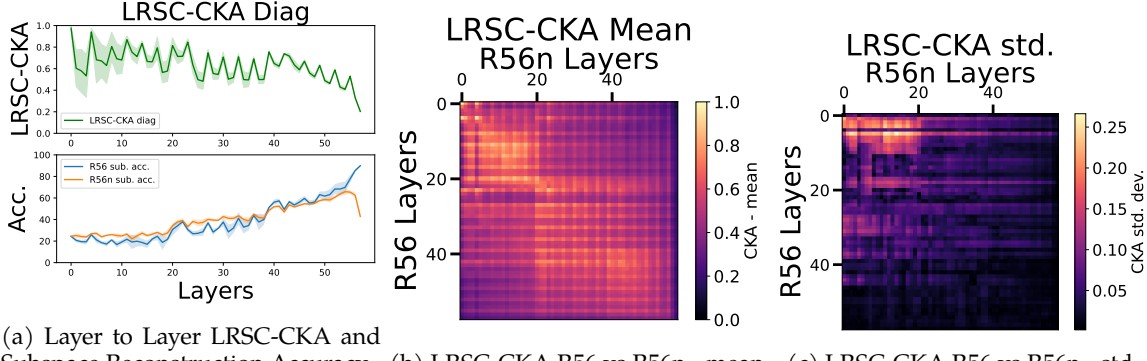

(a) Layer to Layer LRSC-CKA and Subspace Reconstruction Accuracy    (b) LRSC-CKA R56 vs R56n - mean    (c) LRSC-CKA R56 vs R56n - std.

Figure 7: Earlier analysis comparing networks with strong and poor generalisation, like in Figure 2, focused on a complete pairwise comparison of the 2 networks. Here we conduct a simpler (also very limiting) layer to layer comparison of the generalising networks with memorising networks in Figure 7a, it is also accompanied by a corresponding subspace reconstruction based analysis over the 2 types of networks wherein the earlier observations of differences between generalising and memorising networks of Section 6.2 are reaffirmed. All analysis presented in this figure are done over 5 pairs of generalising and memorising ResNet 56s trained on clean and noisy CIFAR 10 respectively. Corresponding complete layerwise LRSC-CKA analysis over these 5 pairs is shown in Figure 7b and Figure 7c, mean and standard deviation, respectively. Additional results are shown in Section H.5.

## C. Subspace Analysis of Networks with LRSC-CKA - Addendum

As a final note to conclude this analysis we add a secondary set of results focusing on a more narrow comparison between memorising and generalising networks wherein we only compare layers of networks with topological correspondence, i.e. layers with the same depth, in Figure 7. We note that this is a limited way to compare 2 neural networks as the representations of a layer of 1 network can share varying degrees of similarity to the corresponding topological neighbourhood of another network, as different networks may converge to different solutions at different layers. This point may especially be exacerbated by the fact that for our comparison both networks have been trained on different underlying data distributions, and are not guaranteed to have layer to layer correspondence.

### C.1. Corroboration on Mini Image Net 100

In this section we perform additional analysis along the lines of Section 6.1 and Section 6.2 on Mini-ImageNet [45]. In Figure 8 we perform analysis analogous to Figure 18 and demonstrate that for normally trained networks self-expressive structures emerge in the later layers, this is in consensus with results from Section 6.1. We also demonstrate that networks trained to memorise data begin to differ for normally trained in later layers, This conclusion is in consensus with experimental results

of Section 6.2. Additional experimental details and results accompanying Figure 8 are shown in Appendix J

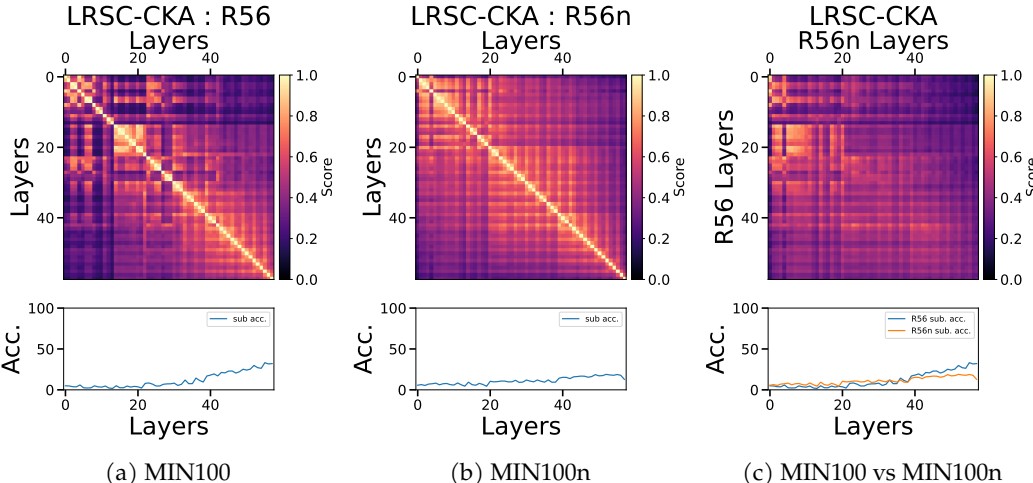

(a) MIN100         (b) MIN100n       (c) MIN100 vs MIN100n

Figure 8: Analogous to Figure 18 here we study the LRSC based comparison of ResNet56 when trained on a clean and noisy version of Mini Image Net 100. In Figure 8a we show the LRSC-CKA map for a ResNet56 trained on Mini Image Net 100 and Figure 8b for a ResNet56 overfitting on the noisy Mini Image Net 100. Same as Figure 18, Figure 8c offers a direct pairwise comparison of the two networks. We observe that the 2 networks share very little similarity towards the later layers, This is more concretely shown by a divergence in the subspace reconstruction based accuracy for those layers.

Additionally along the lines of Section 6.1 and Section 6.2, we also demonstrate the robustness and utility of subspace reconstruction based classification by evaluating its correlation to the generalisation of a linear probe trained on the same layer of the network. We do this to reassert the conclusions established for CIFAR10 and CIFAR100 in Section 6.1 and Section 6.2 on Mini Image-Net 100. In Table 4 we demonstrate the subspace reconstruction based accuracy correlates strongly with that of a linear classifier in networks trained on both well generalising and memorising regimes. This helps us establish its validity as a probing tool to evaluate neural network representations especially when compared to raw coefficients based approaches highlighted in Section A.1 and Row 2 and 3 of Table 4. Additional results for experiments in this section are presented in Appendix J.

| Pearson's Correlation $\rho$ - layer wise linear probe accuracy vs metrics | | | | | | | | | | |
|---|---|---|---|---|---|---|---|---|---|---|
| Datasets→ | | Mini Image Net 100 | | | | | Mini Image Net 100 (Noisy) | | | |
| Metric ↓ | V1 | V2 | V3 | V4 | V5 | N1 | N2 | N3 | N4 | N5 |
| **LRSC recon. acc.** | **0.97** | **0.97** | **0.96** | **0.97** | **0.97** | **0.9** | **0.90** | **0.92** | **0.92** | **0.93** |
| LRSC coeff. acc. | 0.96 | 0.94 | 0.95 | 0.9 | 0.94 | 0.89 | 0.87 | 0.91 | 0.9 | 0.89 |
| LinCKA coeff. acc. | 0.89 | 0.88 | 0.86 | 0.89 | 0.9 | 0.79 | 0.74 | 0.76 | 0.75 | 0.83 |

Table 4: Pearson's Correlation Coefficient $\rho$ between layer wise linear probe accuracies and LRSC-CKA and Linear-CKA metrics based accuracy for networks trained on clean and noisy Mini Image Net

# D. Limitations of probing with linear structures

In Section 6.2, we demonstrated that when comparing networks that generalise and memorise, the meaningful differences in learned representations only start to appear in the later layers. These findings were also observed in [19] using a Mean-Field Theoretic Manifold Analysis (MFTMA) technique [46],[47–49]. MFTMA computes Manifold Capacity ($\alpha_M$), which estimates the linear-separability of a set of manifolds by measuring the amount of class information embedded in given

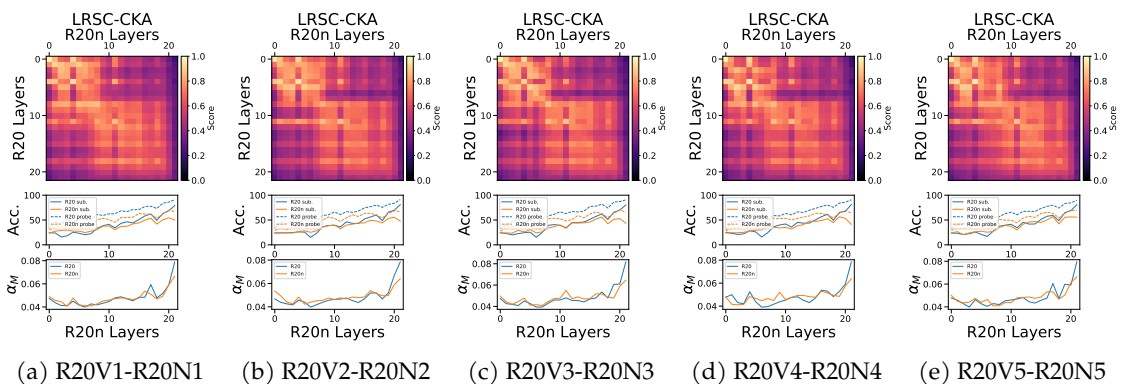

Figure 9: LRSC, Linear Probe and MFTMA based comparison between 5 pairs of ResNet 20 trained with ReLU Activations under normal and noisy label settings on the CIFAR10 dataset. We observe that when comparing a pair of normal and over fitted network for most of their depth, the subspace accuracy, the linear probe accuracy and Manifold capacity all demonstrate similar behaviour, with differences arising only in the last few layers.

set of points. A large value of $\alpha_M$ implies well-separated manifolds. More details can be found in Appendix M.

Next, as a sanity check, we observe the relationship between subspace reconstruction and linear probe accuracy with MFTMA - $\alpha_M$ on ResNets trained on CIFAR10 in normal and noisy regimes, just like in Section 6.2. For this purpose, we take pairs of ResNet-20 trained on CIFAR10 under normal and noisy settings and compare their layerwise behaviour in Figure 9. We observe that just as the layers between the two different networks start to diverge in their LRSC-CKA similarity, linear probe accuracy and the subspace reconstruction accuracy, manifold capacity $\alpha_M$ of the 2 networks starts to diverge as well with the network trained to memorise now exhibiting a lower value of $\alpha_M$, indicating reduced linear separability in final layer representations, this is expected and is along the lines of observations made in Section 6.2. Additionally as shown in Figure 9, the subspace reconstruction accuracy, linear probe accuracy and MFTMA Manifold capacity $\alpha_M$ between the layers of a normal and a noisy network are similar throughout its depth with the exception of a final few layers, where we see a deviation between a normal and an over fitted (noisy) network. The observations Figure 9 differ from the ones made in [43] where the authors observe that effects of memorisation aren't necessarily localised and a more complete set of counterpart results to Figure 9 are shown in Section M.1 and the corresponding Linear-CKA analysis is shown in Section M.3. Since, Subspace Reconstruction Accuracy, Linear Probe Accuracy and MFTMA all try to quantify the degree of linear separability of the underlying data manifold. In order to better understand the relation between the 3 quantities we compute correlations among them and present those in Table 5. We also augment Table 5 with similar correlations computed for Linear-CKA Coefficient based classification. We observe that while the subspace reconstruction and linear probe layerwise accuracies strongly correlate under both normal and noisy training regimes, the layerwise MFTMA is somewhat correlated with layerwise subspace reconstruction accuracy and weakly correlated with layerwise linear probe accuracy.

Next, we repeat the same set of experiments with ResNet-20s having rational polynomial activations based on [21] and compare the differences between models that generalise and memorise to better understand the differences in geometry of generalisation and memorisation for models with fundamentally different learning abilities. Rational Neural Networks [21] are networks with trainable activations that are low-degree rational polynomials. Composition of such functions offers good approximation power with a small computational overhead and [21] show that such networks require lesser depth than ReLU networks to approximate smooth functions. The LRSC-CKA similarity results for some pairs of normally and noisily trained rational resnets are presented in Figure 10. We observe that unlike ReLU based ResNets from Figure 9, the final layers

| Pearson's Correlation $\rho$ between the respective pairs of metrics | | | | | | | | | | | | |
|---|---|---|---|---|---|---|---|---|---|---|---|---|
| Datasets→ | CIFAR 10 | | | | | | CIFAR 10n | | | | | |
| Test ↓ | V1 | V2 | V3 | V4 | V5 | $\mu$ | N1 | N2 | N3 | N4 | N5 | $\mu$ |
| **sub. rec. vs probe** | 0.94 | 0.95 | 0.95 | 0.94 | 0.95 | **0.94** | 0.94 | 0.96 | 0.93 | 0.91 | 0.96 | **0.94** |
| Lin. coeff. vs probe | 0.9 | 0.9 | 0.89 | 0.92 | 0.88 | 0.9 | 0.83 | 0.86 | 0.88 | 0.78 | 0.87 | 0.84 |
| **sub. rec. vs MFTMA** | 0.82 | 0.83 | 0.82 | 0.77 | 0.82 | **0.81** | 0.71 | 0.59 | 0.55 | 0.56 | 0.69 | **0.62** |
| Lin. coeff. vs MFTMA | 0.85 | 0.86 | 0.87 | 0.75 | 0.87 | 0.84 | 0.83 | 0.75 | 0.77 | 0.75 | 0.83 | 0.79 |
| **probe vs MFTMA** | 0.7 | 0.7 | 0.7 | 0.62 | 0.68 | **0.68** | 0.68 | 0.53 | 0.65 | 0.67 | 0.66 | **0.64** |

Table 5: Pearson's Correlation Coefficient $\rho$ between layer wise linear probe accuracy, LRSC based subspace reconstruction accuracy, Linear-CKA coefficient based accuracy and MFTMA for ResNet-20s trained on clean and noisy CIFAR10 datasets. Pairwise correlations between subspace reconstruction, linear probe and MFTMA are highlighted in bold. We observe that subspace reconstruction based accuracy is strongly correlated with Linear Probe accuracy and is generally more correlated to Manifold Capacity than a Linear Probe would be, indicating that subspace reconstruction accuracy is a strong proxy for testing linear-separability of manifolds.

of Rational ResNets as shown in Figure 10a-Figure 10e under normal and noisy regimes don't show significant dissimilarities, even though the performance of the normal and the noisy networks are very different. This also translates to similar, though still divergent subspace reconstruction and linear probe accuracies when comparing the final few layers of the normal and noisy networks. This behaviour is also reflected in MFTMA $\alpha_M$, which indicates that linear separability of the data manifolds is similar between the 2 regimes. This demonstrates that in memorisation regimes changes induced due to non-linear activations lead to structures whose projections on lower dimensional subspaces is similar to networks trained in a generalisation regimes, thereby indicating that the use of non-linear activations helps the model learn structures which are not easily resolvable with linear models. A more comprehensive set of companion results to Figure 10 is shown in Figure 56 of Section M.2. Additionally, just as was done for ReLU ResNets in Table 5, we also perform a similar analysis to measure the correlations between Subspace Reconstruction Accuracy, Linear Probe Accuracy and MFTMA and present the results in Table 6. We observe the same correlation strengths as before with Subspace Reconstruction and Linear Probe Accuracy being the most strongly correlated, whereas MFTMA being mildly correlated to the subspace reconstruction metric and being very weakly correlated with linear probe performance.

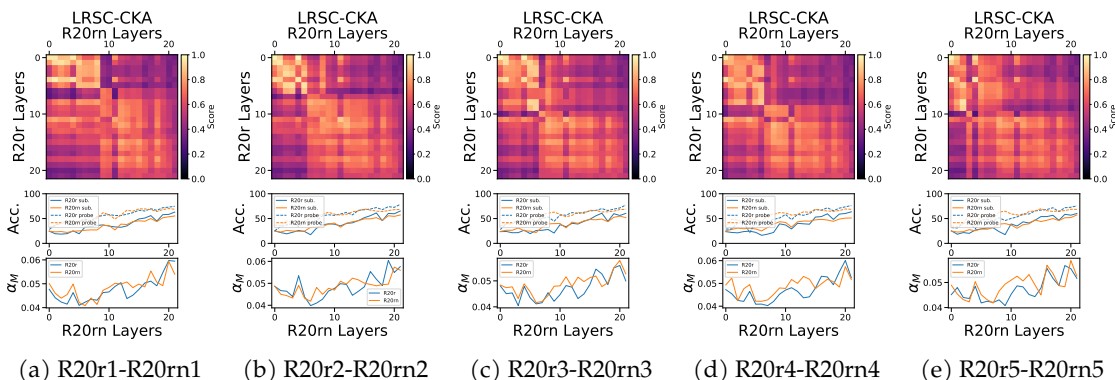

(a) R20r1-R20rn1 (b) R20r2-R20rn2 (c) R20r3-R20rn3 (d) R20r4-R20rn4 (e) R20r5-R20rn5

Figure 10: Along the lines of Figure 9 a similar comparison involving LRSC, Linear Probe and MFTMA between pairs ResNet 20 trained with Rational Polynomial Activations under normal and noisy label settings on the CIFAR10 dataset. We observe that despite the differences in generalisation ability of a normal and an over fitted network they have similar trajectory of metrics throughout their depth. Indicating that the differences between a normal and noisily trained network manifests along more non-linear manifolds in rational networks.

| Pearson's Correlation $\rho$ between the respective pairs of metrics | | | | | | | | | | | | |
|---|---|---|---|---|---|---|---|---|---|---|---|---|
| Datasets→ | CIFAR 10 | | | | | | CIFAR 10n | | | | | |
| Test ↓ | R1 | R2 | R3 | R4 | R5 | $\mu$ | RN1 | RN2 | RN3 | RN4 | RN5 | $\mu$ |
| **sub. rec. vs probe** | 0.91 | 0.93 | 0.9 | 0.9 | 0.91 | **0.91** | 0.92 | 0.92 | 0.92 | 0.94 | 0.85 | **0.91** |
| Lin. coeff. vs probe | 0.87 | 0.83 | 0.81 | 0.87 | 0.88 | 0.85 | 0.82 | 0.8 | 0.75 | 0.84 | 0.72 | 0.78 |
| **sub. rec. vs MFTMA** | 0.81 | 0.71 | 0.8 | 0.85 | 0.76 | **0.79** | 0.72 | 0.7 | 0.67 | 0.54 | 0.58 | **0.64** |
| Lin. coeff. vs MFTMA | 0.84 | 0.68 | 0.78 | 0.79 | 0.77 | 0.77 | 0.65 | 0.65 | 0.66 | 0.46 | 0.53 | 0.59 |
| **probe vs MFTMA** | 0.69 | 0.55 | 0.6 | 0.66 | 0.61 | **0.62** | 0.58 | 0.6 | 0.53 | 0.48 | 0.4 | **0.52** |

Table 6: Analogous to Table 5 here we present the Pearson's Correlation Coefficient $\rho$ between layer wise linear probe accuracy, LRSC based subspace reconstruction accuracy, Linear-CKA coefficient based accuracy and MFTMA for Rational ResNet-20s trained on clean and noisy CIFAR10 datasets.

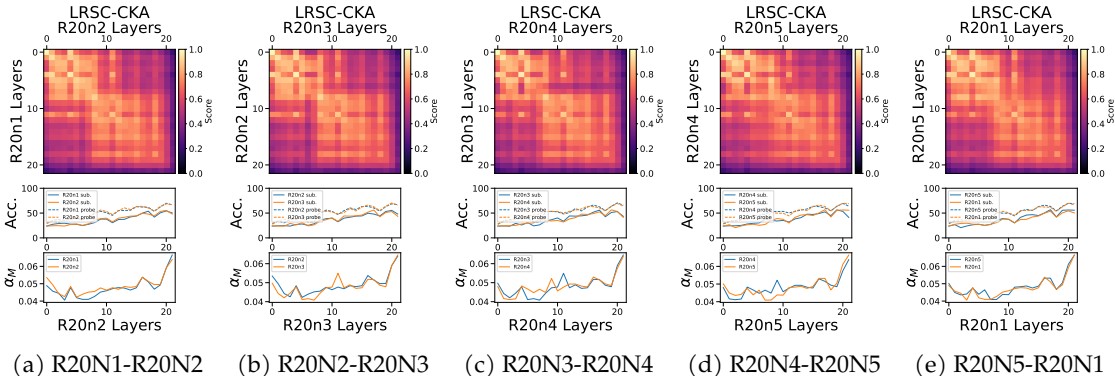

(a) R20N1-R20N2   (b) R20N2-R20N3   (c) R20N3-R20N4   (d) R20N4-R20N5   (e) R20N5-R20N1

Figure 11: LRSC-CKA based pairwise comparison of 5 ReLU ResNets trained on noisy CIFAR10

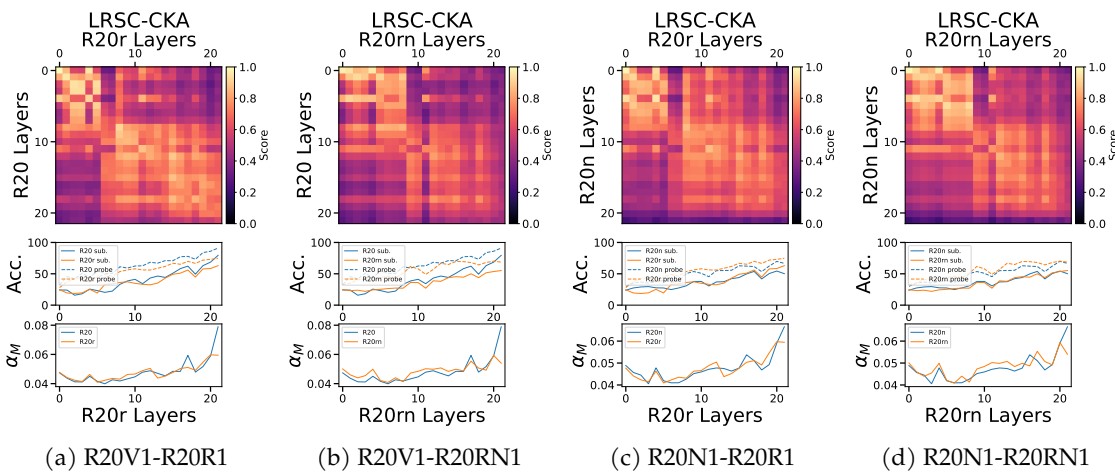

(a) R20V1-R20R1   (b) R20V1-R20RN1   (c) R20N1-R20R1   (d) R20N1-R20RN1

Figure 12: Cross Architectural LRSC-CKA based comparison of ResNet 20 trained with ReLU and Rational Polynomial Activations under normal and noisy label settings on the CIFAR10 dataset. Figure 12c and Figure 12d demonstrate that a noisy ReLU Network learns different representations in its final layers.

To further consolidate our understanding of limitations of network analysis with linear structures, we compare ReLU ResNets and Rational ResNets trained in clean and noisy regimes and present the results in Figure 12 for LRSC-CKA. In Figure 12a we compare a normally trained ReLU ResNet (R20) to a normally trained Rational ResNet (R20r) and observe that both networks learn similar representations along their depth, with slightly higher final layer linear probe and subspace reconstruction based accuracies for the ReLU network. Interestingly this is also accompanied a corresponding difference in layerwise manifold capacity of $\alpha_M$ with the ReLU network showing a higher $\alpha_M$ than

the rational network, therefore providing supplementary evidence for increased linear separation in case of ReLU networks.

Next, comparing the same ReLU resnet (R20) with a noisily trained rational resnet (R20rn) in Figure 12b we observe that last layers of both ResNets aren't similar to the same degree as was the case earlier in Figure 12a when both networks were normally trained. This also indicated by diverging subspace reconstruction and linear probe accuracies and much lower $\alpha_M$ for noisy rational ResNet than normal ReLU ResNet. Next we proceed to analyse the final two combinations and compare a noisily trained ReLU ResNet with a normal and noisy Rational ResNet. In Figure 12c we compare the noisily trained ReLU ResNet(R20n) to a normally trained Rational ResNet(R20r), analogous to the observations that were made in Section 6.2, we observe that the last layers of noisy ReLU ResNet(R20n) are very dissimilar to all layers of Rational ResNet(R20r) and as one would expect the subspace reconstruction and linear probe performance of the noisy relu resnet is lower than that of the normally trained rational network but the underlying manifold is still more linearly separable. Just like how the normally trained relu resnet shared similarities with all the layers of a similarly trained rational resnet as shown earlier in Figure 12a, the noisy relu resnet does the same but for the last layers, thereby also indirectly offering a corroboration of results in Section 6.2 where we demonstrated that normally and noisily trained relu networks tend to differ only in the later layers. Finally comparing noisily trained versions of both networks in Figure 12d we again observe that the final layers of noisy relu resnet are not similar to any layer of its noisy rational counterpart. Even though the linear separability ($\alpha_M$) of manifolds in the final layer representations are different, both the networks exhibit similar linear probe and subspace reconstruction accuracies. The noisy rational network doesn't show a similar behaviour, its final layers are similar to various layers of a noisy ReLU ResNet. Figure 11 compares 5 noisy relu networks one by one, and it clearly demonstrates that the last few layers of each network is dissimilar from the rest. Based on these observations, the set of experiments described in this section clearly establish that structures learnt by rational networks when trained to fit noisy training data are completely different to those learnt by ReLU networks and Linear Probes, Subspace Reconstruction and MFTMA are less efficient at discovering the differences between generalising and memorising geometries in rational neural networks. Additional results for experiments in this section are presented in Appendix M and Appendix N.

# E. Analysis of Networks trained with subspace separation loss vs classification loss

To better observe the emergence of class homogeneous self-expressive structures in deeper layers of a network we compare networks trained on cross-entropy (CE) with networks trained on Maximum Coding Rate Reduction[22] (MCRR), which we describe next for completeness.

Given a dataset $\mathbf{X} = [\mathbf{x}_1, \ldots, \mathbf{x}_N] \in \mathbb{R}^{d \times N}$ coming from a disjoint union of manifolds where $\mathcal{M} = \sqcup_{i=1}^{k} \mathcal{M}_i$ in ambient space $\mathbb{R}^d$ and a network $f(\mathbf{x}, \theta) : \mathbb{R}^d \to \mathbb{R}^p$, the Maximum Coding Rate Reduction(MCRR) [22] training framework learns a mapping $\mathbf{z} = f(\mathbf{x}, \theta) \in \mathbb{R}^p$ such that $\mathbf{Z} = [\mathbf{z}_1, \ldots, \mathbf{z}_N] \in \mathbb{R}^{p \times N}$ belongs to a disjoint union of linear subspaces $\mathcal{S} = \sqcup_{i=1}^{k} \mathcal{S}_i$ in ambient space $\mathbb{R}^p$. The MCRR training framework encourages the following properties - (1) Representations for inputs from different classes are uncorrelated and belong to different linear subspaces. (2) Representations for inputs from the same class are correlated and belong to the same linear subspace. (3) The dimension or volume of the space occupied by inputs from a class should be as large as possible as long as they are uncorrelated with the rest.

Works like [50],[51],[52],[53] try to enforce the self-expressive property in the learned representations but cannot ensure all the 3 previously listed properties in the learned representation. Given data samples $\mathbf{X} = [\mathbf{x}_1, \ldots, \mathbf{x}_N]$ and a network $f(\mathbf{x}, \theta)$ where $\mathbf{z}_i = f(\mathbf{x}_i, \theta)$ is the learned representation for $\mathbf{x}_i$, thereby creating a learned representation matrix $\mathbf{Z} = [\mathbf{z}_1, \ldots, \mathbf{z}_N]$ encoded each input data point. According to [54] the total number of bits needed to encode $\mathbf{Z}$ up to a precision $\epsilon$ on a per input formulation is defined in Equation 7.

$$\mathcal{R}(\mathbf{Z}, \epsilon) = \left(\frac{1}{2}\right) \log \det \left(\mathbf{I} + \frac{p}{n\epsilon^2} \mathbf{Z}\mathbf{Z}^T\right) \quad (7) \qquad \mathcal{R}^c(\mathbf{Z}, \epsilon|\mathbf{\Pi}) = \sum_{j=1}^{k} \frac{tr(\mathbf{\Pi}_j)}{2N} \log \det \left(\mathbf{I} + \frac{p}{tr(\mathbf{\Pi}_j)\epsilon^2} \mathbf{Z}\mathbf{\Pi}_j\mathbf{Z}^T\right)$$
$$(8)$$

For $\mathbf{Z}$ belonging to multiple classes such that $\mathbf{Z} = [\mathbf{Z}_1, \ldots, \mathbf{Z}_K]$ where each $\mathbf{Z}_i \in \mathcal{S}_i$. Let $\mathbf{\Pi} = \{\mathbf{\Pi}_j \in \mathbb{R}^{N \times N}\}_{j=1}^{k}$ be a set of diagonal matrices which encode class membership information of all N samples. Then the average number of bits per sample which respects the partition $\mathbf{\Pi}$ based on [54] is defined as in Equation 8.

As mentioned in the desiderata for the MCRR training framework, features from different classes should be uncorrelated and thus span the largest possible volume of the space, implying that the coding rate of the entire set $\mathbf{Z}$ should be high. Whereas features from the same class should occupy a smaller volume as they should be highly correlated to each other. Therefore learning a representation $\mathbf{Z}$ from $\mathbf{X}$ given a partition $\mathbf{\Pi}$ involves maximising the difference between the coding rate for the full dataset and all class subsets. This formulation known as MCRR is summarised in Equation 9.

$$\max_{\mathbf{Z}, \mathbf{\Pi}} \Delta R(\mathbf{Z}, \mathbf{\Pi}, \epsilon) = \mathcal{R}(\mathbf{Z}, \epsilon) - \mathcal{R}^c(\mathbf{Z}, \epsilon|\mathbf{\Pi})$$
$$\text{where } \|\mathbf{z}_i\|_2^2 = 1 \ \forall i \qquad (9)$$
$$\text{and } \mathbf{\Pi} = \{\mathbf{\Pi}_j \in \mathbb{R}^{N \times N}\}_{j=1}^{k}$$

|     |     |     |     |     |     |
| --- | --- | --- | --- | --- | --- |
| (a) CE | (b) MCRR | (c) MCRR - CE | (d) CE | (e) MCRR | (f) MCRR - CE |

Figure 13: LRSC-CKA Analysis of a ResNet 18 trained with Maximal Coding Rate Reduction (MCRR) and Cross Entropy (CE) loss on CIFAR10 and CIFAR100. Figure 13a and Figure 13b show the LRSC-CKA for networks trained on CIFAR10 with Cross Entropy Loss and MCRR Loss respectively. For MCRR loss in Figure 13b the layers of the network are divided into 2 stages, with the latter stage exhibiting a high subspace reconstruction accuracy, indicating the presence of self-expressive structures. Figure 13c shows a comparison between the 2 networks demonstrating the emergence of self expressive structures in final layers of a cross entropy network as indicated by its similarity to the second stage of MCRR layers along with a high subspace reconstruction accuracy. Corresponding analysis for CIFAR100 is shown in Figure 13d - Figure 13f.

Given that class coherent self-expressive structures emerge in networks trained on cross-entropy (CE) loss, our goal is to compare and contrast such a network to one trained with MCRR loss and analyse the representations learned in the 2 frameworks. To do so we follow the experimental setup in the original work on MCRR Loss [22] and train ResNets with cross-entropy loss and MCRR loss on CIFAR10 and CIFAR100 using super class labels. Our analysis with LRSC-CKA presented in Figure 13 finds that networks trained with MCRR Loss exhibit a 2 stage partition of layers, with layers of a stage having high intra-stage similarity while exhibiting extremely low inter-stag similarity. As shown by the evolution of layer wise subspace reconstruction accuracy in Figure 13b and Figure 13e, the second stage of layers in the MCRR network corresponds to layers which separates out data into different subspaces based on their class - indicated by a higher subspace reconstruction based accuracy. When compared with networks trained on the CE loss we observe that layers in the first stage of blocks in a network trained with MCRR loss are similar to most layers of the network trained with CE loss and the layers in the second stage of the MCRR trained network shares some similarity with those final few layers of the CE trained network whose subspace reconstruction based accuracy is

high, indicating the emergence of class coherent self-expressive structures. This analysis establishes the emergence of class coherent self-expressive structures in networks trained with CE loss and also indicates that regardless of the training objective large parts of the network learn representations that are very similar, with meaningful differences emerging only much later in the network. [55] also showed a similar late divergence of representation in networks trained with different classification losses and [10] on the other hand demonstrates that in the case of self-supervised training of Vision Transformers [5] the choice of objectives, namely, Joint-Embedding [56],[57] vs Reconstruction based learning [58],[59] leads to dissimilar features that appear quite early in a network.

A more complete set of results for LRSC-CKA and Linear-CKA along with the details of the experimental setup is provided in Appendix O.

# F. Connection between LRSC-CKA and Linear-CKA

Linear-CKA and LRSC-CKA are two versions of weighted sums of cosine similarity between the right-singular vectors of the original representations. Given activation matrices of layer $i$ and $j$, namely $\mathbf{X} \in \mathbf{R}^{d_i \times N}$ and $\mathbf{Y} \in \mathbf{R}^{d_j \times N}$, CKA [2] computes their similarity via Equation 3. For given 2 layer wise neural activation matrices $X = U_X \Sigma_X V_X^T$ and $Y = U_Y \Sigma_Y V_Y^T$, which are centred, we first demonstrate why Linear-CKA [2] is more sensitive to first few principal components in Section F.1 and then we demonstrate how Linear-CKA [2] is related to LRSC-CKA in Section F.2 while also showing how LRSC-CKA alleviates some shortcomings of Linear-CKA [2].

## F.1. Analysis of Linear-CKA

Given centred neural activation matrices $X = U_X \Sigma_X V_X^T$ and $Y = U_Y \Sigma_Y V_Y^T$, where each column of the matrix is the representation for a data point. Linear-CKA [2] then requires the computation of Linear-Kernel Gram Matrices as shown in Equation 10.

$$X^T X = V_X \Sigma_X^2 V_X^T \text{ and } Y^T Y = V_Y \Sigma_Y^2 V_Y^T. \tag{10}$$

Re-writing Equation 10 as follows,

$$X^T X = V_X \Lambda_X V_X^T \text{ and } Y^T Y = V_Y \Lambda_Y V_Y^T. \text{ Where } \Lambda = \Sigma^2. \tag{11}$$

From the first part of Equation 3 and Equation 11.

$$CKA(\mathbf{K}, \mathbf{L}) = \frac{HSIC(\mathbf{K}, \mathbf{L})}{\sqrt{HSIC(\mathbf{K}, \mathbf{K})HSIC(\mathbf{L}, \mathbf{L})}} \tag{12}$$

Where,

$$HSIC(\mathbf{K}, \mathbf{L}) = \frac{tr(HKHHLH)}{(N-1)^2} \tag{13}$$

$$H = I - \frac{1}{N}\mathbf{1}\mathbf{1}^T \tag{14}$$

Letting $\mathbf{K} = X^T X$ and $\mathbf{L} = Y^T Y$. As $X$ and $Y$ are already centred, Computing the 3 Hilbert Space Independence Criterion (HSIC) values. Computing the numerator of Equation 12,

$$
\begin{aligned}
HSIC(\mathbf{X^T X}, \mathbf{Y^T Y}) &= \frac{tr(HX^T XHHY^T YH)}{(N-1)^2} \\
HSIC(\mathbf{X^T X}, \mathbf{Y^T Y}) &= \frac{tr(X^T XY^T Y)}{(N-1)^2} \\
HSIC(\mathbf{X^T X}, \mathbf{Y^T Y}) &= \frac{tr(V_X \Lambda_X V_X^T V_Y \Lambda_Y V_Y^T)}{(N-1)^2} \\
HSIC(\mathbf{X^T X}, \mathbf{Y^T Y}) &= \frac{tr(\Lambda_X V_X^T V_Y \Lambda_Y V_Y^T V_X)}{(N-1)^2} \\
HSIC(\mathbf{X^T X}, \mathbf{Y^T Y}) &= \frac{\sum_{i=1}^{r_1} \sum_{j=1}^{r_2} \lambda_X^i \lambda_Y^j \langle v_X^i, v_Y^j \rangle^2}{(N-1)^2}
\end{aligned}
\tag{15}
$$

Computing the denominator of Equation 12,

$$
\begin{aligned}
HSIC(\mathbf{X^T X}, \mathbf{X^T X}) &= \frac{tr(X^T XX^T X)}{(N-1)^2} \\
HSIC(\mathbf{X^T X}, \mathbf{X^T X}) &= \frac{tr(V_X \Lambda_X V_X^T V_X \Lambda_X V_X^T)}{(N-1)^2} \\
HSIC(\mathbf{X^T X}, \mathbf{X^T X}) &= \frac{tr(\Lambda_X V_X^T V_X \Lambda_X V_X^T V_X)}{(N-1)^2} \\
HSIC(\mathbf{X^T X}, \mathbf{X^T X}) &= \frac{\sum_{i=1}^{r_1} \left( \lambda_X^i \right)^2}{(N-1)^2}
\end{aligned}
\tag{16}
$$

A similar compution with matrix $Y$ yields,

$$HSIC(\mathbf{Y^T Y}, \mathbf{Y^T Y}) = \frac{\sum_{j=1}^{r_2} \left(\lambda_Y^i\right)^2}{(N-1)^2} \tag{17}$$

Combining Equation 15, Equation 16 and Equation 17 yields the formula for Linear-CKA [2] in terms of eigen-decomposition of the linear kernels of respective neural activation matrices, as shown in Equation 3 and Equation 18.

$$CKA_{Linear}(\mathbf{X}^T\mathbf{X}, \mathbf{Y}^T\mathbf{Y}) = \frac{\sum_{i=1}^{r_1}\sum_{j=1}^{r_2} \lambda_X^i \lambda_Y^j \langle v_X^i, v_Y^j \rangle^2}{\sqrt{\sum_{i=1}^{r_1} \left(\lambda_X^i\right)^2}\sqrt{\sum_{j=1}^{r_2} \left(\lambda_Y^j\right)^2}} \tag{18}$$

Works like [60] empirically demonstrate that the eigen-values of real world data and kernel matrices tend to decay rapidly. [61] show that data that can derived from a latent variable model can be approximated by a low rank matrix, the proof of which is detailed in Section F.3. [62] further provide bounds on the Singular Values of matrices with Displacement Structure and demonstrate exponential decay of singular values.

For the purpose of our analysis of Linear-CKA [2] we adopt a simplified exponential decay model over singular values from [63], whereas more involved results exist in [62].

In an exponential decay model [63], we assume that given an eigen-decomposition of the linear kernel matrix, its $i^{th}$ eigen-value $\lambda_i = \mathcal{O}(\rho^{\beta i})$, where $\rho < 1$. More concretely, for linear kernels,

$$\text{Given any activation's linear kernel matrix } X^T X = V\Sigma^2 V^T, \text{ let } \lambda_i = \lambda_1 \rho^{i-1} \tag{19}$$

Computing the sum of square of eigen values of any $X^T X$,

$$\sum_{i=1}^{n} \left(\lambda_i\right)^2 = \lambda_1^2 + \lambda_2^2 + \cdots + \lambda_n^2$$

$$\sum_{i=1}^{n} \left(\lambda_i\right)^2 = \lambda_1^2 + \lambda_1^2\tau + \cdots + \lambda_1^2\tau^{n-1} \text{ , where } \tau = \rho^2 \ll 1$$

$$\sum_{i=1}^{n} \left(\lambda_i\right)^2 = \lambda_1^2(1 + \tau + \cdots + \tau^{n-1}) \tag{20}$$

$$\sum_{i=1}^{n} \left(\lambda_i\right)^2 = \lambda_1^2 \frac{1 - \tau^n}{1 - \tau}$$

$$\sum_{i=1}^{n} \left(\lambda_i\right)^2 \approx \lambda_1^2 \frac{1}{1 - \tau}$$

As a consequence of Equation 20,

$$\frac{\lambda_1^2}{\sum_{i=1}^{n} \left(\lambda_i\right)^2} \approx \frac{\lambda_1^2(1 - \tau)}{\lambda_1^2}$$

$$\frac{\lambda_1^2}{\sum_{i=1}^{n} \left(\lambda_i\right)^2} \approx 1 - \tau \tag{21}$$

$$\frac{\lambda_1}{\sqrt{\sum_{i=1}^{n} \left(\lambda_i\right)^2}} \approx \sqrt{1 - \tau} \text{ , where } \tau \ll 1$$

Therefore, substituting the result in Equation 21 into the summation for $i = 1$ and $j = 1$ in Equation 18, we obtain -

$$\frac{\lambda_X^1 \lambda_Y^1 \langle v_X^1, v_Y^1 \rangle^2}{\sqrt{\sum_{i=1}^{r_1} \left(\lambda_X^i\right)^2}\sqrt{\sum_{j=1}^{r_2} \left(\lambda_Y^j\right)^2}} \approx \sqrt{1 - \tau_X}\sqrt{1 - \tau_Y}\langle v_X^1, v_Y^1 \rangle^2 \tag{22}$$

Similarly, In a polynomial decay model [63] model we assume that $\lambda_i^2 = \mathcal{O}(i^{-\alpha})$, where $\alpha > 1$. Therefore for Linear Kernels $\lambda_i^2 = \lambda_1^2 i^{-\alpha}$. Therefore conducting a similar computation to Equation 20-Equation 22,

Computing the sum of square of eigen values of any $X^T X$,

$$\begin{aligned}
\sum_{i=1}^n (\lambda_i)^2 &= \lambda_1^2 + \lambda_2^2 + \cdots + \lambda_n^2 \\
\sum_{i=1}^n (\lambda_i)^2 &= \lambda_1^2 + \lambda_1^2 2^{-\alpha} + \cdots + \lambda_1^2 n^{-\alpha} \\
\sum_{i=1}^n (\lambda_i)^2 &\leq \lambda_1^2 (2 + \frac{2^{1-\alpha}}{\alpha - 1}) \text{ , from Theoreom A.4 [63]} \\
\sum_{i=1}^n (\lambda_i)^2 &\leq 3\lambda_1^2 \text{ , for } \alpha \gg 1
\end{aligned} \tag{23}$$

Using Equation 23 and computing the fraction of square of first kernel eigenvalue to the sum of squares as in Equation 21 -

$$\begin{aligned}
\frac{\lambda_1^2}{\sum_{i=1}^n (\lambda_i)^2} &\geq \frac{\lambda_1^2}{3\lambda_1^2} \\
\frac{\lambda_1^2}{\sum_{i=1}^n (\lambda_i)^2} &\approx \frac{1}{3} \\
\frac{\lambda_1}{\sqrt{\sum_{i=1}^n (\lambda_i)^2}} &\approx \frac{1}{\sqrt{3}}
\end{aligned} \tag{24}$$

Analogously to Equation 22, substituting from Equation 24 for a polynomial decay of eigen values into the summation for $i = 1$ and $j = 1$ in Equation 18,

$$\frac{\lambda_X^1 \lambda_Y^1 \langle v_X^1, v_Y^1 \rangle^2}{\sqrt{\sum_{i=1}^{r_1} \left(\lambda_X^i\right)^2}\sqrt{\sum_{j=1}^{r_2} \left(\lambda_Y^j\right)^2}} \approx \frac{1}{3}\langle v_X^1, v_Y^1 \rangle^2 \tag{25}$$

Which reveals Linear-CKA assigns a higher weight to the cosine similarity between the top right singular values of activation matrices, thereby demonstrating why Linear-CKA is insensitive to changes in most but the top singular vectors [11], [12].

## F.2. Analysis of LRSC-CKA

Continuing the analysis further for LRSC-CKA having given the same (assuming centred) $X = U_X \Sigma_X V_X^T$ and $Y = U_Y \Sigma_Y V_Y^T$ as in Section F.1. We first compute their respective LRSC Affinity matrices $C_X = V_X V_X^T$ and $C_Y = V_Y V_Y^T$ by Equation 1, where $V_X$ and $V_Y$ are rank-r (assumed same for simplicity) truncated right singular vectors of $X$ and $Y$ respectively. Essentially when comparing LRSC-CKA with Linear-CKA we observe that LRSC Affinity is a Linear Kernel with all singular values below a cut-off threshold (rank-r, for simplicity) set to 0 and all singular values above this threshold clamped to 1. Then, the corresponding LRSC-CKA based on Equation 18 is given by Equation 26.

$$CKA_{LRSC}(\mathbf{C_X}, \mathbf{C_Y}) = \sum_{i=1}^{r} \sum_{j=1}^{r} \frac{1}{r} \langle v_X^i, v_Y^j \rangle^2 \qquad (26)$$

Given that real data matrices are largely low rank [61] with $r = \mathcal{O}(\log(n+d))$, we can see that when compared to Linear-CKA, Equation 25, the cosine similarity of larger right singular vectors of $X$ and $Y$ contribute a smaller fraction to the LRSC-CKA output. This analysis offers an additional view into the experimental findings of Section 5 regarding why LRSC-CKA is more sensitive throughout the span of its singular vectors and why Linear-CKA is mostly sensitive to a few top singular vectors.

### F.3. Big Data Matrices are Low Rank

Here we state Theorem 2.6 from [61], Big Data Matrices are Low Rank, for the sake of completeness. We begin by stating the Johnson-Lindenstrauss Lemma (JL-Lemma) and its variants [64].

**Lemma F.1 (Johnson-Lindenstrauss Lemma [61])** *Let* $0 < \epsilon_{JL} < 1$, *Given* $N$ *data samples* $\mathbf{x}_1, \ldots, \mathbf{x}_N \in \mathbb{R}^d$ *and* $r = 8(\log n)/\epsilon_{JL}^2$. *Then,* $\exists\, Q : \mathbb{R}^d \to \mathbb{R}^r$ *such that*

$$(1 - \epsilon_{JL})\|\mathbf{x}_i - \mathbf{x}_j\|^2 \le \|\mathbf{Q}\mathbf{x}_i - \mathbf{Q}\mathbf{x}_j\|^2 \le (1 + \epsilon_{JL})\|\mathbf{x}_i - \mathbf{x}_j\|^2, 1 \le i, j \le N, w.h.p \qquad (27)$$

[61] then proposes a variant of the JL-Lemma based on the differences of inner products, states as follows.

**Lemma F.2 (Variant of the JL-Lemma [61])** *Let* $0 < \epsilon_{JL} < 1$, *Given* $N$ *data samples* $\mathbf{x}_1, \ldots, \mathbf{x}_N \in \mathbb{R}^d$ *and* $r = 8(\log n)/\epsilon_{JL}^2$. *Then,* $\exists\, Q : \mathbb{R}^d \to \mathbb{R}^r$ *such that*

$$|\mathbf{x}_i^T \mathbf{x}_j - \mathbf{x}_i^T \mathbf{Q}^T \mathbf{Q}\mathbf{x}_j| \le \epsilon_{JL}(\|\mathbf{x}_i\|^2 + \|\mathbf{x}_j\|^2 - \mathbf{x}_i^T \mathbf{x}_j), 1 \le i, j \le N, w.h.p \qquad (28)$$

**Theorem F.3 (Big Data Matrices are Low Rank [61])** *Let* $\mathbf{X} \in \mathbb{R}^{m \times n}$ *with* $m \ge n$ *and* $0 < \epsilon < 1$. *Then, with* $r = \lceil 72 \log(2n + 1)/\epsilon^2 \rceil$ *we have*

$$\inf_{rank(Y) \le r} \|X - Y\|_{max} \le \epsilon \|X\|_2 \qquad (29)$$

*where* $\|\cdot\|_{max}$ *is the maximum absolute entry norm and* $\|\cdot\|_2$ *is the spectral norm.*

# G. Additional results on correlation between layer wise linear probe performance with LRSC and Linear-CKA coefficients

In this section we provide an additional and a more complete set of results for the material presented in Figure 1 and Table 2 of Section 6.1.

## G.1. Correlation of layerwise LRSC Coefficients with Linear Probe accuracy for CIFAR10

In this section we demonstrate layerwise dynamics observed for the correlation between Subspace Coefficient based classification and linear probes as shown in row 2 of Table 2 for CIFAR10.

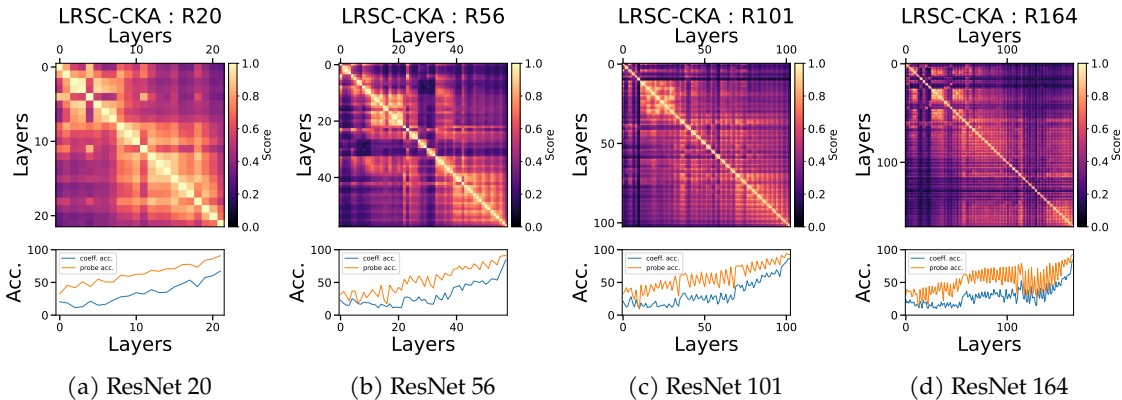

(a) ResNet 20        (b) ResNet 56        (c) ResNet 101        (d) ResNet 164

Figure 14: Comparison between layer-wise linear probe accuracy vs LRSC Coefficient based classification accuracy on various networks trained on CIFAR 10.

## G.2. Correlation of layer wise Linear-CKA Coefficients with Linear Probe accuracy for CIFAR10

In this section we demonstrate layerwise dynamics observed for the correlation between Linear-CKA Coefficient based classification and linear probes as shown in row 3 of Table 2 for CIFAR10.

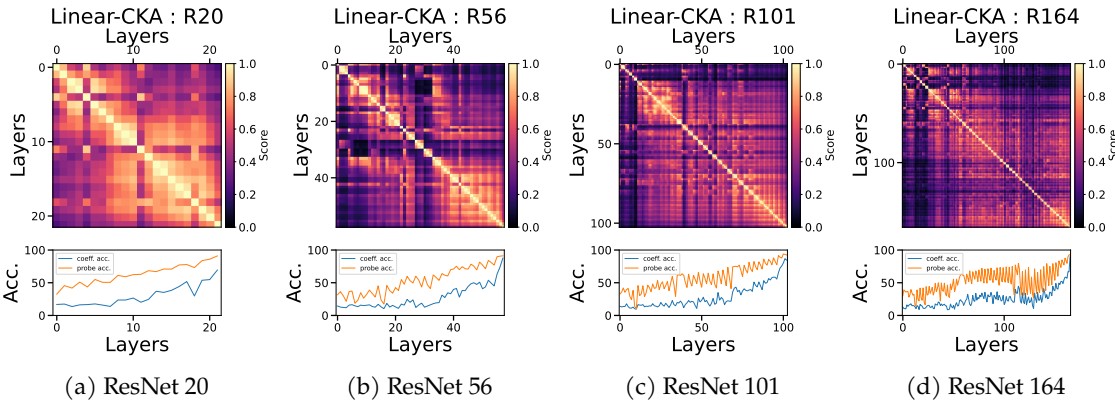

(a) ResNet 20        (b) ResNet 56        (c) ResNet 101        (d) ResNet 164

Figure 15: Comparison between layer-wise linear probe accuracy vs Linear-CKA Coefficient based classification accuracy on various networks trained on CIFAR 10

## G.3. Correlation of layerwise LRSC Coefficients with Linear Probe accuracy for CIFAR100

We demonstrate layerwise dynamics observed for the correlation between Subspace Coefficient based classification and linear probes as shown in row 2 of Table 2 for CIFAR100.

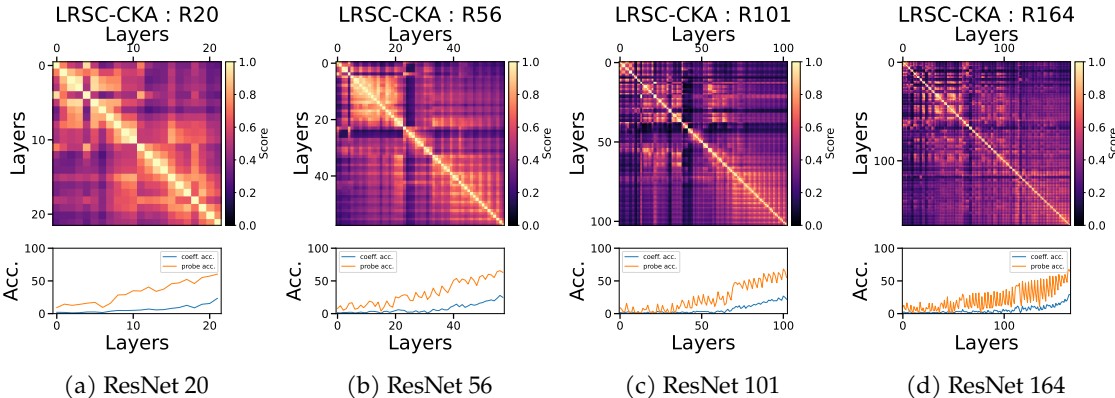

(a) ResNet 20          (b) ResNet 56          (c) ResNet 101          (d) ResNet 164

Figure 16: Comparison between layer-wise linear probe accuracy vs LRSC Coefficient based classification accuracy on various networks trained on CIFAR 100

## G.4. Correlation of layer wise Linear-CKA Coefficients with Linear Probe accuracy for CIFAR100

We demonstrate layerwise dynamics observed for the correlation between Linear-CKA Coefficient based classification and linear probes as shown in row 3 of Table 2 for CIFAR100.

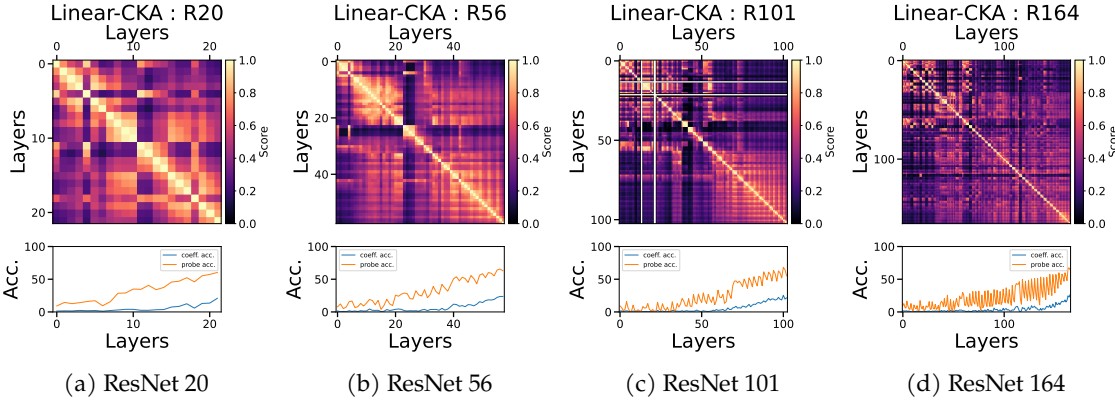

(a) ResNet 20          (b) ResNet 56          (c) ResNet 101          (d) ResNet 164

Figure 17: Comparison between layer-wise linear probe accuracy vs Linear-CKA Coefficient based classification accuracy on various networks trained on CIFAR 100

# H. Additional Results comparing the effects of memorisation and generalization on neural networks

In this section we detail additional experimental results comparing the effects of memorisation and generalisation in support of the results shown in Section 6.2. First, in Table 7 and Table 8 we demonstrate the performance of various networks used for these tasks. Subsequently from Section I.2 - Section I.6 we demonstrate the differences between networks trained in normal clean-label training regimes and network trained in noisy label regimes by using LRSC-CKA and Linear-CKA on CIFAR10 and CIFAR100.

Table 7: Performance of ReLU Networks of various depths used in these experiments on the probing set of CIFAR10.

| Performance (%) over normal and noisy training regimes | | | | | | | |
|---|---|---|---|---|---|---|---|
| Regime→ | Normal CIFAR 10 | | | | Noisy CIFAR 10 | | | |
| Metric ↓ | R20 | R56 | R101 | R164 | R20n | R56n | R101n | R164n |
| **Accuracy %** | 91.2 | 90.2 | 94.39 | 93.2 | 65.4 | 61.4 | 50.8 | 54 |

Table 8: Performance of ReLU Networks of various depths used in these experiments on the probing set of CIFAR100.

| Performance (%) over normal and noisy training regimes | | | | | | | |
|---|---|---|---|---|---|---|---|
| Regime→ | Normal CIFAR 100 | | | | Noisy CIFAR 100 | | | |
| Metric ↓ | R20 | R56 | R101 | R164 | R20n | R56n | R101n | R164n |
| **Accuracy %** | 66.6 | 71.3 | 71.3 | 74 | 40.3 | 29.6 | 26.5 | 27.8 |

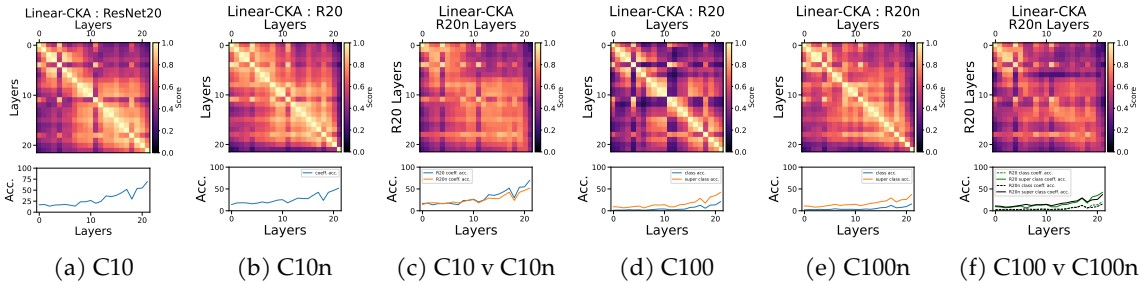

| (a) C10 | (b) C10n | (c) C10 v C10n | (d) C100 | (e) C100n | (f) C100 v C100n |

Figure 18: Analogous to Figure 2. Linear CKA based comparison of ResNet20 when trained on CIFAR10 and CIFAR100 vs their Corrupted Versions.

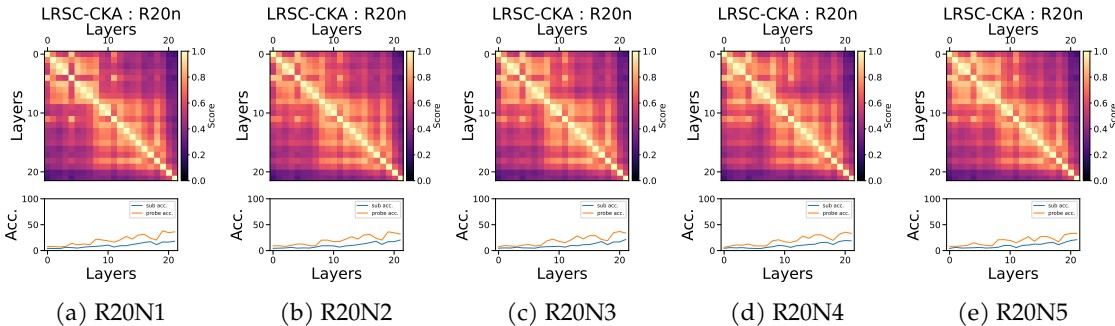

| (a) R20N1 | (b) R20N2 | (c) R20N3 | (d) R20N4 | (e) R20N5 |

Figure 19: Analogous to Figure 3, Correlation study between layer-wise LRSC subspace reconstruction accuracy vs linear probe based classification accuracy on various networks trained on Noisy CIFAR 100.

## H.1. Additional corroborative analysis with LRSC-CKA on effects Memorisation on ResNets trained on CIFAR10

In Figure 20 we document the behaviour of normally and noisily trained ResNets of various depths, denoted by R20, R56, R101 and R164.

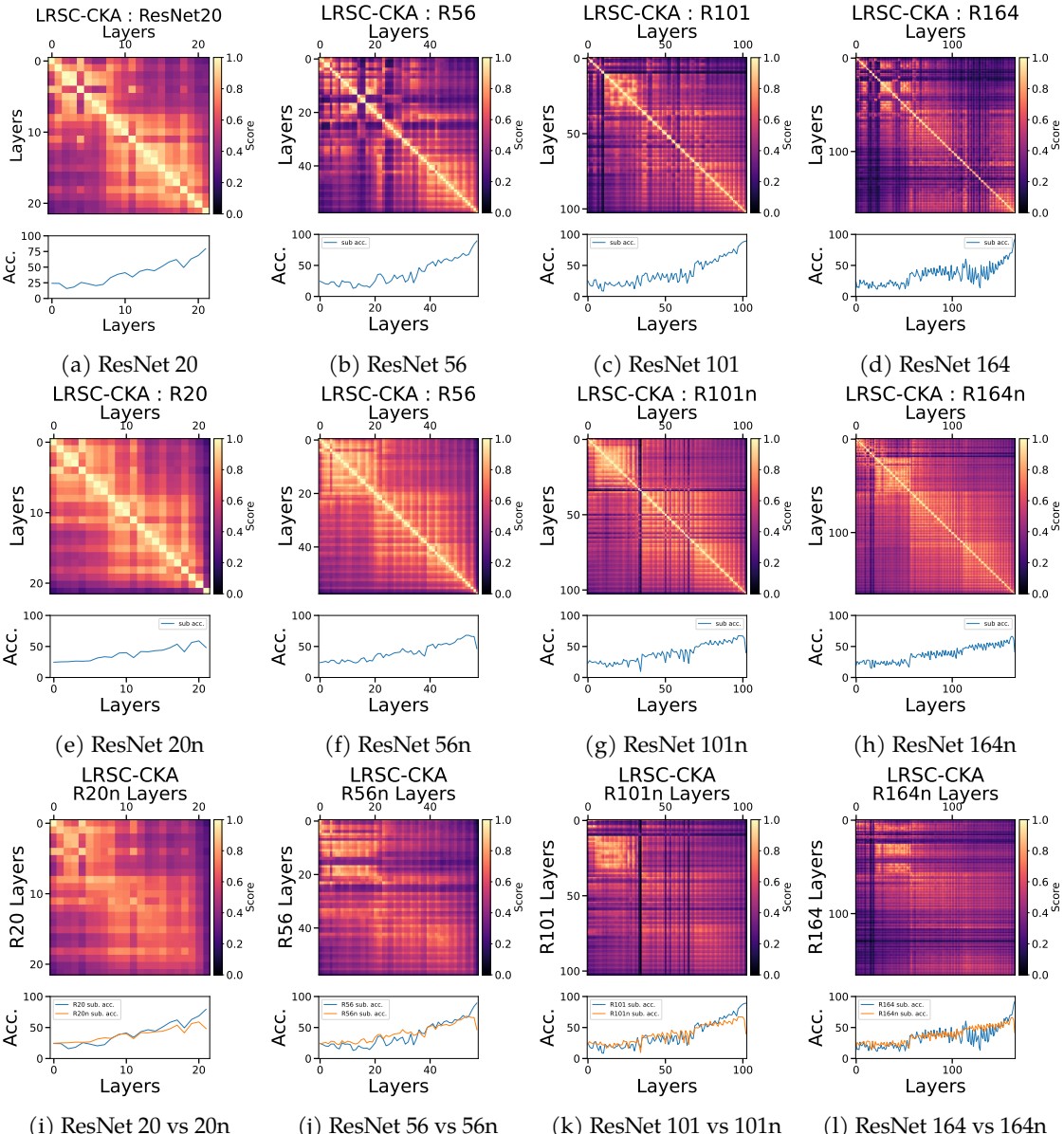

Figure 20: Analysis of ResNets trained on CIFAR10 and Noisy CIFAR10 with LRSC-CKA. The top row of the figure various architectures trained on clean labels as part of a normal training setup. The second row contains corresponding ResNets training on data with 50% of the labels being assigned uniformly at random. The last row is a comparison between normal and noisily trained network of a given depth. Therefore as a consequence, each column of this figure represents a normally trained network, a noisily trained network of the same depth and a comparison between the two.

In Figure 21 we add the LRSC Subspace Coefficient based analysis as described in Section A.1 and [33] just for completeness, though it is not central to our arguments.

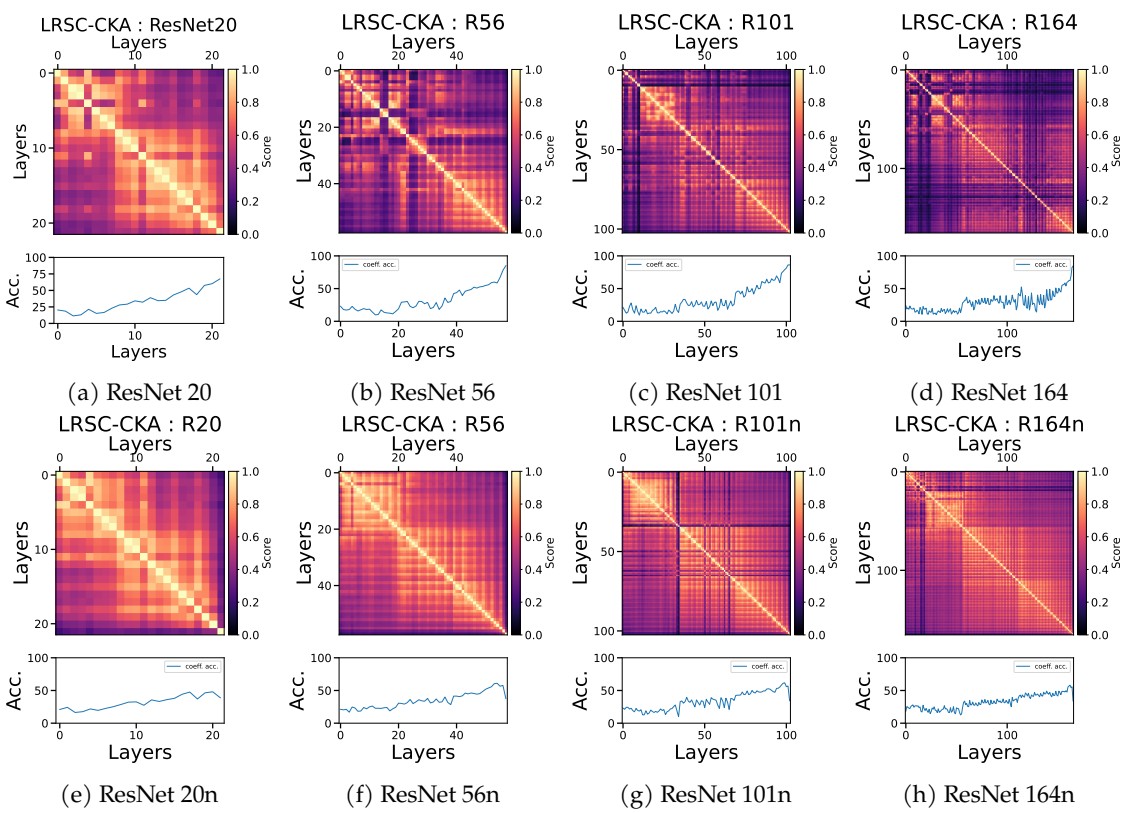

Figure 21: Coefficient based analysis of ResNets trained on CIFAR10 and Noisy CIFAR10 with LRSC

## H.2. Additional corroborative analysis with LRSC-CKA on effects Memorisation on ResNets trained on CIFAR100

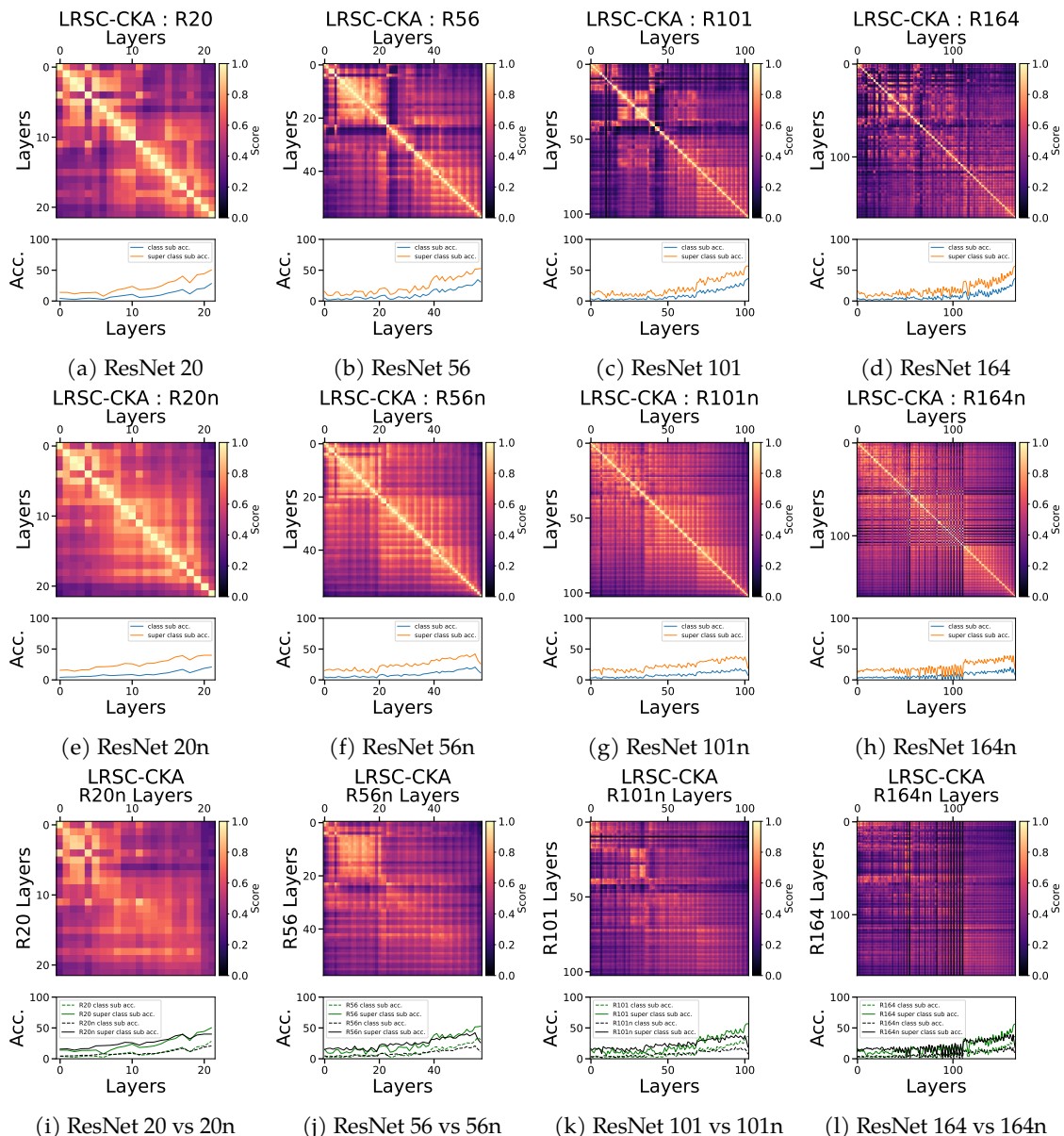

Figure 22: Analysis of ResNets trained on CIFAR100 and Noisy CIFAR100 with LRSC. As before, the top row of the figure shows various architectures trained on clean labels. The second row contains corresponding ResNets training on noisy labels. The last row is a comparison between normal and noisily trained network of a given depth.

## H.3. Corresponding corroborative analysis with Linear-CKA on effects Memorisation on ResNets trained on CIFAR10

In Figure 24 of this section we demonstrate the Linear-CKA analogue of Section H.1.

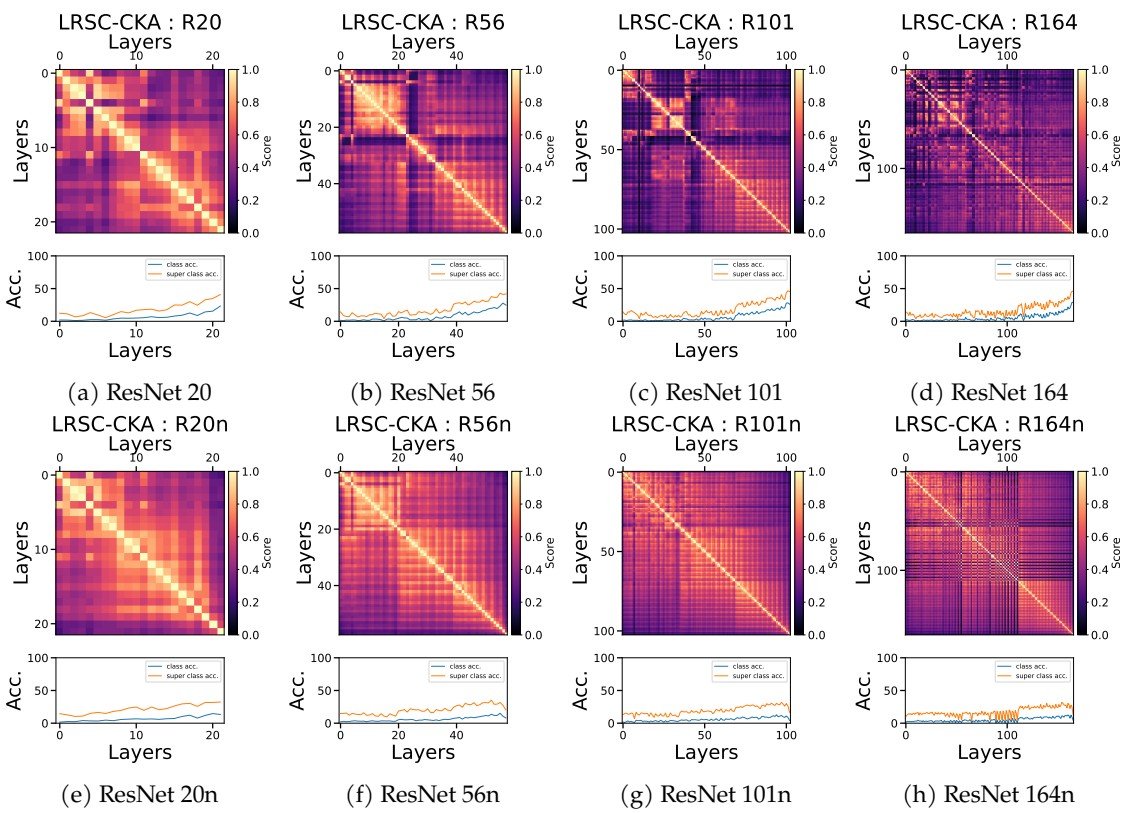

Figure 23: Coefficient based analysis of ResNets trained on CIFAR100 and Noisy CIFAR100 with LRSC

## H.4. Corresponding corroborative analysis with Linear-CKA on effects Memorisation on ResNets trained on CIFAR100

In this section we present the Linear-CKA analysis on CIFAR100 dataset corresponding to Figure 22.

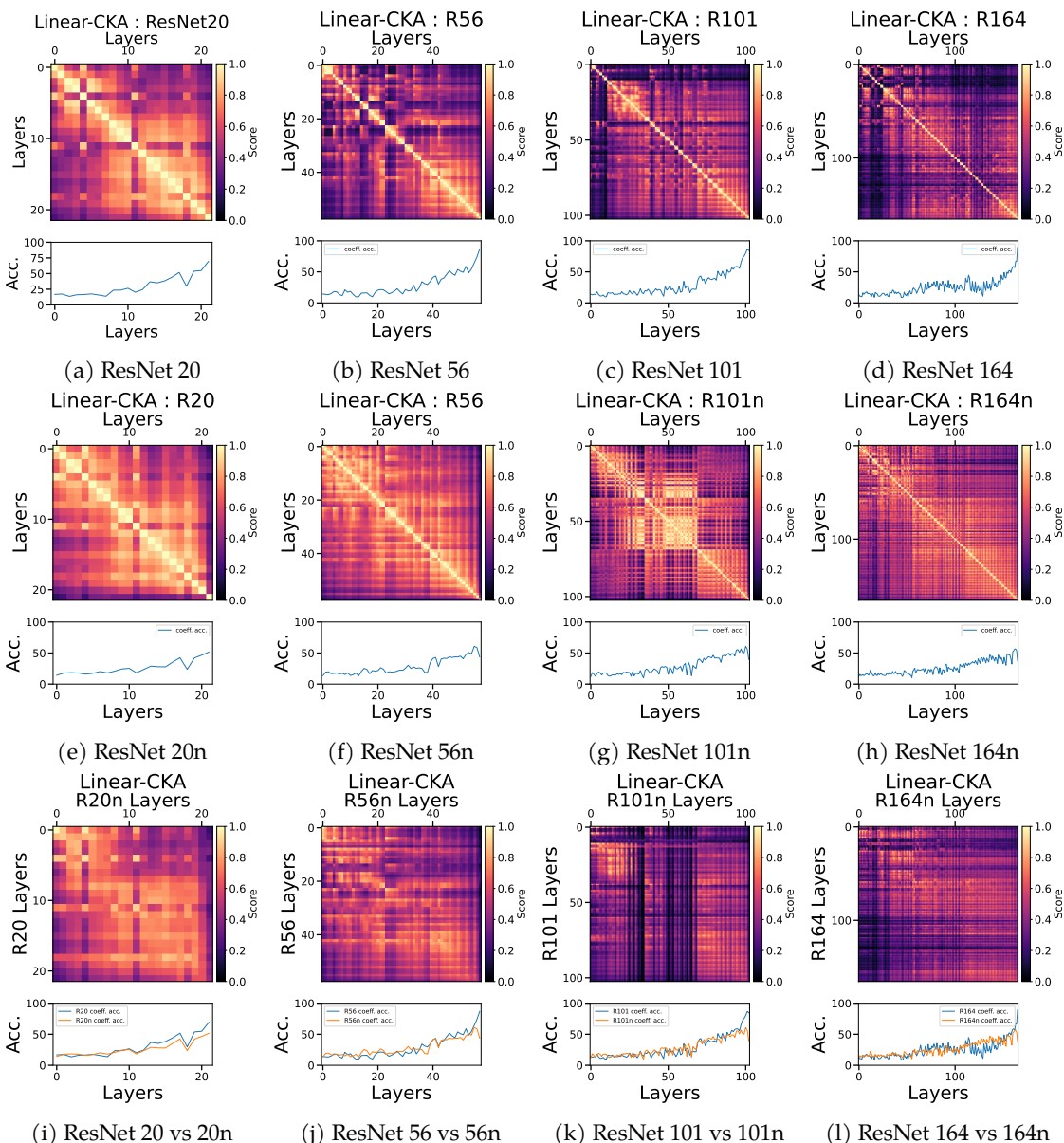

Figure 24: Corresponding to Figure 20 here we present an analysis of ResNets trained on CIFAR10 and Noisy CIFAR10 with Linear-CKA. As stated earlier, the top row of the figure has various architectures trained on clean labels. The second row contains corresponding ResNets training on noisy labels. The last row is a comparison between normal and noisily trained network of a given depth.

## H.5. Topologically corresponding Layer to Layer comparison

In this section we present additional results accompanying Figure 7. In Figure 26 we show the Linear-CKA layer to layer (diagonal) analysis on CIFAR10. In Figure 27 and Figure 28 we show the corresponding LRSC-CKA and Linear-CKA analysis on CIFAR100, respectively.

## H.6. LRSC Variance Threshold Sensitivity

As an extension to experiments of Section B.1 we show LRSC-CKA over varying values of variance thresholds to learn the LRSC Kernel. Here in Figure 29 via the means of Subspace Reconstruction based accuracy in Section 4.2 we demonstrate that the comparison of networks via LRSC is fairly

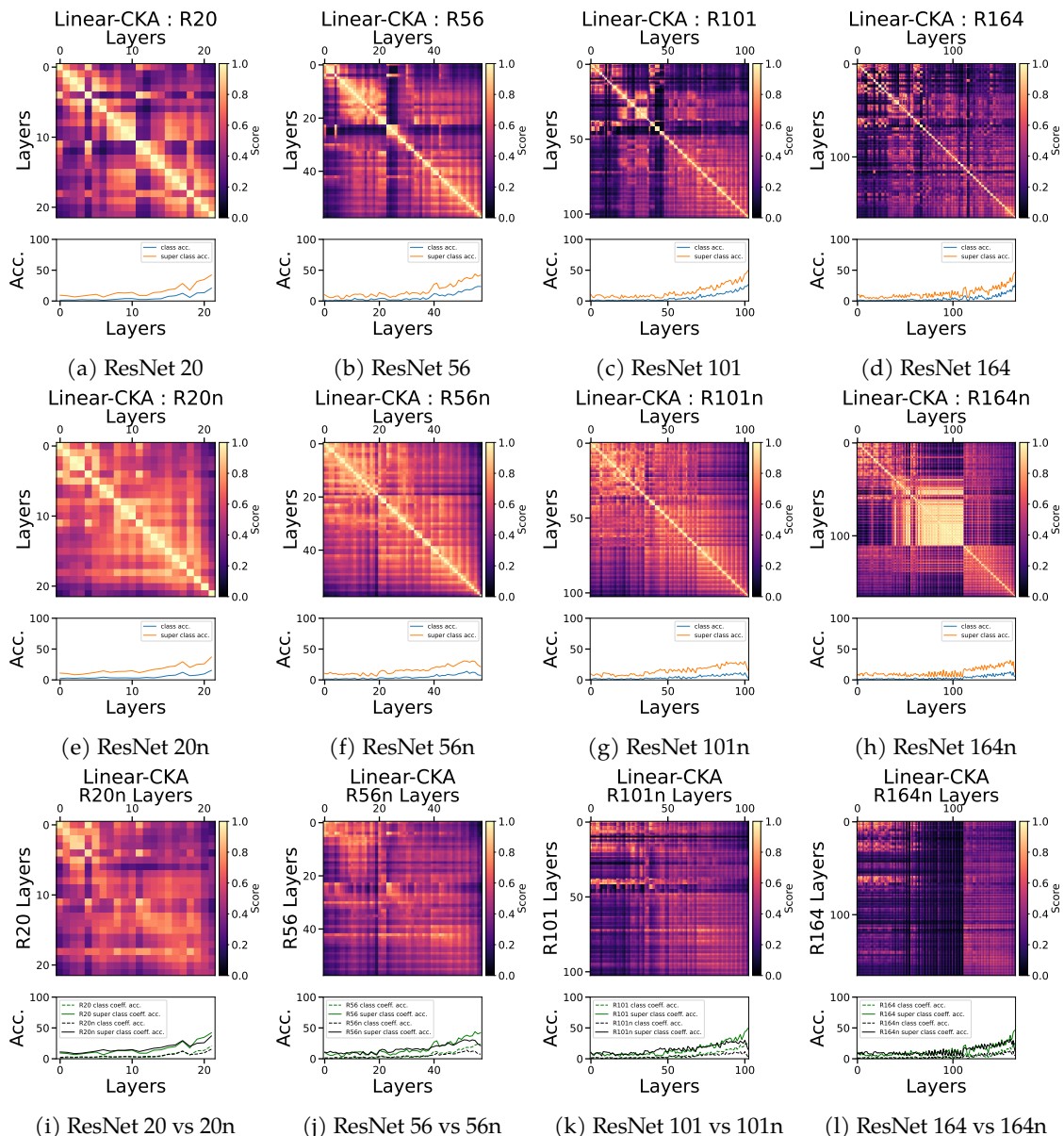

Figure 25: Counterpart to Figure 22 here we show the analysis of ResNets trained on CIFAR100 and Noisy CIFAR100 with Linear-CKA. The top row of the figure shows various architectures trained on clean labels. The second row contains corresponding ResNets training on noisy labels. The last row is a comparison between normal and noisily trained network of a given depth.

insensitive to the value of variance thresholding used to learn the LRSC Kernel. The upper half of Figure 29a shows the subspace reconstruction accuracy over varying thresholds from 50% - 90% variance explained for networks trained on clean CIFAR10. The bottom half of Figure 29a does the same for networks trained on noisy CIFAR10. In Figure 29a we observe the differences between generalising and memorising networks as observed in Section 6.2 for all values of variance thresholds. A similar analysis is shown for CIFAR100 in Figure 29b.

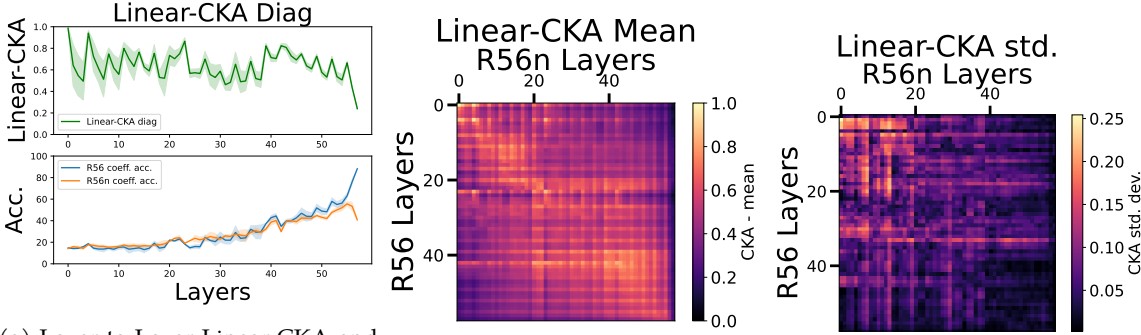

(a) Layer to Layer Linear-CKA and Subspace Reconstruction Accuracy  (b) Linear-CKA R56 vs R56n - mean  (c) Linear-CKA R56 vs R56n - std.

Figure 26: Linear CKA Analogue to the analysis of Figure 7

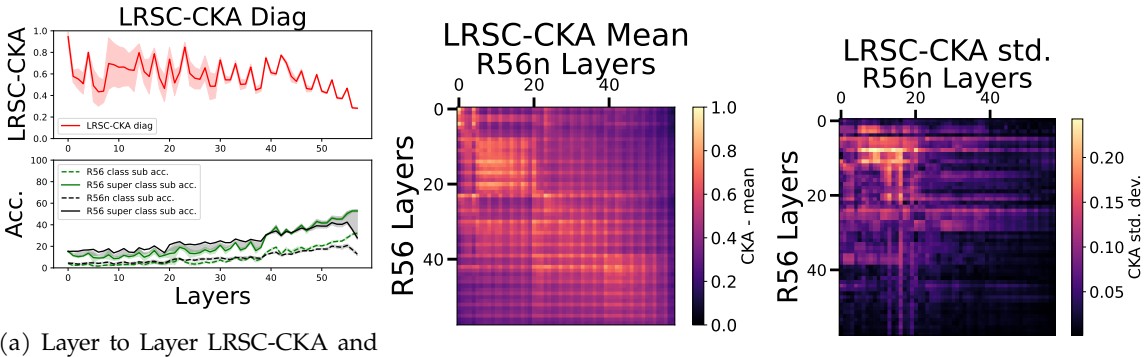

(a) Layer to Layer LRSC-CKA and Subspace Reconstruction Accuracy  (b) LRSC-CKA R56 vs R56n - mean  (c) LRSC-CKA R56 vs R56n - std.

Figure 27: CIFAR100 counterpart to LRSC-CKA Layer to Layer analysis of Figure 7

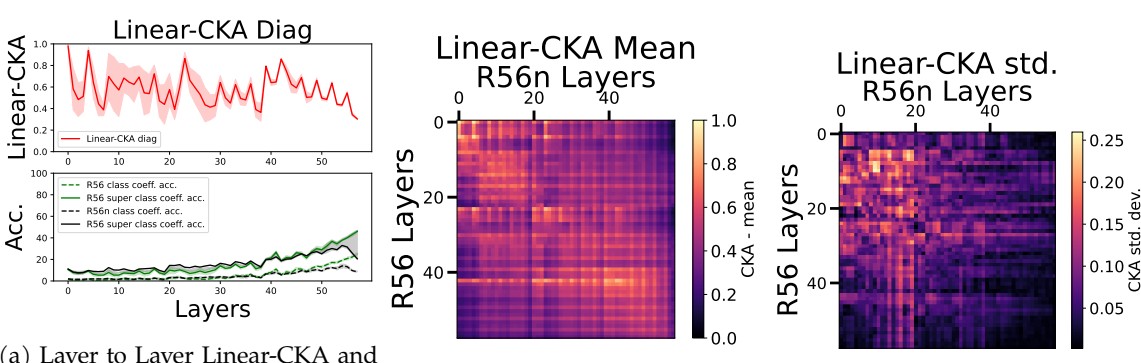

(a) Layer to Layer Linear-CKA and Subspace Reconstruction Accuracy  (b) Linear-CKA R56 vs R56n - mean  (c) Linear-CKA R56 vs R56n - std.

Figure 28: Linear CKA Analogue to the analysis of Figure 27

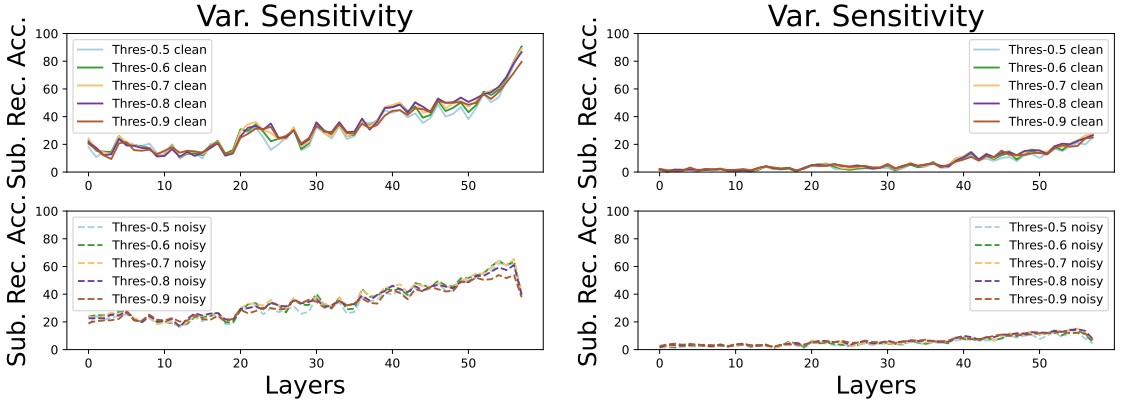

(a) LRSC Variance parameter characteristics CIFAR10 (b) LRSC Variance parameter characteristics CIFAR10

Figure 29: LRSC Variance parameter sensitivity analysis

# I. Additional results on correlation between layer wise linear probe performance with LRSC and Linear-CKA coefficients for networks with memorisation

In this section we lay out detailed LRSC-CKA results for the experiments conducted in Table 3 of Section 6.2 to demonstrate the correlation of layerwise subspace reconstruction based accuracy with layerwise linear probe accuracy for networks that memorise. In each subsection we demonstrate LRSC-CKA outputs used to compute the correlations for each row of Table 3. Section I.1 demonstrates the layerwise subspace reconstruction and linear probe accuracies of 5 ResNets labeled N1-N5 which were trained to memorise CIFAR10. Section I.2 demonstrates the corresponding correlations between subspace coefficient based accuracy as defined in Section A.1 and [33]. Finally Section I.3 shows the correlations between Linear-CKA coefficient and linear probe accuracy on CIFAR10. Section I.4 - Section I.6 host the corresponding results for CIFAR100.

## I.1. Correlation of layer wise LRSC subspace reconstruction accuracy with Linear Probe accuracy for Noisy CIFAR10

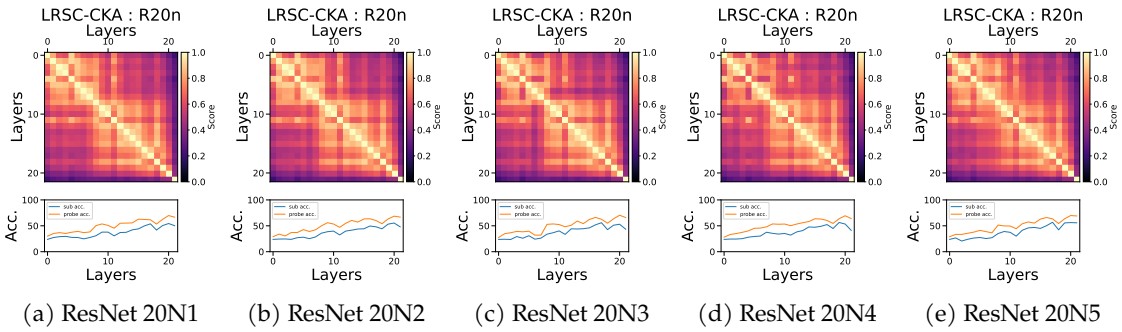

(a) ResNet 20N1  (b) ResNet 20N2  (c) ResNet 20N3  (d) ResNet 20N4  (e) ResNet 20N5

Figure 30: Correlational comparison between layer-wise LRSC subspace reconstruction accuracy vs linear probe based classification accuracy on various networks trained on Noisy CIFAR 10

## I.2. Correlation of layer wise LRSC coefficient based accuracy with Linear Probe accuracy for Noisy CIFAR10

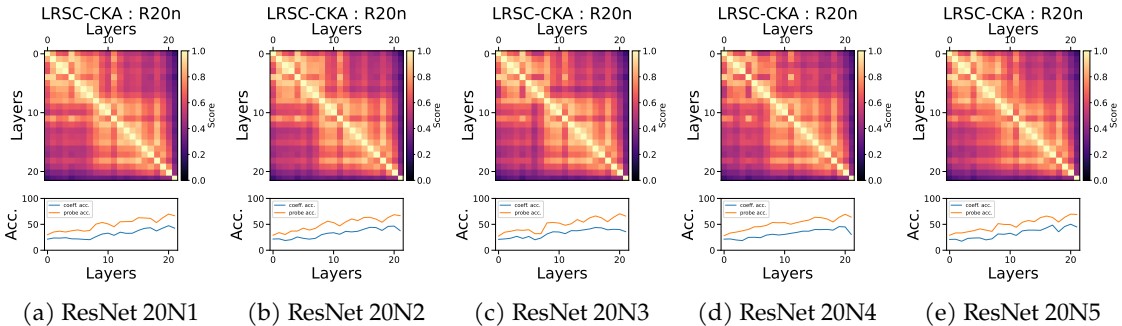

(a) ResNet 20N1    (b) ResNet 20N2    (c) ResNet 20N3    (d) ResNet 20N4    (e) ResNet 20N5

Figure 31: Correlational comparison between layer-wise LRSC coefficient based accuracy vs linear probe based classification accuracy on various networks trained on Noisy CIFAR 10

## I.3.  Correlation of layer wise Linear CKA coefficient based accuracy with Linear Probe accuracy for Noisy CIFAR10

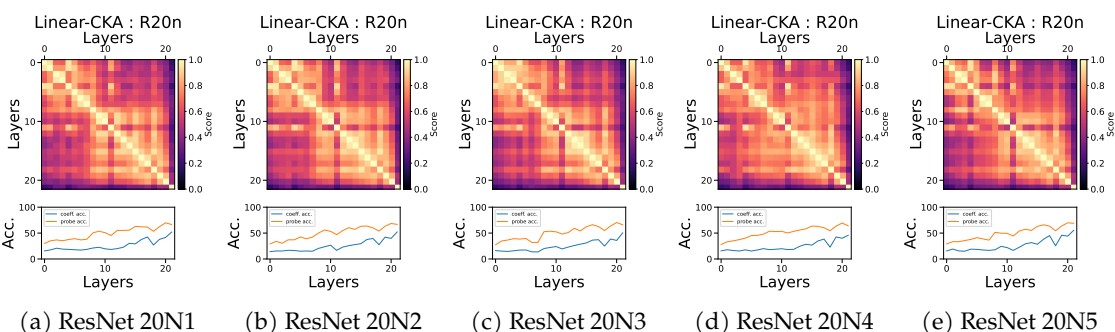

(a) ResNet 20N1    (b) ResNet 20N2    (c) ResNet 20N3    (d) ResNet 20N4    (e) ResNet 20N5

Figure 32: Correlational comparison between layer-wise Linear-CKA coefficient based accuracy vs linear probe based classification accuracy on various networks trained on Noisy CIFAR 10

## I.4.  Correlation of layer wise LRSC subspace reconstruction accuracy with Linear Probe accuracy for Noisy CIFAR100

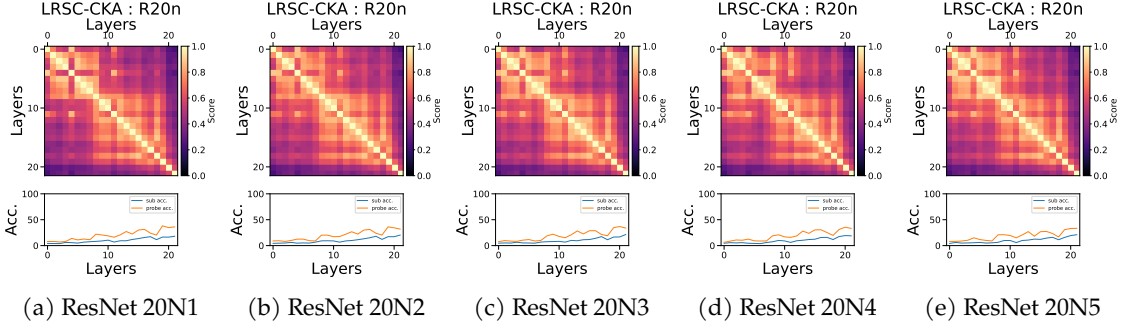

(a) ResNet 20N1    (b) ResNet 20N2    (c) ResNet 20N3    (d) ResNet 20N4    (e) ResNet 20N5

Figure 33: Correlational comparison between layer-wise LRSC subspace reconstruction accuracy vs linear probe based classification accuracy on various networks trained on Noisy CIFAR 100

## I.5. Correlation of layer wise LRSC coefficient based accuracy with Linear Probe accuracy for Noisy CIFAR100

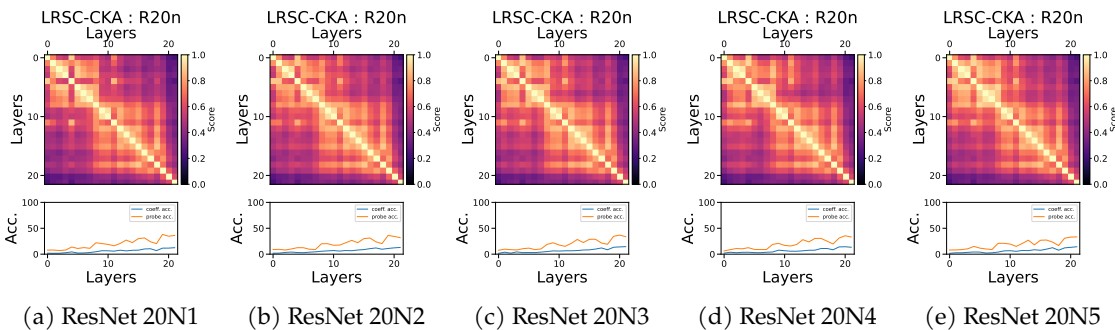

(a) ResNet 20N1    (b) ResNet 20N2    (c) ResNet 20N3    (d) ResNet 20N4    (e) ResNet 20N5

Figure 34: Correlational comparison between layer-wise LRSC coefficient based accuracy vs linear probe based classification accuracy on various networks trained on Noisy CIFAR 100

## I.6. Correlation of layer wise Linear CKA coefficient based accuracy with Linear Probe accuracy for Noisy CIFAR100

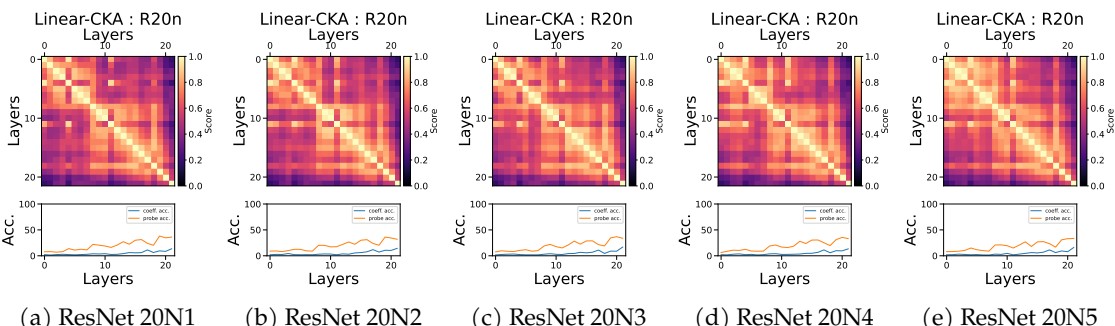

(a) ResNet 20N1    (b) ResNet 20N2    (c) ResNet 20N3    (d) ResNet 20N4    (e) ResNet 20N5

Figure 35: Correlational comparison between layer-wise Linear-CKA coefficient based accuracy vs linear probe based classification accuracy on various networks trained on Noisy CIFAR 100

# J. Additional results for analysis conducted on Mini ImageNet 100

In this section we show additional supporting results for the analysis conducted in Section C.1. We begin so by demonstrating the performance of networks used perform the analysis of Table 4 in Table 9.

Table 9: Performance of ResNet 20s used in these experiments on the probing set.

| Performance (%) over normal and noisy training regimes | | | | | | | | | |
|---|---|---|---|---|---|---|---|---|---|
| Regime→ | Mini ImageNet 100 | | | | | Noisy Mini ImageNet 100 | | | | |
| Metric ↓ | V1 | V2 | V3 | V4 | V5 | N1 | N2 | N3 | N4 | N5 |
| **Accuracy %** | 57.5 | 58.2 | 58.1 | 58.3 | 59.1 | 29.5 | 29.3 | 29.9 | 29.5 | 30.7 |

Next in Table 10 we also provide the details for networks used to generate additional results supplementing the analysis of Figure 8, comparing Clean and Noisly Trained ResNet 56s via LRSC-CKA. The corresponding Linear-CKA analysis is shown in Section J.1.

Table 10: Performance of ResNet 56s used in these experiments on the probing set.

| Performance (%) over normal and noisy training regimes | | | | | | | | | |
|---|---|---|---|---|---|---|---|---|---|
| Regime→ | Mini ImageNet 100 | | | | | Noisy Mini ImageNet 100 | | | | |
| Metric ↓ | V1 | V2 | V3 | V4 | V5 | N1 | N2 | N3 | N4 | N5 |
| **Accuracy %** | 64.07 | 63.7 | 65.5 | 62.8 | 61.7 | 22.05 | 22.5 | 21.3 | 22.03 | 21.6 |

## J.1. Linear-CKA Analysis of the ResNet 56 on Mini ImageNet 100

Next we demonstrate the Linear-CKA counterpart analysis of Figure 8 from Section C.1. As stated earlier in Section C.1 we trained ResNet 56s on clean and noisy versions of Mini ImageNet 100 with the cleanly trained ResNet 56 achieving an accuracy of 64.07% on the probing set and the noisily trained ResNet 56 achieving an accuracy of only 22.05%.

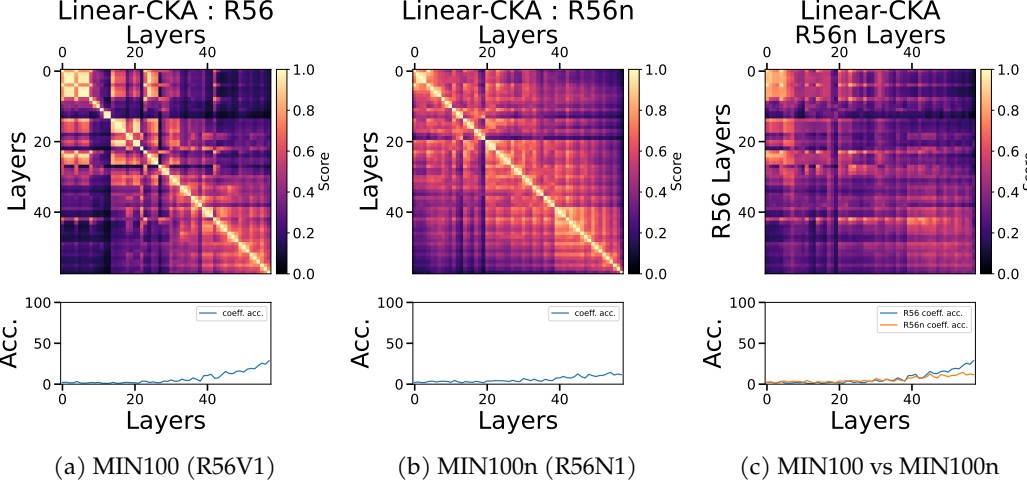

(a) MIN100 (R56V1)  (b) MIN100n (R56N1)  (c) MIN100 vs MIN100n

Figure 36: Linear-CKA analogue of LRSC-CKA results in Figure 8.

## J.2. Aggregated network pairwise and one to one layer comparison of clean and noisy ResNet 56s on Mini Image Net 100

In this section, similar to Figure 7 in Section 6.2 we show the aggregated Pairwise and One to One Analysis of network layers analyzing clean and noisily trained ResNet 56s (5 for each setting) on Mini Image Net 100 using LRSC-CKA in Figure 37 and Linear-CKA inFigure 38.

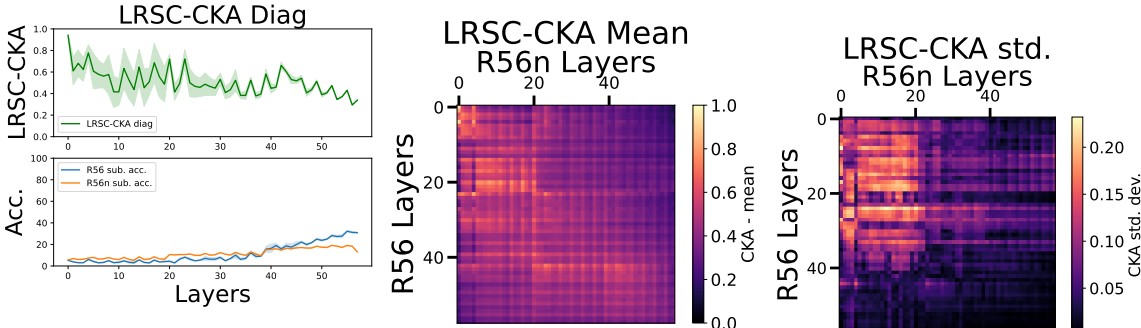

(a) Layer to Layer LRSC-CKA and Subspace Reconstruction Accuracy   (b) LRSC-CKA R56 vs R56n - mean   (c) LRSC-CKA R56 vs R56n - std.

Figure 37: Similar to Figure 7, All analysis presented in this figure are done over 5 pairs of generalising and memorising ResNet 56s trained on clean and noisy Mini Image Net 100 respectively. Figure 37a shows the Layer to Layer comparison between the two networks. Corresponding complete layerwise LRSC-CKA analysis over these 5 pairs is shown in Figure 37b and Figure 37c, mean and standard deviation, respectively. As shown in Figure 37a the final layers of clean and noisy network start to deviate significantly in their performance.

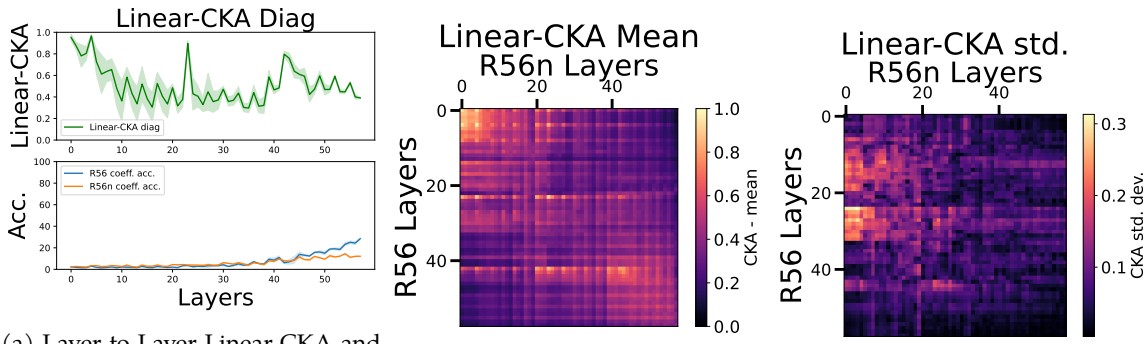

(a) Layer to Layer Linear-CKA and Subspace Reconstruction Accuracy   (b) Linear-CKA R56 vs R56n - mean   (c) Linear-CKA R56 vs R56n - std.

Figure 38: Linear-CKA Counter Part to Figure 37.

## J.3. Additional LRSC-CKA Analysis of the ResNet 56s on Mini ImageNet 100

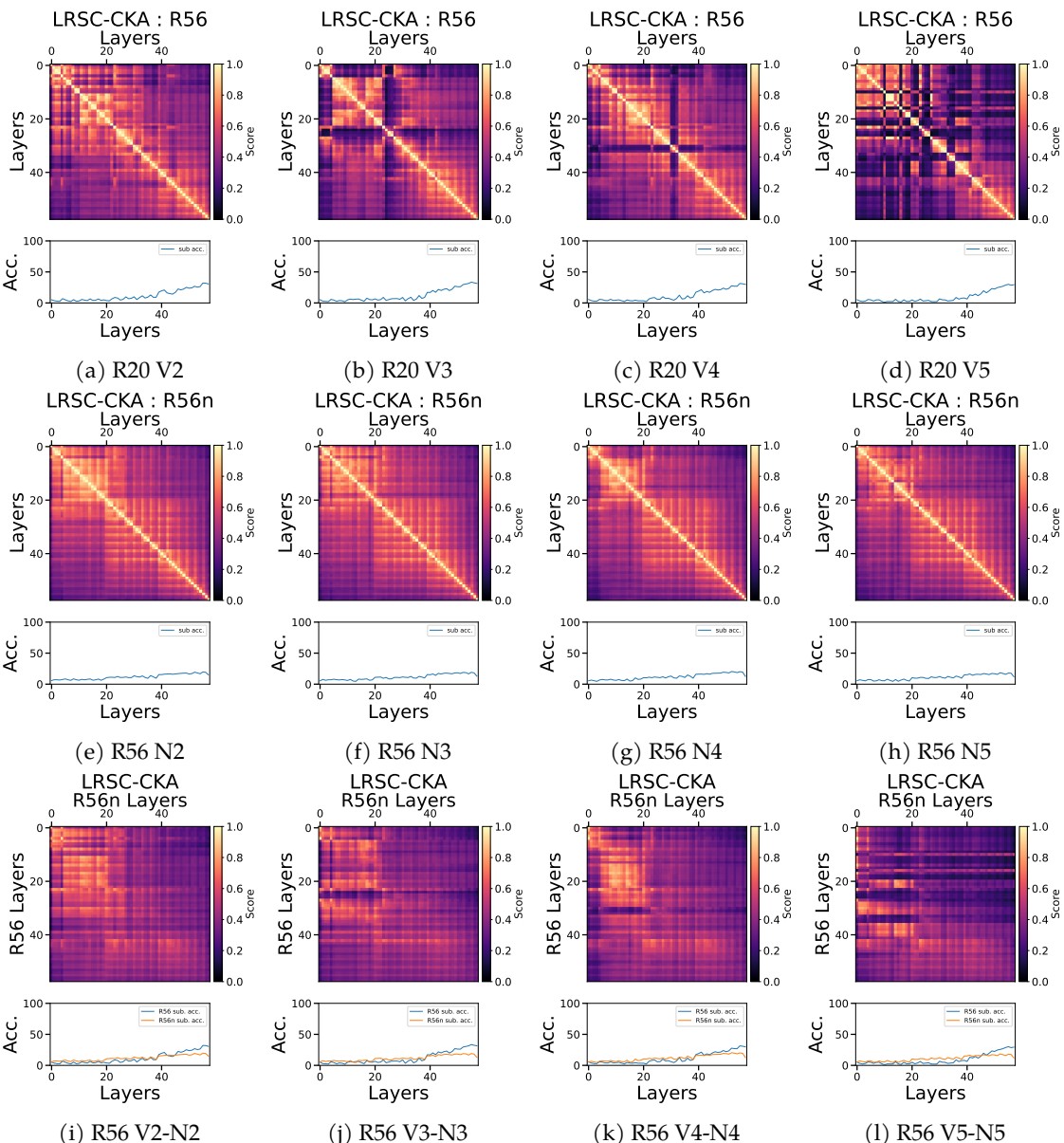

Figure 39: LRSC-CKA Analysis on Mini Image Net comparing normally and noisily trained networks. First row consists of 4 normally trained ResNet 56s, the second row consists of 5 noisily trained ResNet 56s. The third row is the comparison between 4 normally and noisily trained ResNets. Therefore, each column in the figure demonstrates a normally trained ReLU network, a noisily trained ReLU network and their comparison.

## J.4. Additional Linear-CKA Analysis of the ResNet 56s on Mini ImageNet 100

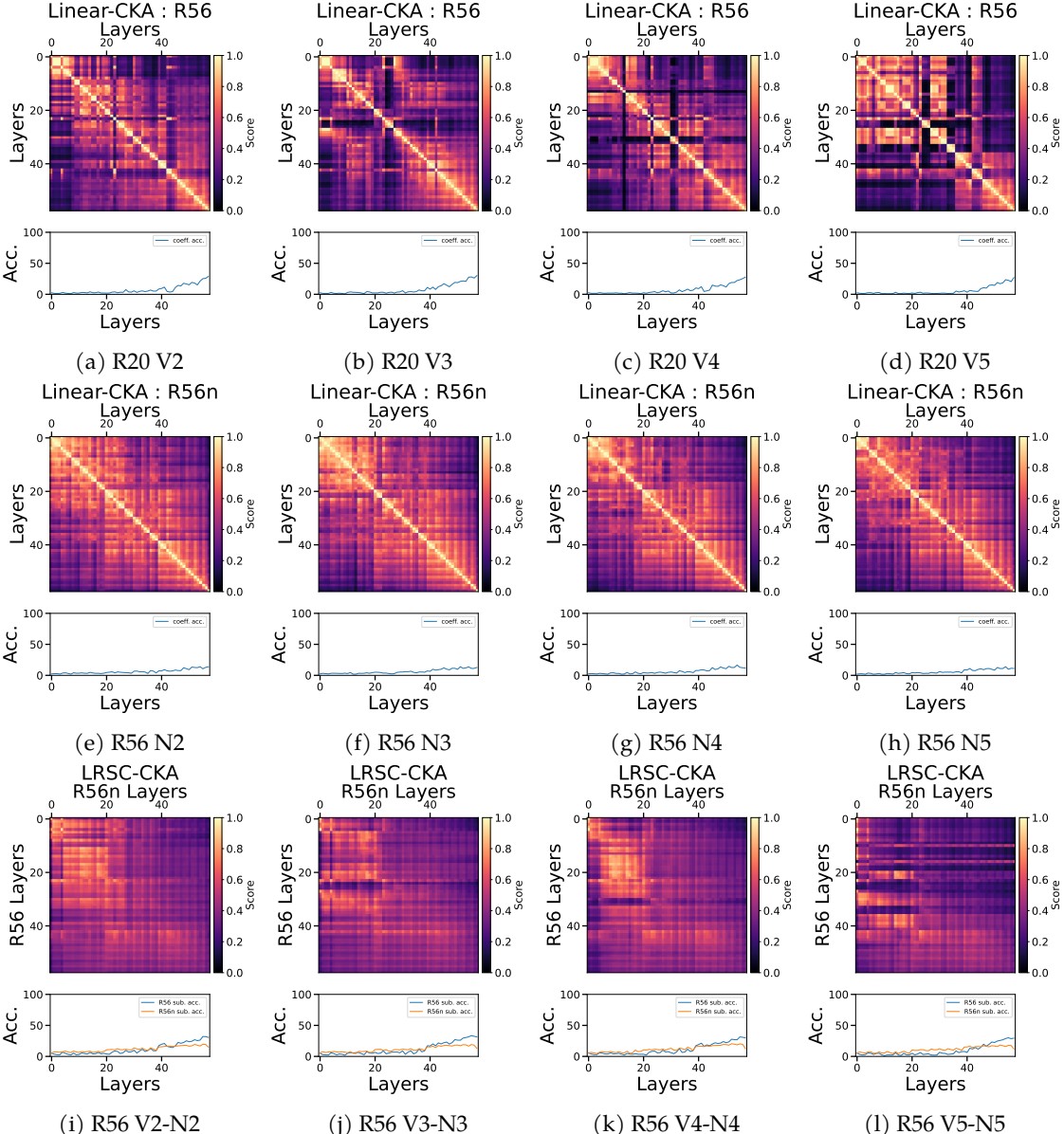

Figure 40: Linear-CKA Analysis on Mini Image Net comparing normally and noisily trained networks. First row consists of 4 normally trained ResNet 56s, the second row consists of 5 noisily trained ResNet 56s. The third row is the comparison between 4 normally and noisily trained ResNets. Therefore, each column in the figure demonstrates a normally trained ReLU network, a noisily trained ReLU network and their comparison.

## J.5. Additional LRSC-CKA Analysis of the previous networks on Mini ImageNet 100

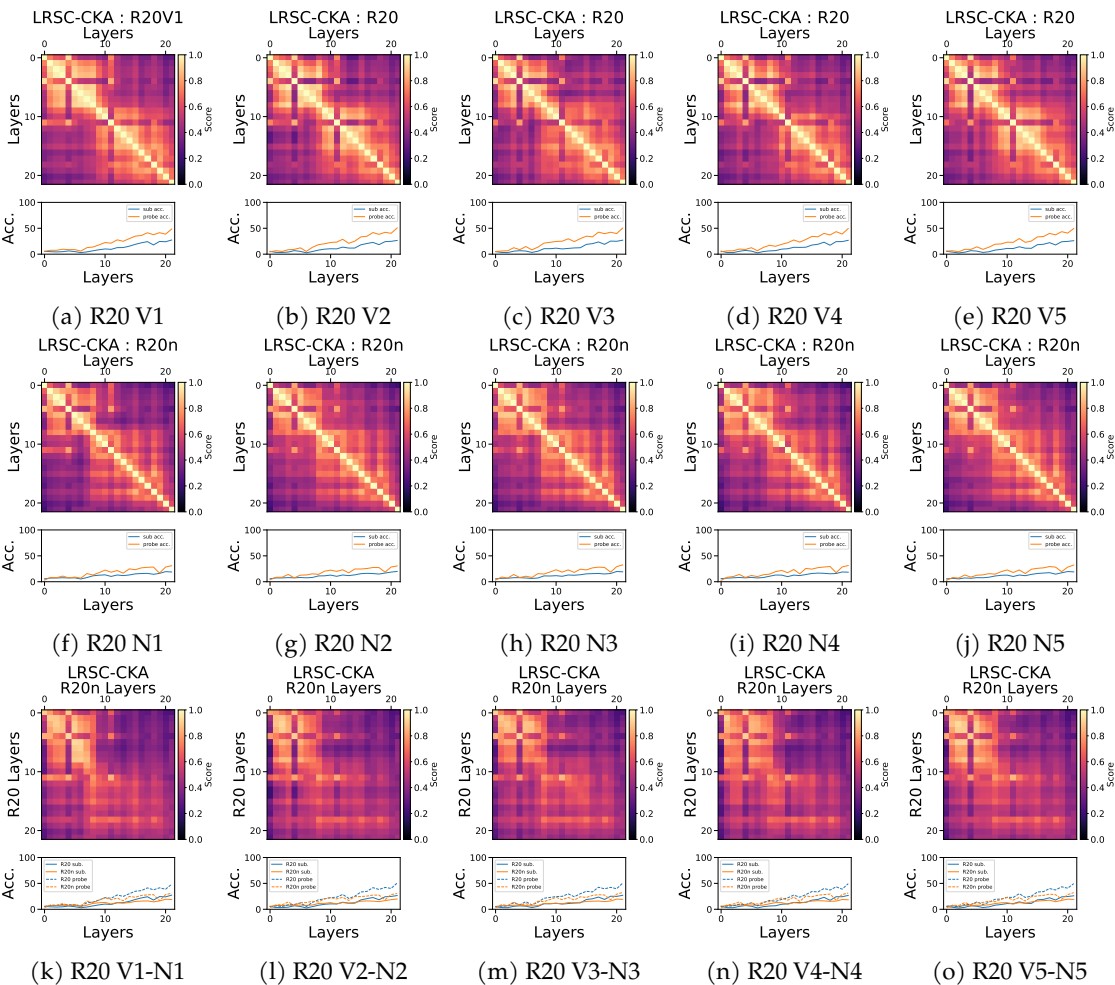

Figure 41: LRSC-CKA Analysis on Mini Image Net comparing normally and noisily trained networks. First row consists of 5 normally trained ResNet 20s, the second row consists of 5 noisily trained ResNet 20s. The third row is the comparison between 5 normally and noisily trained ResNets. Therefore, each column in the figure demonstrates a normally trained ReLU network, a noisily trained ReLU network and their comparison.

## J.6. Corresponding Linear-CKA Analysis of the previous networks on Mini ImageNet 100

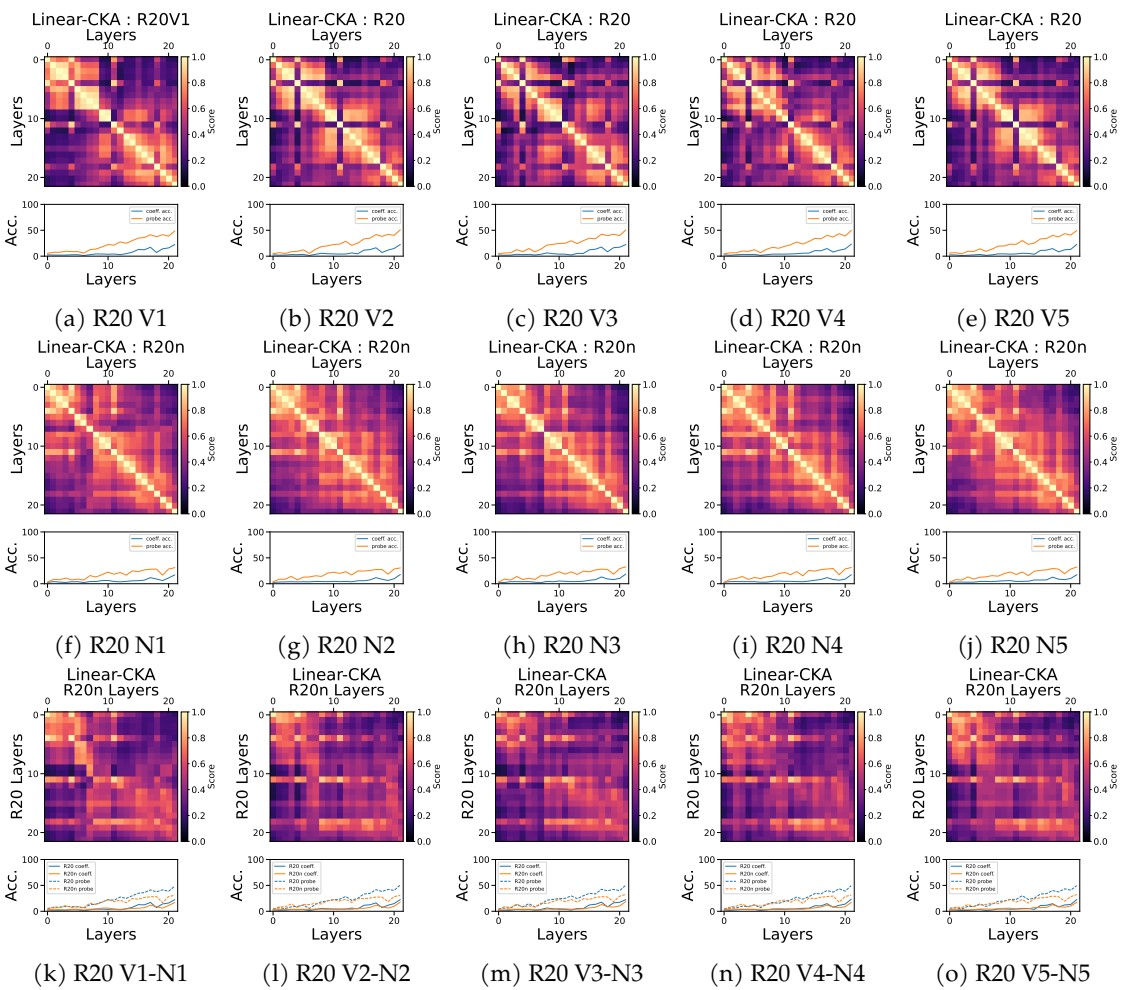

Figure 42: Linear-CKA Analysis on Mini Image Net comparing normally and noisily trained networks.

# K. Additional Results comparing the effects of memorisation and generalization on Bottleneck Multi Layer Perceptrons (B-MLP)

Corresponding the previous analysis comparing self-expressive structures in ResNets under different conditions of generalization in Section 6.2, Section C.1 and Appendix J, in this section we demonstrate a similar set of results for Bottleneck MLPs [65].

## K.1. Aggregated analysis - Bottleneck Multi Layer Perceptrons (B-MLP) on CIFAR10

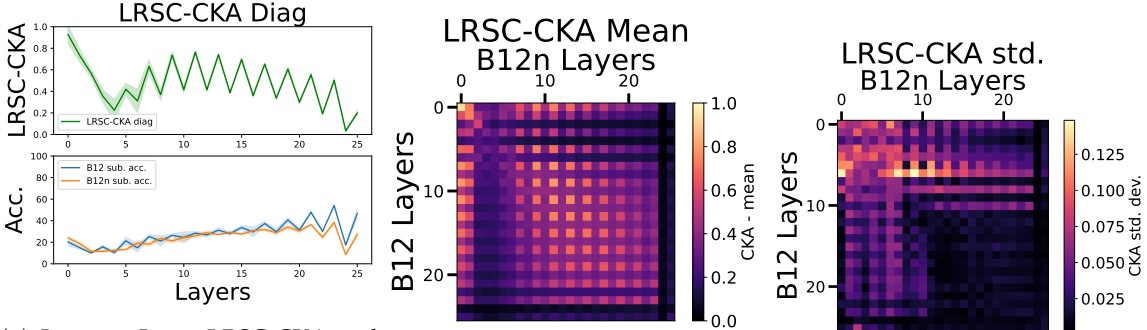

(a) Layer to Layer LRSC-CKA and Subspace Reconstruction Accuracy  (b) LRSC-CKA B12 vs B12n - mean  (c) LRSC-CKA B12 vs B12n - std.

Figure 43: Similar to Figure 7, All analysis presented in this figure are done over 5 pairs of generalising and memorising B-12/Wi-1024 MLPs trained on clean and noisy CIFAR 10 respectively. Figure 43a shows the Layer to Layer comparison between the two networks. Corresponding complete layerwise LRSC-CKA analysis over these 5 pairs is shown in Figure 43b and Figure 43c, mean and standard deviation, respectively.

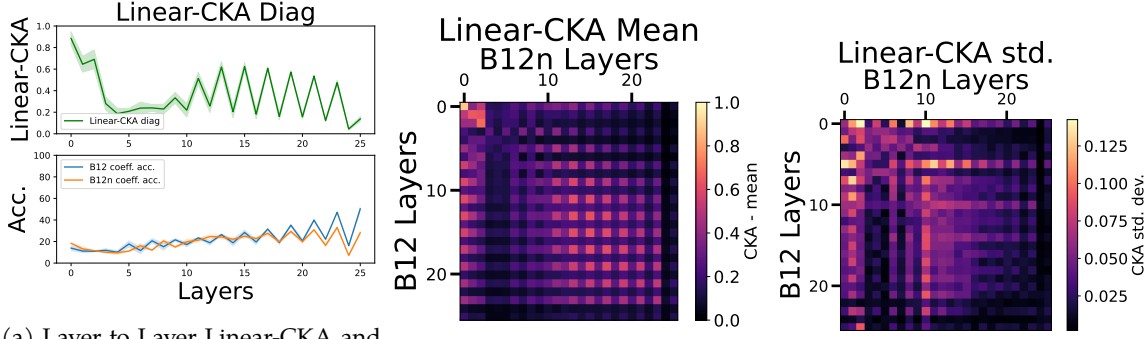

(a) Layer to Layer Linear-CKA and Subspace Reconstruction Accuracy  (b) Linear-CKA B12 vs B12n - mean  (c) Linear-CKA B12 vs B12n - std.

Figure 44: Similar to Figure 26, All analysis presented in this figure are done over 5 pairs of generalising and memorising B-12/Wi-1024 MLPs trained on clean and noisy CIFAR 10 respectively. Figure 43a shows the Layer to Layer comparison between the two networks. Corresponding complete layerwise LRSC-CKA analysis over these 5 pairs is shown in Figure 43b and Figure 43c, mean and standard deviation, respectively.

## K.2. Correlation LRSC analysis - Bottleneck Multi Layer Perceptrons (B-MLP) on CIFAR10

| Pearson's Correlation - layer wise linear probe accuracy vs metrics | | | | | | | | | |
|---|---|---|---|---|---|---|---|---|---|
| Datasets→ | CIFAR 10 | | | | | CIFAR 10n | | | | |
| Metric ↓ | V1 | V2 | V3 | V4 | V5 | N1 | N2 | N3 | N4 | N5 |
| **LRSC recon. acc.** | 0.81 | 0.73 | 0.73 | 0.74 | 0.74 | 0.7 | 0.65 | 0.7 | 0.7 | 0.66 |
| LRSC coeff. acc. | 0.8 | 0.72 | 0.7 | 0.7 | 0.75 | 0.67 | 0.68 | 0.7 | 0.76 | 0.69 |
| LinCKA coeff. acc. | 0.72 | 0.63 | 0.66 | 0.69 | 0.7 | 0.69 | 0.61 | 0.69 | 0.71 | 0.63 |

Table 11: Pearson's Correlation Coefficient between layer wise linear probe accuracies and LRSC-CKA and Linear-CKA metrics based accuracy for B-MLPs trained on CIFAR10.

| Spearman's Correlation - layer wise linear probe accuracy vs metrics | | | | | | | | | |
|---|---|---|---|---|---|---|---|---|---|
| Datasets→ | CIFAR 10 | | | | | CIFAR 10n | | | | |
| Metric ↓ | V1 | V2 | V3 | V4 | V5 | N1 | N2 | N3 | N4 | N5 |
| **LRSC recon. acc.** | 0.83 | 0.78 | 0.81 | 0.79 | 0.74 | 0.72 | 0.64 | 0.74 | 0.77 | 0.69 |
| LRSC coeff. acc. | 0.86 | 0.81 | 0.77 | 0.79 | 0.77 | 0.68 | 0.73 | 0.76 | 0.79 | 0.71 |
| LinCKA coeff. acc. | 0.74 | 0.64 | 0.69 | 0.73 | 0.7 | 0.71 | 0.63 | 0.71 | 0.75 | 0.65 |

Table 12: Spearman's Correlation Coefficient between layer wise linear probe accuracies and LRSC-CKA and Linear-CKA metrics based accuracy for B-MLPs trained on CIFAR10.

| Kendall's tau - layer wise linear probe accuracy vs metrics | | | | | | | | | |
|---|---|---|---|---|---|---|---|---|---|
| Datasets→ | CIFAR 10 | | | | | CIFAR 10n | | | | |
| Metric ↓ | V1 | V2 | V3 | V4 | V5 | N1 | N2 | N3 | N4 | N5 |
| **LRSC recon. acc.** | 0.67 | 0.61 | 0.62 | 0.64 | 0.55 | 0.48 | 0.43 | 0.51 | 0.6 | 0.49 |
| LRSC coeff. acc. | 0.71 | 0.63 | 0.62 | 0.63 | 0.58 | 0.48 | 0.52 | 0.55 | 0.58 | 0.53 |
| LinCKA coeff. acc. | 0.58 | 0.52 | 0.54 | 0.56 | 0.52 | 0.52 | 0.46 | 0.49 | 0.55 | 0.44 |

Table 13: Kendall's tau between layer wise linear probe accuracies and LRSC-CKA and Linear-CKA metrics based accuracy for B-MLPs trained on CIFAR10.

## K.3. Aggregated analysis -Bottleneck Multi Layer Perceptrons (B-MLP) on CIFAR100

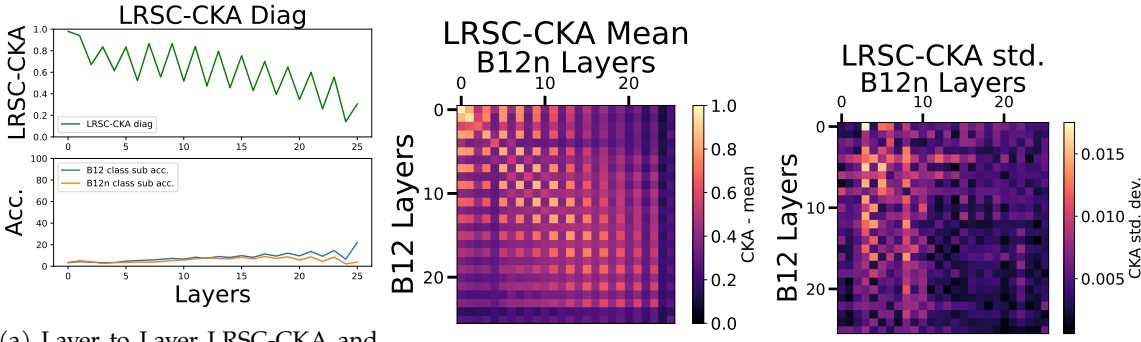

(a) Layer to Layer LRSC-CKA and Subspace Reconstruction Accuracy   (b) LRSC-CKA B12 vs B12n - mean   (c) LRSC-CKA B12 vs B12n - std.

Figure 45: Similar to Figure 43, Corresponding analysis over CIFAR 100.

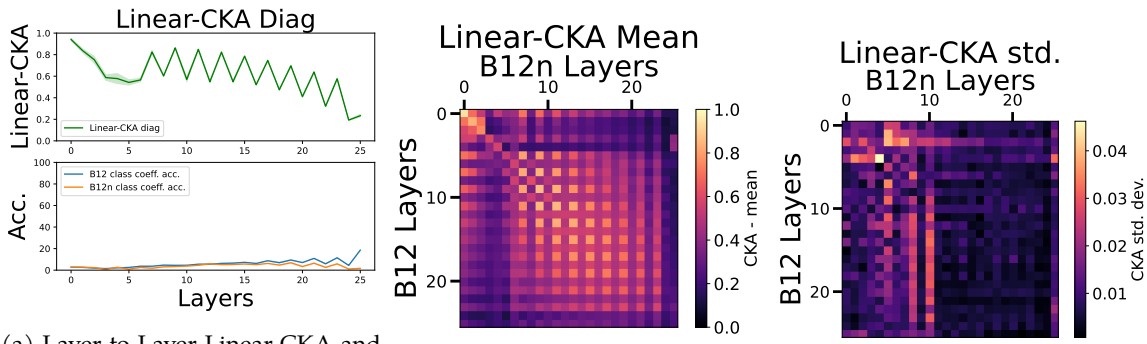

(a) Layer to Layer Linear-CKA and Subspace Reconstruction Accuracy   (b) Linear-CKA B12 vs B12n - mean   (c) Linear-CKA B12 vs B12n - std.

Figure 46: Similar to Figure 44, Corresponding analysis over CIFAR 100.

## K.4. Correlation analysis - Bottleneck Multi Layer Perceptrons (B-MLP) on CIFAR100

| Pearson's Correlation - layer wise linear probe accuracy vs metrics | | | | | | | | | |
|---|---|---|---|---|---|---|---|---|---|
| Datasets→ | CIFAR 100 | | | | | CIFAR 100n | | | |
| Metric ↓ | V1 | V2 | V3 | V4 | V5 | N1 | N2 | N3 | N4 | N5 |
| **LRSC recon. acc.** | 0.83 | 0.8 | 0.79 | 0.85 | 0.8 | 0.62 | 0.71 | 0.63 | 0.57 | 0.54 |
| LRSC coeff. acc. | 0.77 | 0.67 | 0.71 | 0.76 | 0.74 | 0.65 | 0.64 | 0.6 | 0.55 | 0.49 |
| LinCKA coeff. acc. | 0.8 | 0.79 | 0.71 | 0.79 | 0.77 | 0.56 | 0.62 | 0.6 | 0.54 | 0.51 |

Table 14: Pearson's Correlation Coefficient between layer wise linear probe accuracies and LRSC-CKA and Linear-CKA metrics based accuracy for B-MLPs trained on CIFAR100.

| Spearman's Correlation - layer wise linear probe accuracy vs metrics | | | | | | | | | |
|---|---|---|---|---|---|---|---|---|---|
| Datasets→ | CIFAR 100 | | | | | CIFAR 100n | | | |
| Metric ↓ | V1 | V2 | V3 | V4 | V5 | N1 | N2 | N3 | N4 | N5 |
| **LRSC recon. acc.** | 0.91 | 0.88 | 0.91 | 0.92 | 0.93 | 0.58 | 0.73 | 0.61 | 0.52 | 0.49 |
| LRSC coeff. acc. | 0.81 | 0.68 | 0.78 | 0.79 | 0.81 | 0.64 | 0.67 | 0.57 | 0.47 | 0.48 |
| LinCKA coeff. acc. | 0.86 | 0.85 | 0.87 | 0.88 | 0.88 | 0.55 | 0.62 | 0.6 | 0.5 | 0.51 |

Table 15: Spearman's Correlation Coefficient between layer wise linear probe accuracies and LRSC-CKA and Linear-CKA metrics based accuracy for B-MLPs trained on CIFAR100.

| Kendall's tau - layer wise linear probe accuracy vs metrics | | | | | | | | | |
|---|---|---|---|---|---|---|---|---|---|
| Datasets→ | CIFAR 100 | | | | | CIFAR 100n | | | |
| Metric ↓ | V1 | V2 | V3 | V4 | V5 | N1 | N2 | N3 | N4 | N5 |
| **LRSC recon. acc.** | 0.78 | 0.72 | 0.76 | 0.79 | 0.79 | 0.39 | 0.51 | 0.43 | 0.37 | 0.34 |
| LRSC coeff. acc. | 0.64 | 0.5 | 0.59 | 0.6 | 0.64 | 0.43 | 0.49 | 0.42 | 0.34 | 0.31 |
| LinCKA coeff. acc. | 0.69 | 0.68 | 0.72 | 0.72 | 0.72 | 0.35 | 0.38 | 0.41 | 0.35 | 0.37 |

Table 16: Kendall's tau between layer wise linear probe accuracies and LRSC-CKA and Linear-CKA metrics based accuracy for B-MLPs trained on CIFAR100.

## K.5. Aggregated analysis - Bottleneck Multi Layer Perceptrons (B-MLP) on Mini Image Net 100

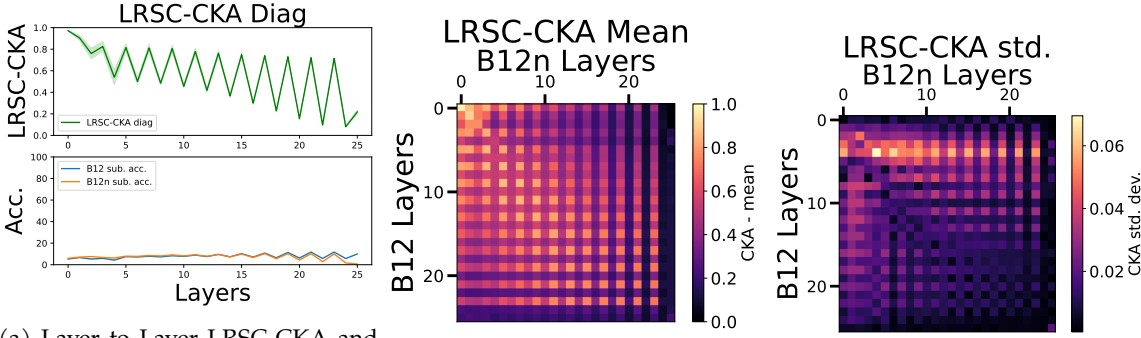

(a) Layer to Layer LRSC-CKA and Subspace Reconstruction Accuracy    (b) LRSC-CKA B12 vs B12n - mean    (c) LRSC-CKA B12 vs B12n - std.

Figure 47: Similar to Figure 43, Corresponding analysis over Mini Image Net 100.

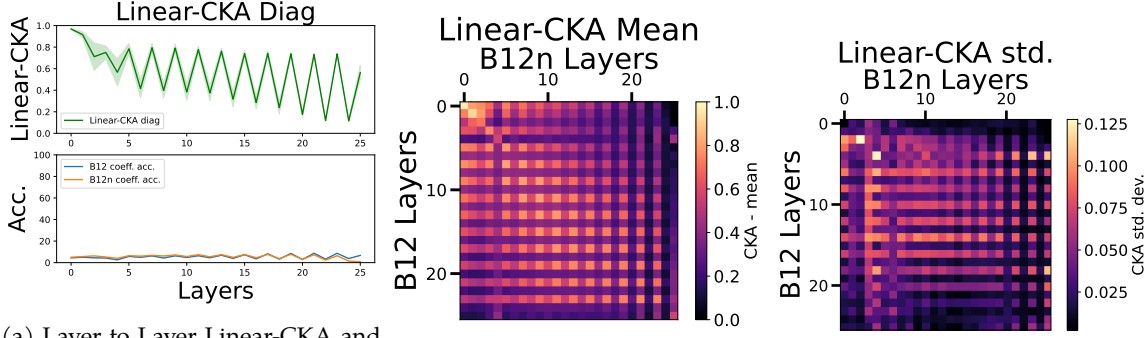

(a) Layer to Layer Linear-CKA and Subspace Reconstruction Accuracy    (b) Linear-CKA B12 vs B12n - mean    (c) Linear-CKA B12 vs B12n - std.

Figure 48: Similar to Figure 44, Corresponding analysis over Mini Image Net 100.

## K.6. Correlation analysis - Bottleneck Multi Layer Perceptrons (B-MLP) on MIN100

| Pearson's Correlation - layer wise linear probe accuracy vs metrics | | | | | | | | | | |
|---|---|---|---|---|---|---|---|---|---|---|
| Datasets→ | | MIN 100 | | | | | MIN 100n | | | |
| Metric ↓ | V1 | V2 | V3 | V4 | V5 | N1 | N2 | N3 | N4 | N5 |
| **LRSC recon. acc.** | 0.81 | 0.82 | 0.82 | 0.83 | 0.84 | 0.57 | 0.57 | 0.55 | 0.51 | 0.51 |
| LRSC coeff. acc. | 0.87 | 0.88 | 0.86 | 0.91 | 0.91 | 0.63 | 0.6 | 0.62 | 0.57 | 0.56 |
| LinCKA coeff. acc. | 0.66 | 0.56 | 0.64 | 0.6 | 0.63 | 0.51 | 0.58 | 0.53 | 0.55 | 0.57 |

Table 17: Pearson's Correlation Coefficient between layer wise linear probe accuracies and LRSC-CKA and Linear-CKA metrics based accuracy for B-MLPs trained on MIN100.

| Spearman's Correlation - layer wise linear probe accuracy vs metrics | | | | | | | | | | |
|---|---|---|---|---|---|---|---|---|---|---|
| Datasets→ | | MIN 100 | | | | | MIN 100n | | | |
| Metric ↓ | V1 | V2 | V3 | V4 | V5 | N1 | N2 | N3 | N4 | N5 |
| **LRSC recon. acc.** | 0.87 | 0.83 | 0.84 | 0.85 | 0.88 | 0.44 | 0.58 | 0.53 | 0.54 | 0.53 |
| LRSC coeff. acc. | 0.95 | 0.88 | 0.89 | 0.96 | 0.95 | 0.49 | 0.56 | 0.56 | 0.54 | 0.48 |
| LinCKA coeff. acc. | 0.71 | 0.54 | 0.6 | 0.59 | 0.6 | 0.37 | 0.55 | 0.43 | 0.53 | 0.51 |

Table 18: Spearman's Correlation Coefficient between layer wise linear probe accuracies and LRSC-CKA and Linear-CKA metrics based accuracy for B-MLPs trained on MIN100.

| Kendall's tau - layer wise linear probe accuracy vs metrics | | | | | | | | | | |
|---|---|---|---|---|---|---|---|---|---|---|
| Datasets→ | | MIN 100 | | | | | MIN 100n | | | |
| Metric ↓ | V1 | V2 | V3 | V4 | V5 | N1 | N2 | N3 | N4 | N5 |
| **LRSC recon. acc.** | 0.69 | 0.63 | 0.67 | 0.68 | 0.75 | 0.35 | 0.51 | 0.42 | 0.44 | 0.44 |
| LRSC coeff. acc. | 0.83 | 0.71 | 0.73 | 0.86 | 0.84 | 0.41 | 0.51 | 0.46 | 0.49 | 0.37 |
| LinCKA coeff. acc. | 0.51 | 0.39 | 0.45 | 0.43 | 0.46 | 0.3 | 0.44 | 0.34 | 0.42 | 0.42 |

Table 19: Kendall's tau between layer wise linear probe accuracies and LRSC-CKA and Linear-CKA metrics based accuracy for B-MLPs trained on MIN100.

# L. Additional Results comparing the effects of memorisation and generalization on Kolmogorov-Arnold Networks (KANs)

Following the results shown in Appendix K, in this section we demonstrate similar results on generalisation and memorisation for Kolmogorov-Arnold Networks (KANs) [66].

## L.1. Aggregated analysis - Kolmogorov-Arnold Networks (KANs) on CIFAR10

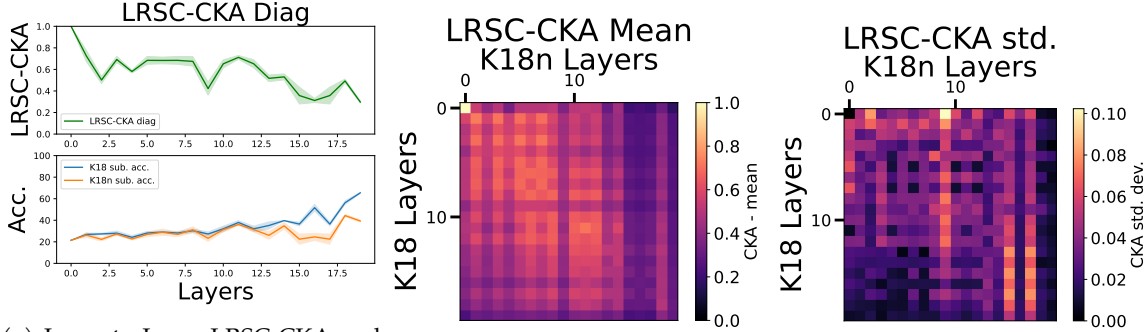

(a) Layer to Layer LRSC-CKA and Subspace Reconstruction Accuracy

(b) LRSC-CKA K18 vs K18n - mean

(c) LRSC-CKA K18 vs K18n - std.

Figure 49: Similar to Figure 43, Corresponding analysis of ResKANs on CIFAR 10.

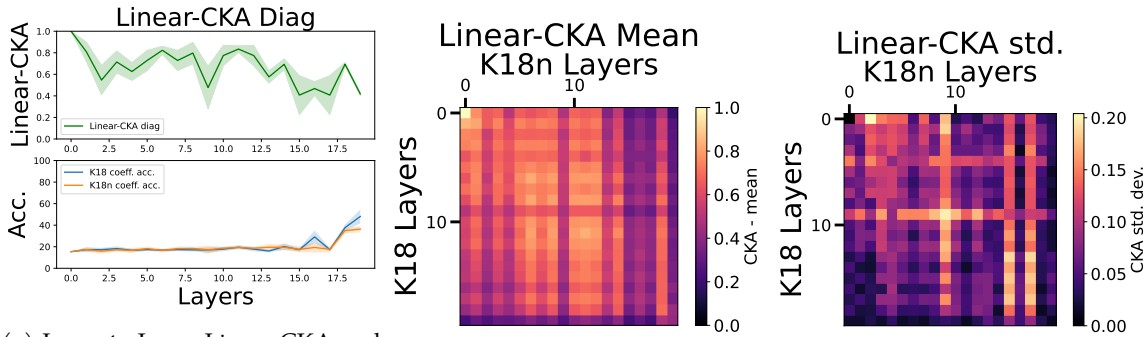

(a) Layer to Layer Linear-CKA and Subspace Reconstruction Accuracy

(b) Linear-CKA K18 vs K18n - mean

(c) Linear-CKA K18 vs K18n - std.

Figure 50: Similar to Figure 44, Corresponding analysis of ResKANs on CIFAR 10.

## L.2. Aggregated analysis - Kolmogorov-Arnold Networks (KANs) on CIFAR100

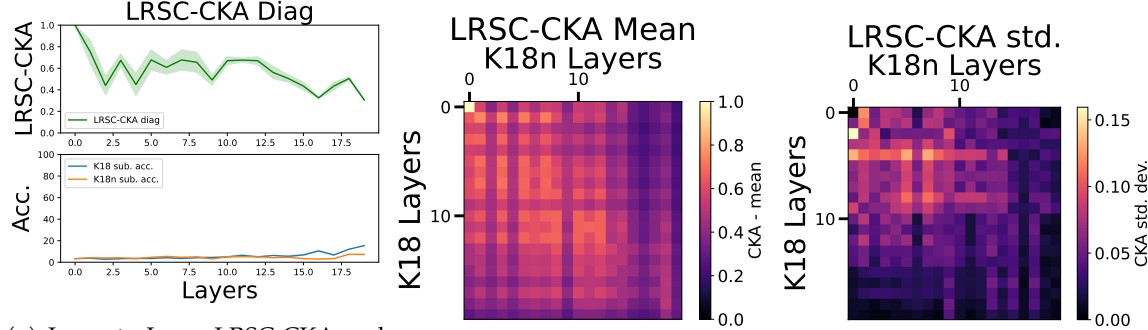

(a) Layer to Layer LRSC-CKA and Subspace Reconstruction Accuracy

(b) LRSC-CKA K18 vs K18n - mean

(c) LRSC-CKA K18 vs K18n - std.

Figure 51: Similar to Figure 43, Corresponding analysis of ResKANs on CIFAR 100.

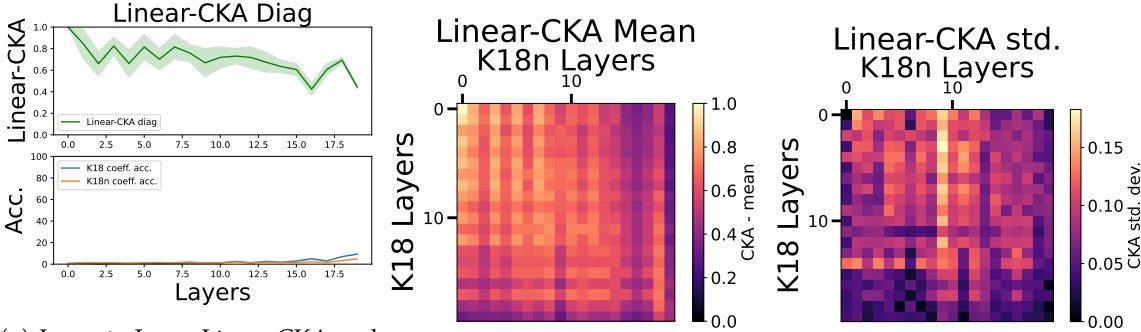

(a) Layer to Layer Linear-CKA and Subspace Reconstruction Accuracy

(b) Linear-CKA K18 vs K18n - mean

(c) Linear-CKA K18 vs K18n - std.

Figure 52: Similar to Figure 44, Corresponding analysis of ResKANs on CIFAR 100.

## L.3. Aggregated analysis - Kolmogorov-Arnold Networks (KANs) on MIN100

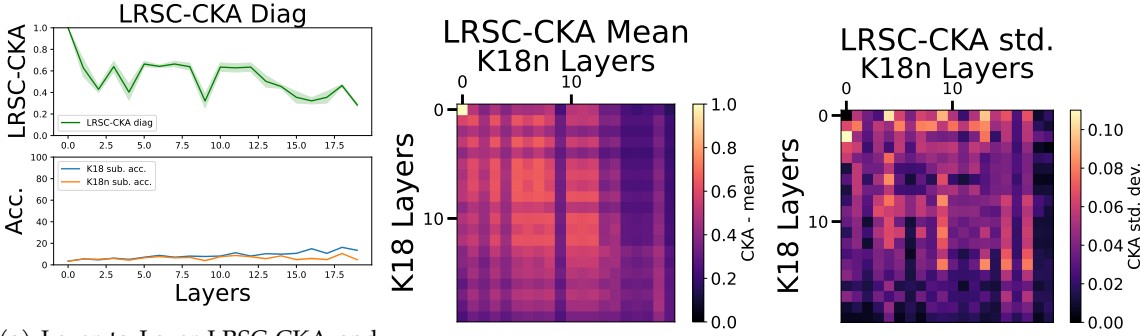

(a) Layer to Layer LRSC-CKA and Subspace Reconstruction Accuracy  (b) LRSC-CKA K18 vs K18n - mean  (c) LRSC-CKA K18 vs K18n - std.

Figure 53: Similar to Figure 43, Corresponding analysis of ResKANs on MIN 100.

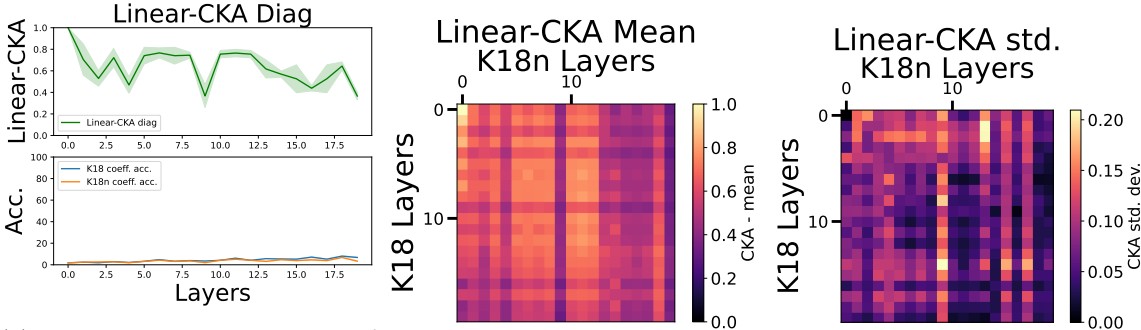

(a) Layer to Layer Linear-CKA and Subspace Reconstruction Accuracy  (b) Linear-CKA K18 vs K18n - mean  (c) Linear-CKA K18 vs K18n - std.

Figure 54: Similar to Figure 44, Corresponding analysis of ResKANs on MIN 100.

# M. Additional Results on LRSC-CKA, Linear-CKA and MFTMA analysis of Rational and ReLU networks under normal and noisy training regimes

In this section we include additional supporting results for experiments conducted in Appendix D. We being by first providing a brief description of manifold capacity as stated in MFTMA[19] - Given K object manifolds with random binary labels each represented by a cloud of points in a D-dimensional space having the same label, [19] defines manifold capacity $\alpha = K/D$ by the number of object manifolds where most manifold dichotomies can be separated by a hyperplane. They [19] relate this to Cover's function counting theorem[67], but the a key deviation being that the fundamental counting objects for MFTMA[19] are manifolds rather than discrete points. This allows [19] to argue about linear separability of manifolds rather than discrete points. Earlier work [49] showed that manifold capacity for point clouds can be formulated based on the manifold capacity of D-dimensional balls of radius R with random orientations in the ambient space, also providing closed form expressions for effective dimension $D_M$ and effective radius $R_M$ for the convex hulls of general point clouds in random orientation. Subsequent advances by [47] help with estimation of manifold capacities in real data. More specifically, it connects linear separability of class manifolds ($\alpha_M$) with the class manifold's geometric properties like manifold dimension - $D_M$, manifold radius - $R_M$ and the correlations between the manifold centres - $\rho_{center}$.

In Figure 55 of Section M.1 we present the full set of results shown in Figure 9 of Appendix D and in Section M.3 we present the corresponding analysis with Linear-CKA. Similarly, for Figure 10 of Appendix D we present the full set of results in Figure 56 of Section M.2 and the corresponding Linear-CKA results in Section M.4.

For all Rational Neural Nets in these experiments, we follow the rational function choices from [21] and train networks in normal and noisy regimes on CIFAR10. The performance of ReLU ResNets used is shown in Table 20 and the performance of Rational networks is shown in Table 21.

Table 20: Performance of ReLU Networks (ResNet 20) used in these experiments on the probing set.

| Performance (%) over normal and noisy training regimes | | | | | | | | | |
|---|---|---|---|---|---|---|---|---|---|
| Regime→ | Normal CIFAR 10 | | | | | Noisy CIFAR 10 | | | | |
| Metric ↓ | V1 | V2 | V3 | V4 | V5 | N1 | N2 | N3 | N4 | N5 |
| **Accuracy %** | 91.2 | 92 | 93 | 92.2 | 90.6 | 66.6 | 67.8 | 64 | 64 | 68.2 |

Table 21: Performance of Rational Networks (ResNet 20 Rat) used in these experiments on the probing set.

| Performance (%) over normal and noisy training regimes | | | | | | | | | |
|---|---|---|---|---|---|---|---|---|---|
| Regime→ | Normal CIFAR 10 | | | | | Noisy CIFAR 10 | | | | |
| Metric ↓ | R1 | R2 | R3 | R4 | R5 | RN1 | RN2 | RN3 | RN4 | RN5 |
| **Accuracy %** | 89 | 88.6 | 87.6 | 89 | 88 | 48.6 | 47 | 43.2 | 45.2 | 45.6 |

## M.1. LRSC-CKA, Linear Probe and MFTMA based comparison of ReLU Networks under normal and noisy training regimes of CIFAR10

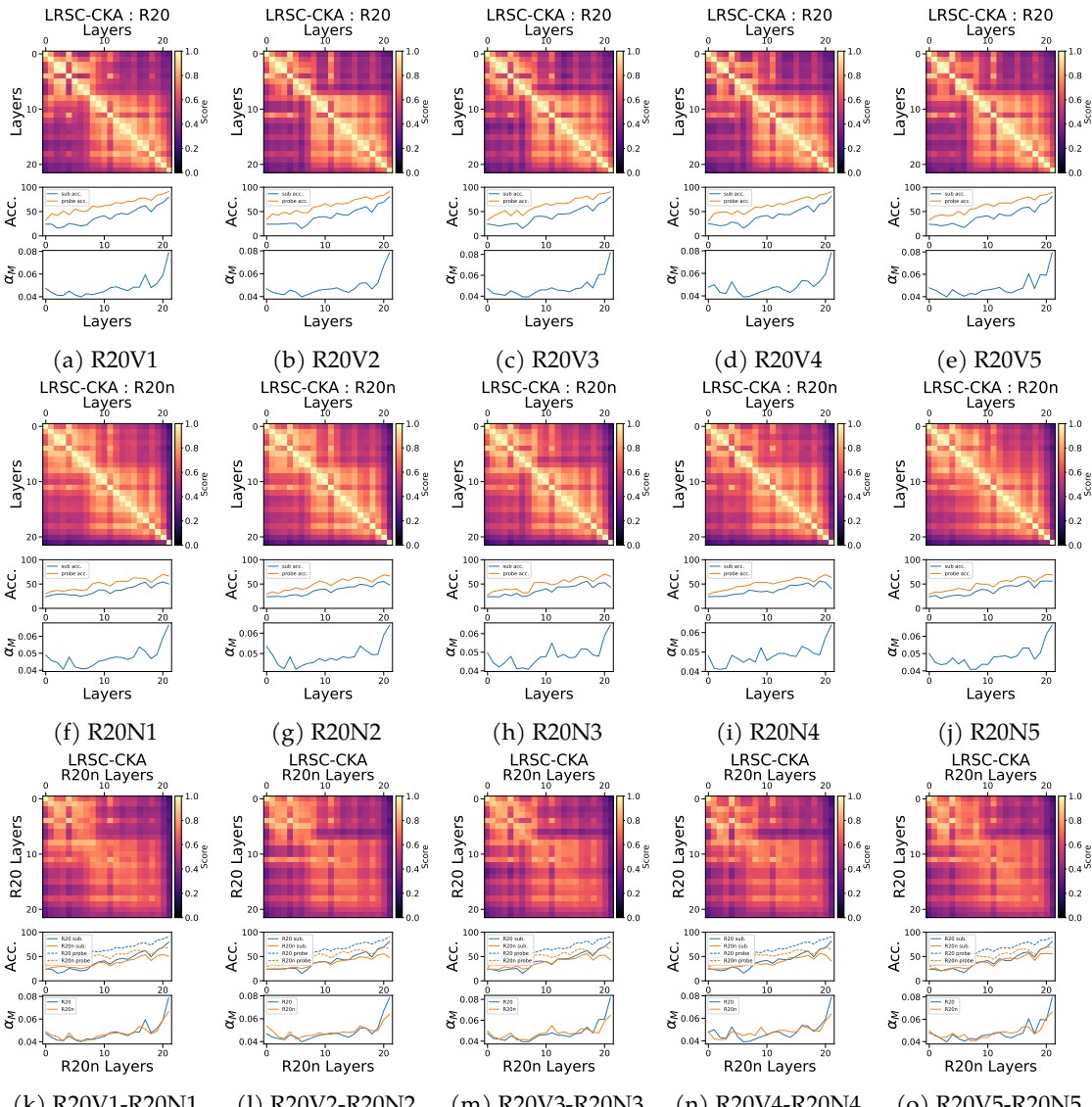

Figure 55: Extended results of LRSC and MFTMA based comparison of ResNet 20 trained with ReLU Activations under normal and noisy label settings on the CIFAR10 dataset. First row consists of 5 normally trained ReLU ResNets, the second row consists of 5 noisily trained ReLU ResNets. The third row is the comparison between 5 normally and noisily trained ReLU ResNets. Therefore, each column in the figure demonstrates a normally trained ReLU network, a noisily trained ReLU network and their comparison. The subsequent figures in Appendix M follow the same layout, whether for LRSC-CKA or Linear-CKA or ReLU ResNets or Rational ResNets.

## M.2. LRSC-CKA, Linear Probe and MFTMA based comparison of Rational Networks under normal and noisy training regimes of CIFAR10

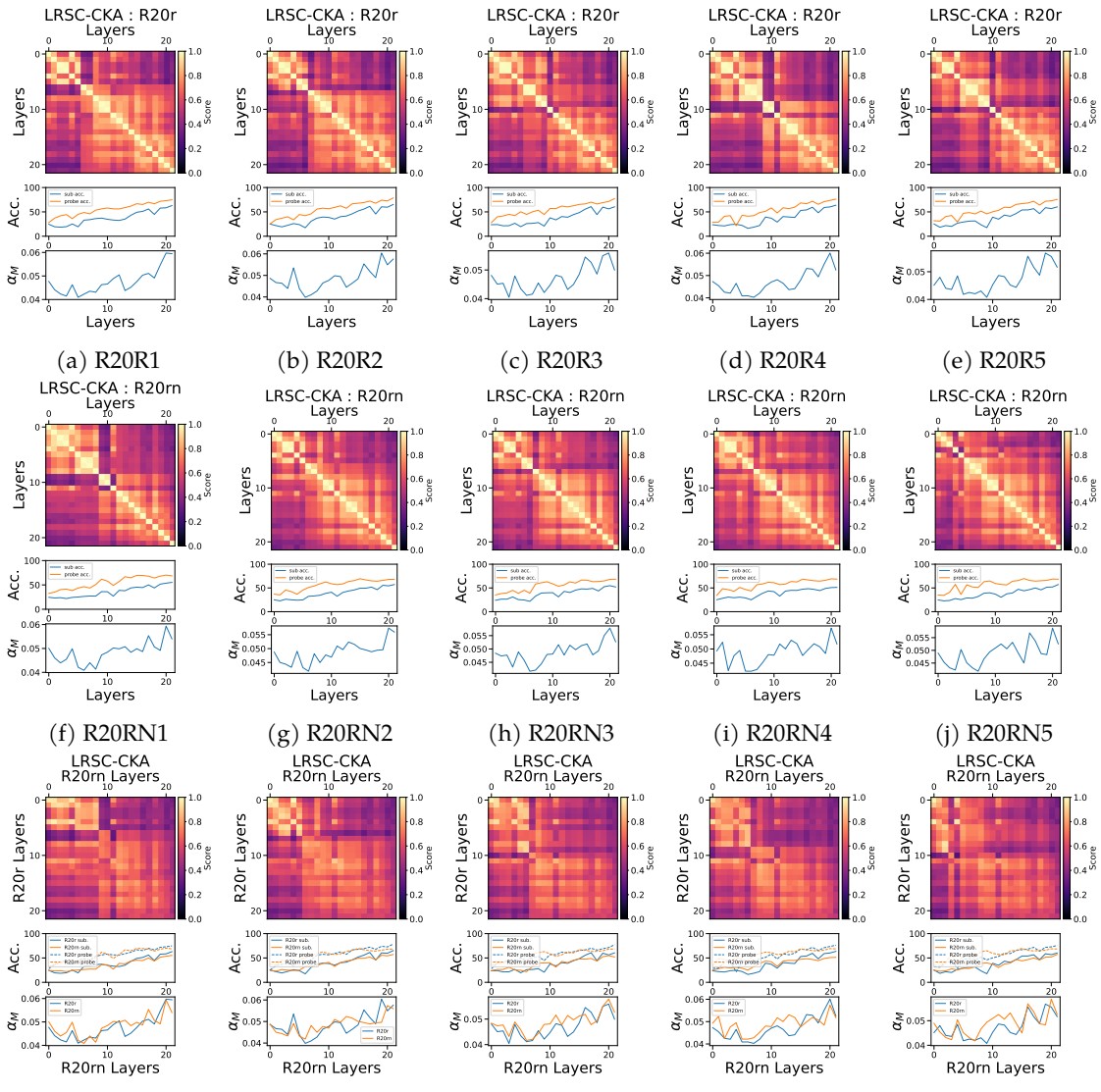

(a) R20R1    (b) R20R2    (c) R20R3    (d) R20R4    (e) R20R5

(f) R20RN1   (g) R20RN2   (h) R20RN3   (i) R20RN4   (j) R20RN5

(k) R20R1-R20RN1   (l) R20R2-R20RN2   (m) R20R3-R20RN3   (n) R20R4-R20RN4   (o) R20R5-R20RN5

Figure 56: Extended results of LRSC and MFTMA based comparison of ResNet 20 trained with Rational Activations under normal and noisy label settings on the CIFAR10 dataset. First row consists of 5 normally trained Rational ResNets, the second row consists of 5 noisily trained Rational ResNets. The third row is the comparison between 5 normally and noisily trained Rational ResNets. Each column in the figure demonstrates a normally trained Rational network, a noisily trained Rational network and their comparison.

## M.3. Linear-CKA, Linear Probe and MFTMA based comparison of ReLU Networks under normal and noisy training regimes of CIFAR10

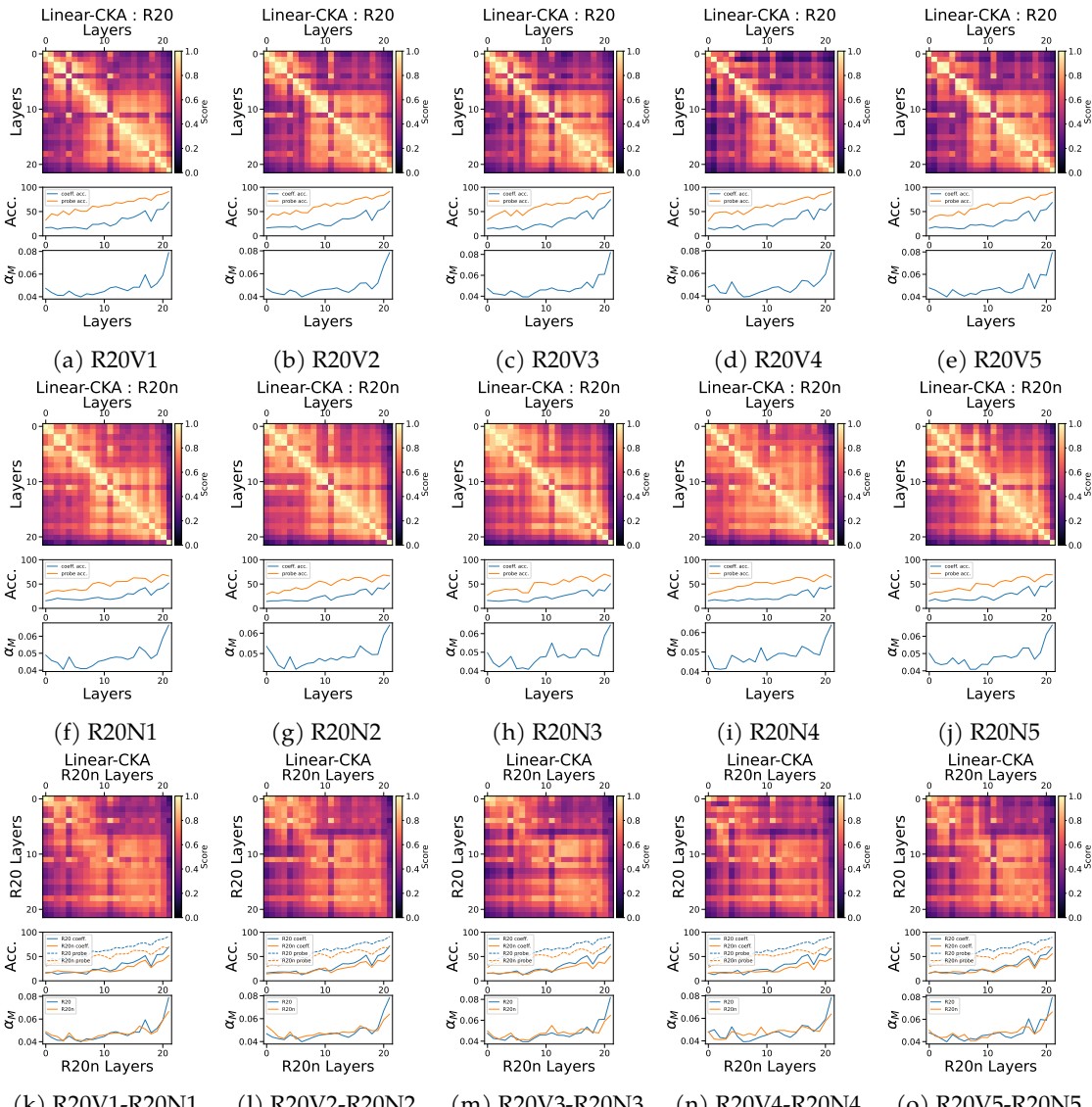

Figure 57: Extended results of Linear-CKA and MFTMA based comparison of ResNet 20 trained with ReLU Activations under normal and noisy label settings on the CIFAR10 dataset. First row consists of 5 normally trained ReLU ResNets, the second row consists of 5 noisily trained ReLU ResNets. The third row is the comparison between 5 normally and noisily trained ReLU ResNets. Each column in the figure demonstrates a normally trained ReLU network, a noisily trained ReLU network and their comparison.

## M.4. Linear-CKA, Linear Probe and MFTMA based comparison of Rational Networks under normal and noisy training regimes of CIFAR10

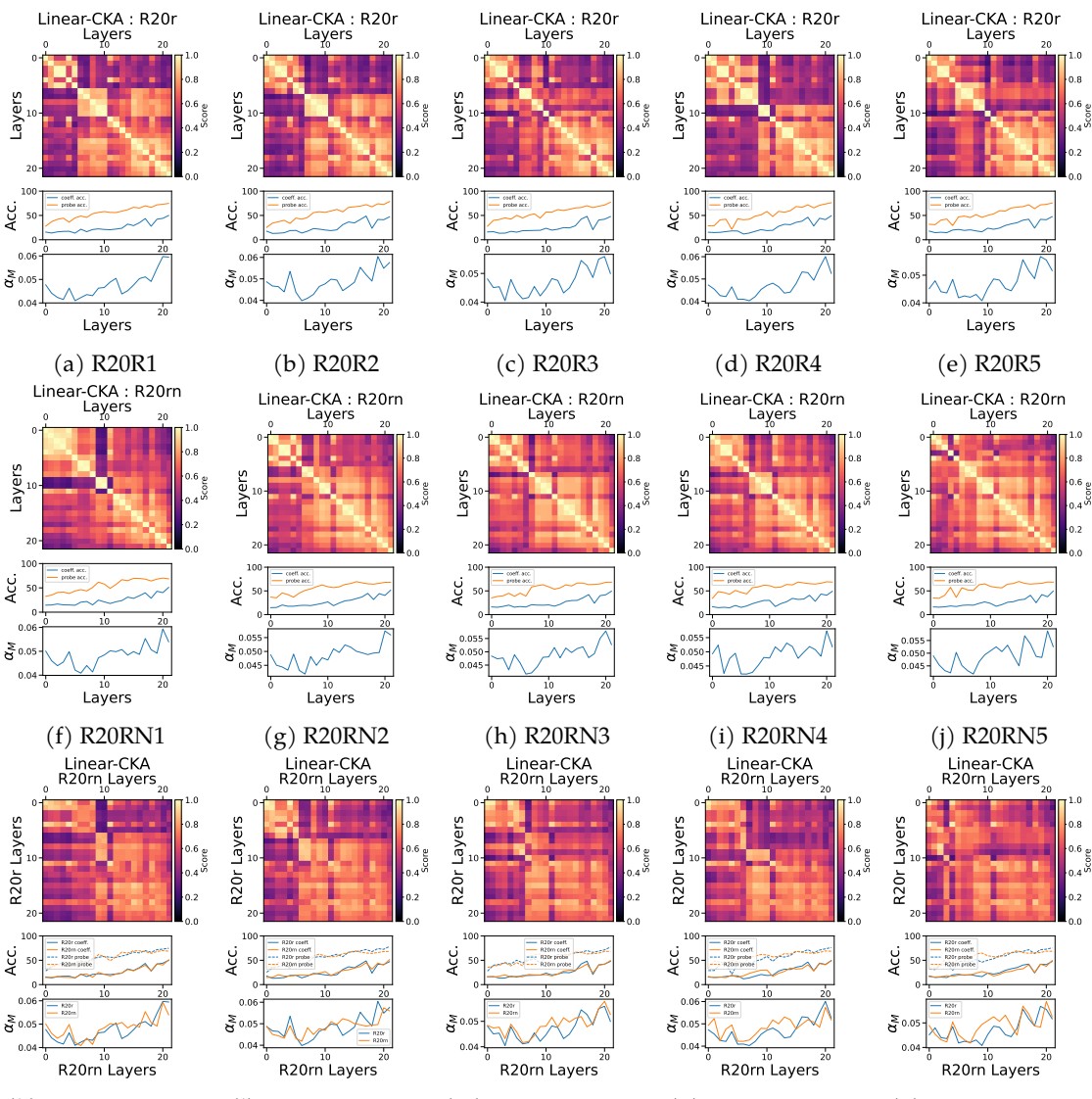

(a) R20R1  (b) R20R2  (c) R20R3  (d) R20R4  (e) R20R5

(f) R20RN1  (g) R20RN2  (h) R20RN3  (i) R20RN4  (j) R20RN5

(k) R20R1-R20RN1  (l) R20R2-R20RN2  (m) R20R3-R20RN3  (n) R20R4-R20RN4  (o) R20R5-R20RN5

Figure 58: Extended results of Linear-CKA and MFTMA based comparison of ResNet 20 trained with Rational Activations under normal and noisy label settings on the CIFAR10 dataset. First row consists of 5 normally trained Rational ResNets, the second row consists of 5 noisily trained Rational ResNets. The third row is the comparison between 5 normally and noisily trained Rational ResNets. Each column in the figure demonstrates a normally trained Rational network, a noisily trained ReLU network and Rational comparison.

# N. Additional results involving comparisons between ReLU and Rational Networks across training regimes

In this section we show additional results for Figure 12 in Appendix D where we take ReLU and Rational ResNets trained across normal and noisy regimes and perform a cross comparison between the two architectures. The goal of Figure 12 in Appendix D was to demonstrate that memorisation in ReLU ResNets caused the final few layers to have different representations from the rest of the network. Especially when compared to normally trained ReLU and Rational networks, additionally we also observed that most but the last layers of a noisily trained ReLU network shared some similarity with all layers of a noisily trained rational network. Indicating that the effects of memorisation manifest differently when using highly non-linear activations. Next we show additional results for Figure 12 of cross-comparison between the quadruplet of networks with different random initialisations in Figure 59 and Figure 60 of Section N.1 using LRSC-CKA. Corresponding comparisons using Linear-CKA are shown in Figure 61 and Figure 62 of Section N.2.

## N.1. LRSC-CKA based comparisons between ReLU and Rational Networks across training regimes

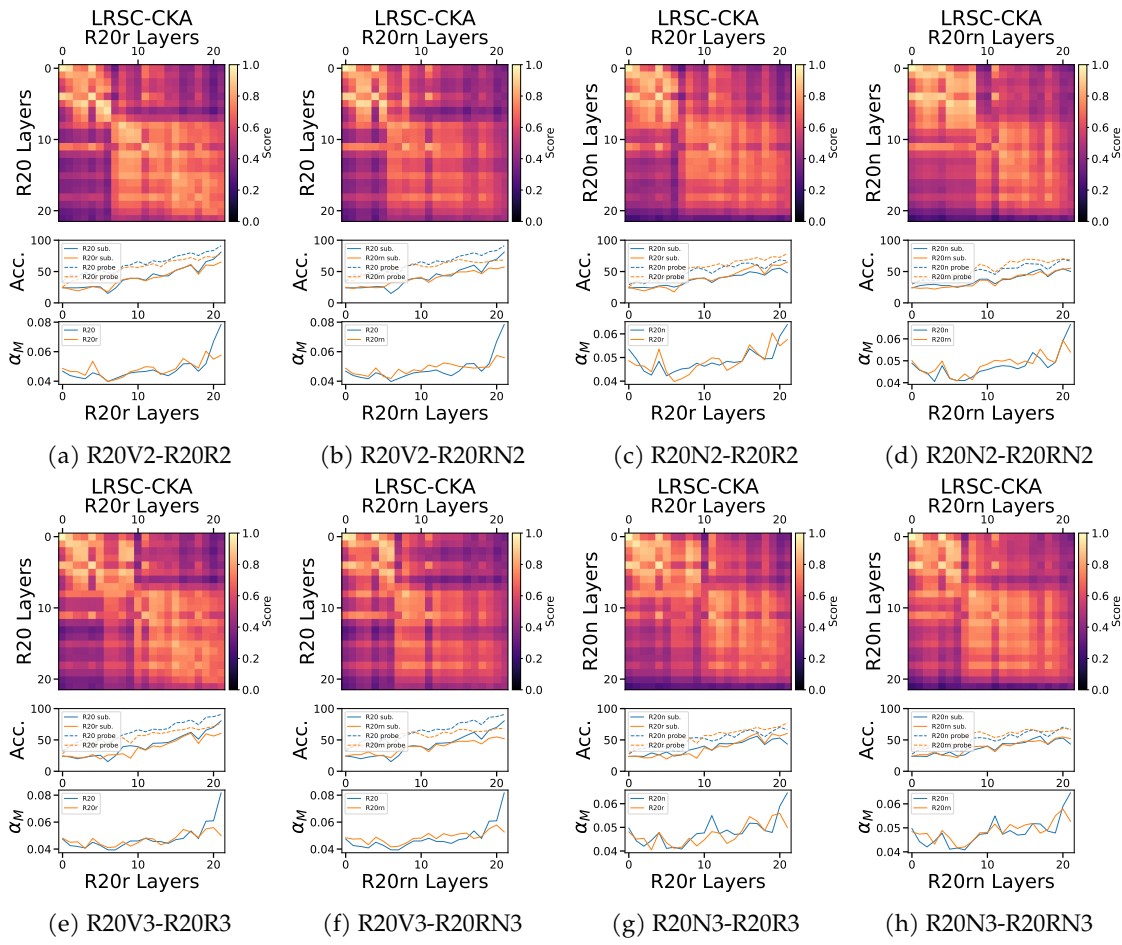

(a) R20V2-R20R2    (b) R20V2-R20RN2    (c) R20N2-R20R2    (d) R20N2-R20RN2

(e) R20V3-R20R3    (f) R20V3-R20RN3    (g) R20N3-R20R3    (h) R20N3-R20RN3

Figure 59: Continuing upon Figure 12 we next show results for cross comparison of ResNet 20 trained with ReLU and Rational Polynomial Activations under normal and noisy label settings. Each row of this figure and the next show a inter-activation comparison of ReLU and Rational ResNets trained in normal and noisy regimes. We demonstrate that the final layers of a noisily trained ReLU ResNet is dissimilar to most layers of a Rational ResNet, regardless of its performance. This isn't true for a normally trained ReLU ResNet.

## N.2. Linear-CKA based comparisons between ReLU and Rational Networks across training regimes

In this section we show the Linear-CKA analogues for the results shown in Section N.1. Figure 61h contains the Linear-CKA counterpart to results of Figure 59 and Figure 62 shows the corresponding Linear-CKA results for Figure 60.

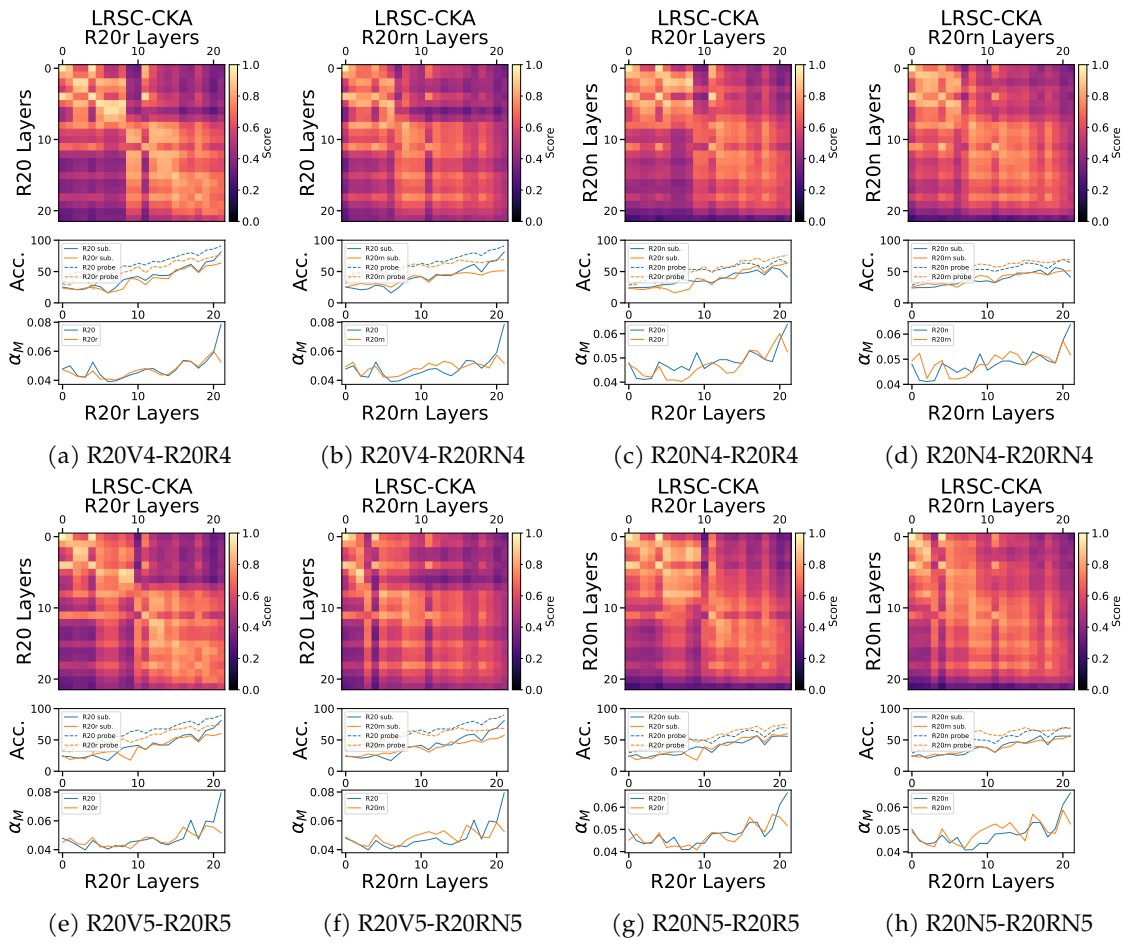

Figure 60: Additional extended results for all pairs comparison of ResNet 20 trained with ReLU and Rational Polynomial Activations under normal and noisy label settings on the CIFAR10 dataset.

# O. Additional Results comparing the effects of training neural networks on cross entropy vs maximum coding rate reduction loss

Here we describe additional results for the experiments conducted in Appendix E. We begin so by first describing the experimental setup used, which is based on the experimental setup in [22]. For the purpose of these experiments we use ResNets [6] with depths ranging from 18 to 101. The networks used in this experiment are based on the code in this github repo[4]. For ResNets trained with Cross-Entropy loss on CIFAR10 and CIFAR100 we use a learning rate of 0.1 with a weight decay of $10^{-5}$ trained for 164 epochs with learning rate step reduction by a factor of 0.1 at epochs 81 and 122. For networks trained with MCRR loss on CIFAR10 and CIFAR100 we use a learning rate of 0.001 and a weight decay of $10^{-4}$ for 800 epochs with learning rate step reduction by 0.1 at epochs 200 and 400. When training the networks with MCRR loss on CIFAR100, we use the 20 super classes as labels based on the protocol described in Appendix B.2 of [22].

---

[4]https://github.com/kuangliu/pytorch-cifar

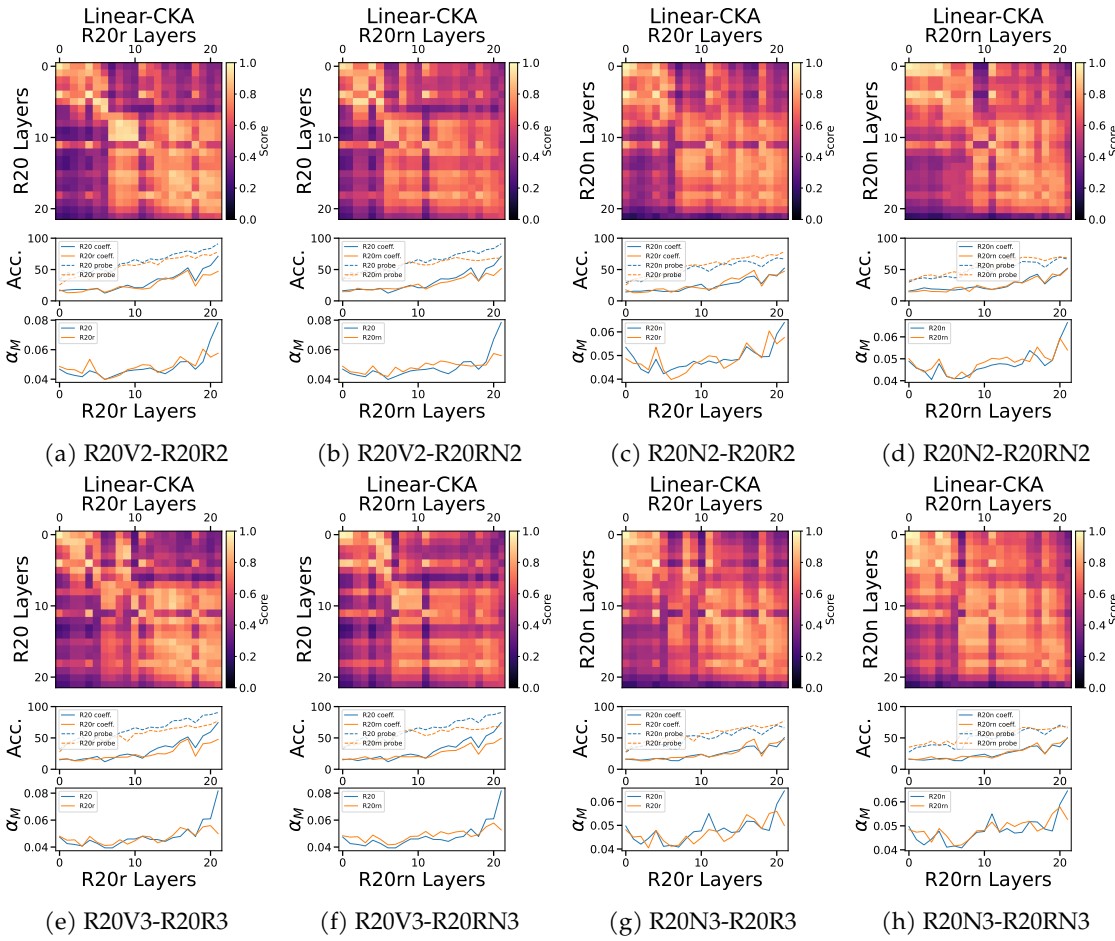

(a) R20V2-R20R2     (b) R20V2-R20RN2     (c) R20N2-R20R2     (d) R20N2-R20RN2

(e) R20V3-R20R3     (f) R20V3-R20RN3     (g) R20N3-R20R3     (h) R20N3-R20RN3

Figure 61: Part 1/2 Linear-CKA analogue of extended results shown for all pairs comparison of ResNet 20 trained with ReLU and Rational Polynomial Activations under normal and noisy label settings on the CIFAR10 dataset.

## O.1. Additional LRSC-CKA results analyzing the effects of training ResNets with Maximal Coding Rate Reduction and Cross Entropy losses on CIFAR10

In this section we show the comparison of ResNets trained on Cross Entropy and MCRR loss across different network sizes. We take 4 ResNets of different sizes, namely - 18,34,50, 101 and train them on the two loses and then compare the same architecture over the 2 losses. The results are shown in Figure 63. These results are an extension of results shown in Figure 13f.

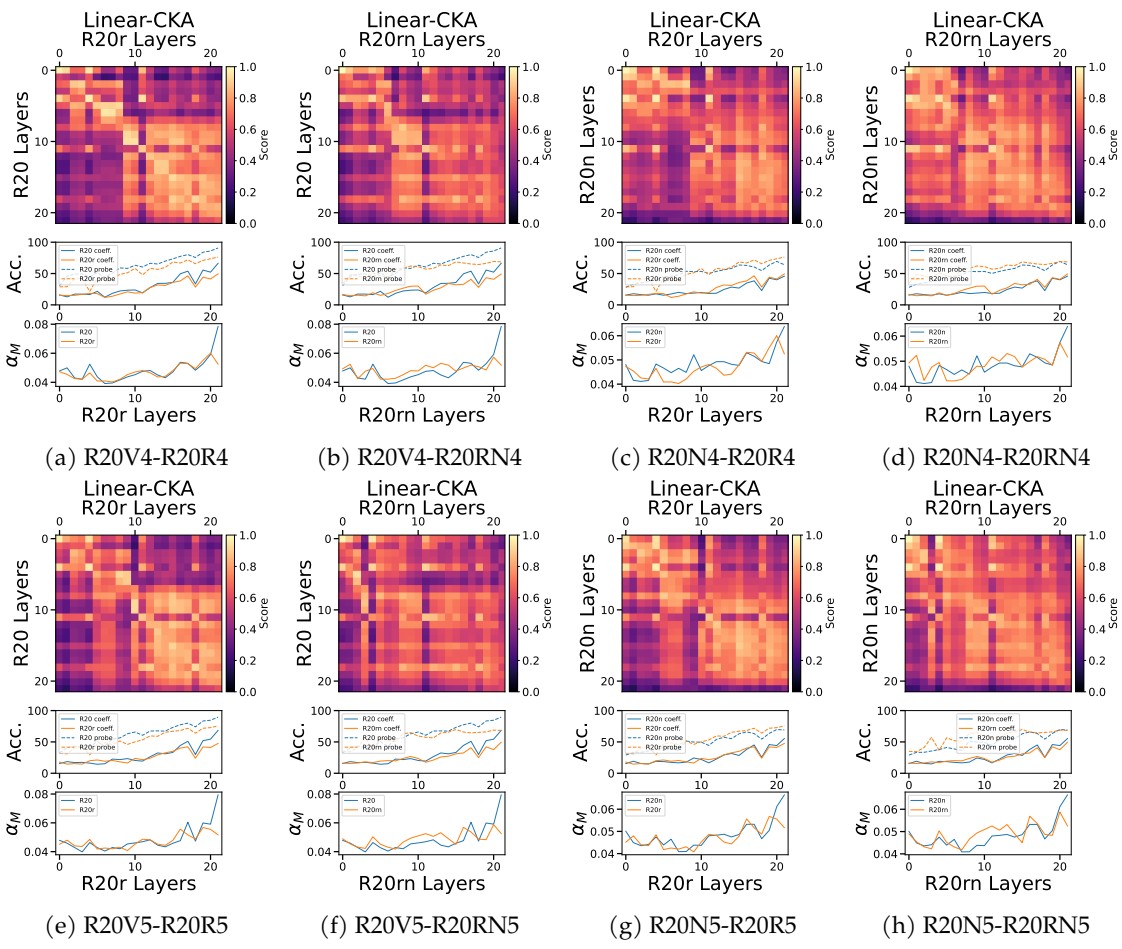

(a) R20V4-R20R4  (b) R20V4-R20RN4  (c) R20N4-R20R4  (d) R20N4-R20RN4

(e) R20V5-R20R5  (f) R20V5-R20RN5  (g) R20N5-R20R5  (h) R20N5-R20RN5

Figure 62: Part 2 - Extended results for all pairs comparison of ResNet 20 trained with ReLU and Rational Polynomial Activations under normal and noisy label settings on the CIFAR10 dataset.

## O.2. Additional LRSC-CKA results analyzing the effects of training ResNets with Maximal Coding Rate Reduction and Cross Entropy losses on CIFAR100

Similar to Section O.1, in Figure 64 of this section we show additional results of comparisons between Cross Entropy Loss and MCRR Loss on CIFAR100 that were presented in Figure 13f.

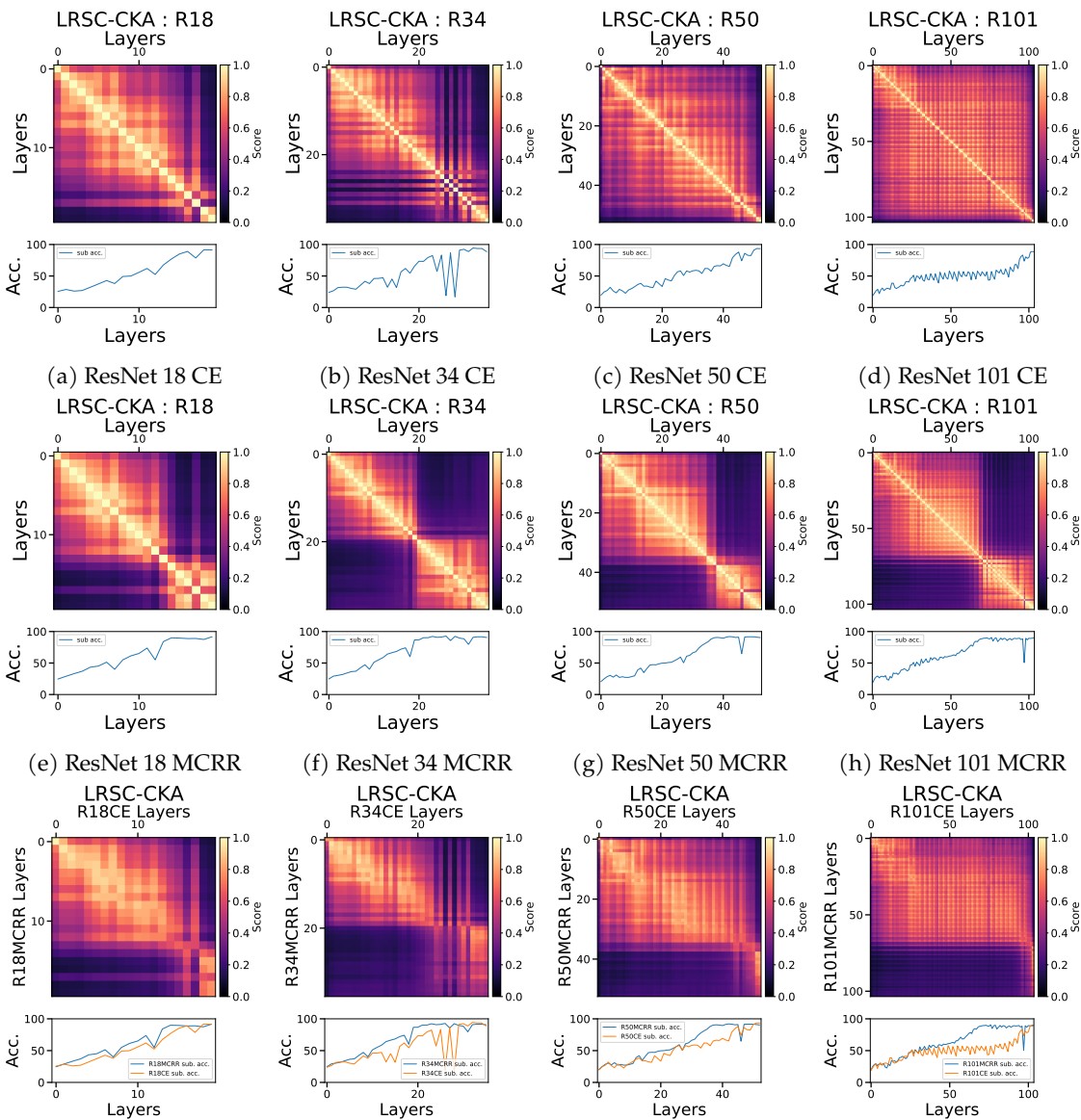

(a) ResNet 18 CE  (b) ResNet 34 CE  (c) ResNet 50 CE  (d) ResNet 101 CE

(e) ResNet 18 MCRR  (f) ResNet 34 MCRR  (g) ResNet 50 MCRR  (h) ResNet 101 MCRR

(i) ResNet 18 MCRR - CE  (j) ResNet 34 MCRR - CE  (k) ResNet 50 MCRR - CE  (l) ResNet 101 MCRR - CE

Figure 63: LRSC Analysis of ResNets trained with Maximal Coding Rate Reduction and Cross Entropy loss on CIFAR10. The first row shows various ResNets trained using the Cross Entropy loss and the second row shows the same networks trained using the MCRR loss. The third row offers corresponding pairwise comparisons between architecture's training on Cross Entropy loss and MCRR loss. A column of this figure therefore indicates a ResNet of a given depth trained on the Cross Entropy Loss, the same ResNet architecture trained on MCRR Loss and a comparison between the two different ResNets. All the 4 columns together demonstrate the emergence of self expressive structures in Cross Entropy trained networks towards their later stages, regardless of network size. All other subsequent figures in Appendix O, whether for LRSC-CKA or Linear-CKA for both CIFAR10 and CIFAR100 follow a similar schematic unless otherwise stated.

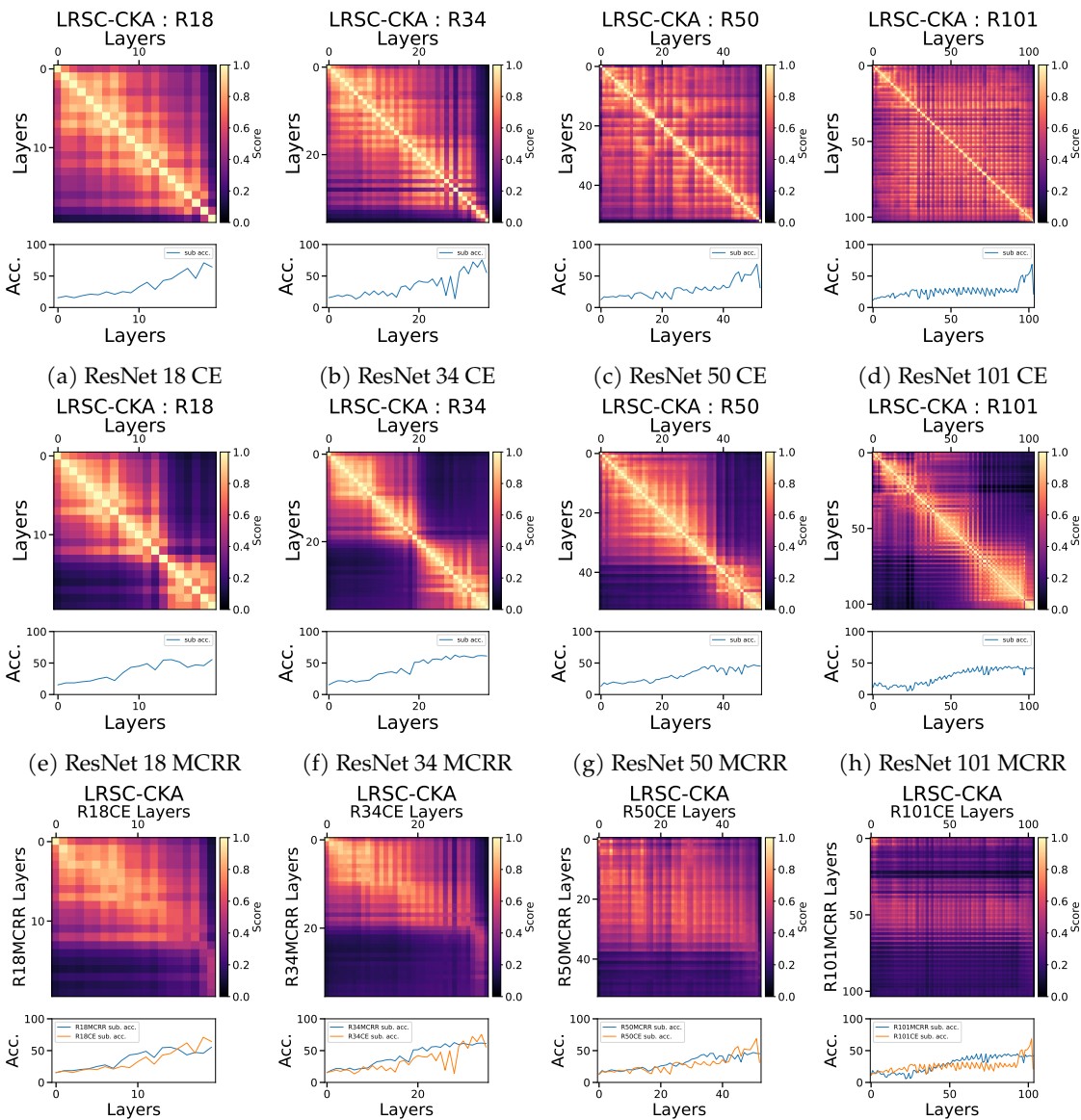

(i) ResNet 18 MCRR - CE  (j) ResNet 34 MCRR - CE  (k) ResNet 50 MCRR - CE  (l) ResNet 101 MCRR - CE

Figure 64: LRSC Analysis of ResNets trained with Maximal Coding Rate Reduction and Cross Entropy loss on CIFAR100. As stated previously, The first row shows various ResNets trained using the Cross Entropy loss, the second row shows the same networks trained using the MCRR loss and the third row offers corresponding pairwise comparisons between architecture's training on Cross Entropy loss and MCRR loss. A column of this figure therefore indicates a ResNet of a given depth trained on the Cross Entropy Loss, the same ResNet architecture trained on MCRR Loss and a comparison between the two different ResNets.

## O.3. Observing and analyzing the effects of training ResNets with Maximal Coding Rate Reduction and Cross Entropy losses on CIFAR10 with Linear-CKA

In this section we lay down the Linear-CKA counter part of the results on CIFAR10 shown in Section O.1 comparing Cross Entropy vs MCRR trained ResNets in Figure 65.

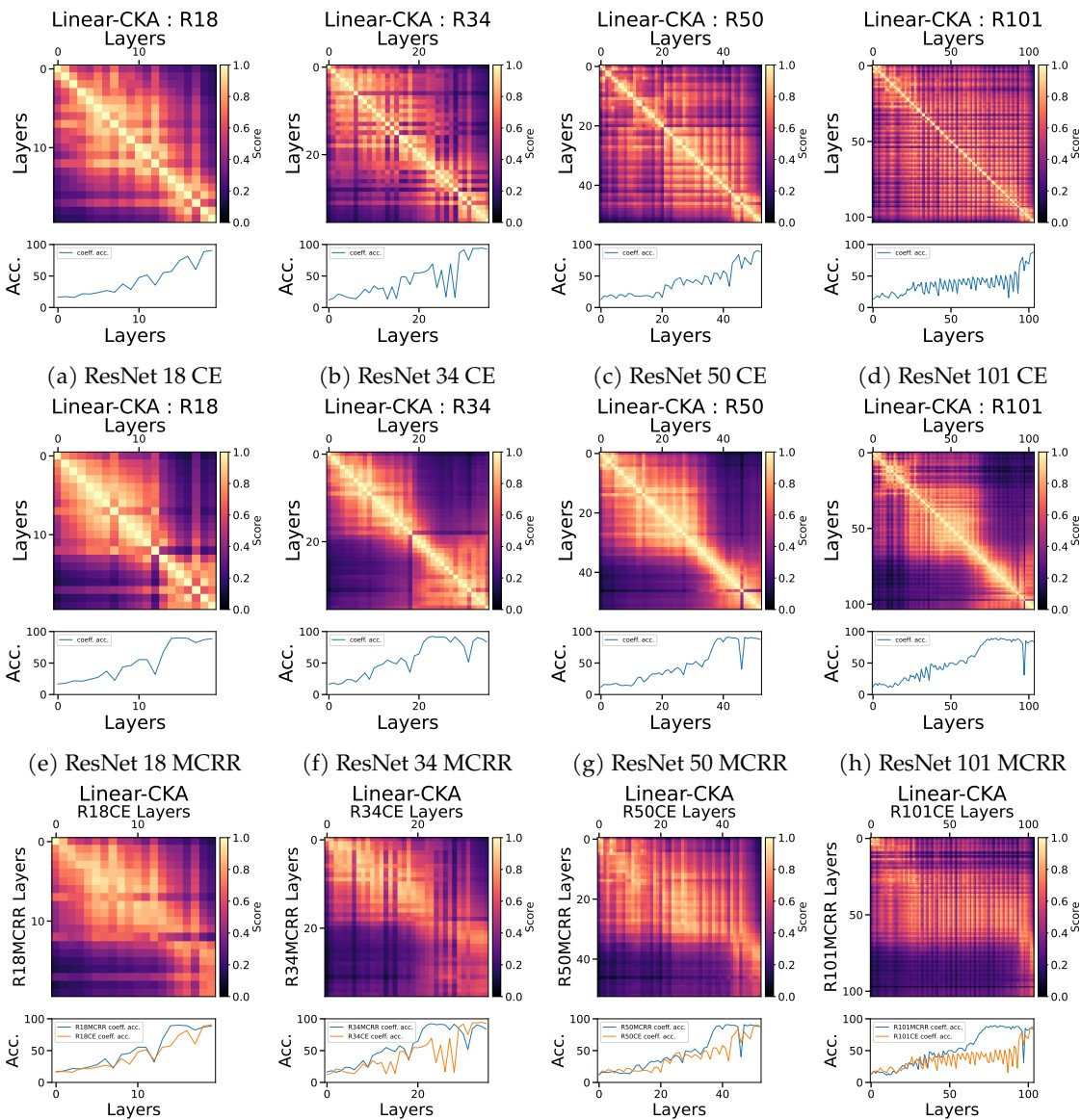

(a) ResNet 18 CE  (b) ResNet 34 CE  (c) ResNet 50 CE  (d) ResNet 101 CE

(e) ResNet 18 MCRR  (f) ResNet 34 MCRR  (g) ResNet 50 MCRR  (h) ResNet 101 MCRR

(i) ResNet 18 MCRR - CE  (j) ResNet 34 MCRR - CE  (k) ResNet 50 MCRR - CE  (l) ResNet 101 MCRR - CE

Figure 65: Linear CKA Analysis of ResNets trained with Maximal Coding Rate Reduction and Cross Entropy loss on CIFAR10.

## O.4. Observing and analyzing the effects of training ResNets with Maximal Coding Rate Reduction and Cross Entropy losses on CIFAR100 with Linear-CKA

In this section we lay down the Linear-CKA counter part of the results on CIFAR100 shown in Section O.2 comparing Cross Entropy vs MCRR trained ResNets in Figure 66.

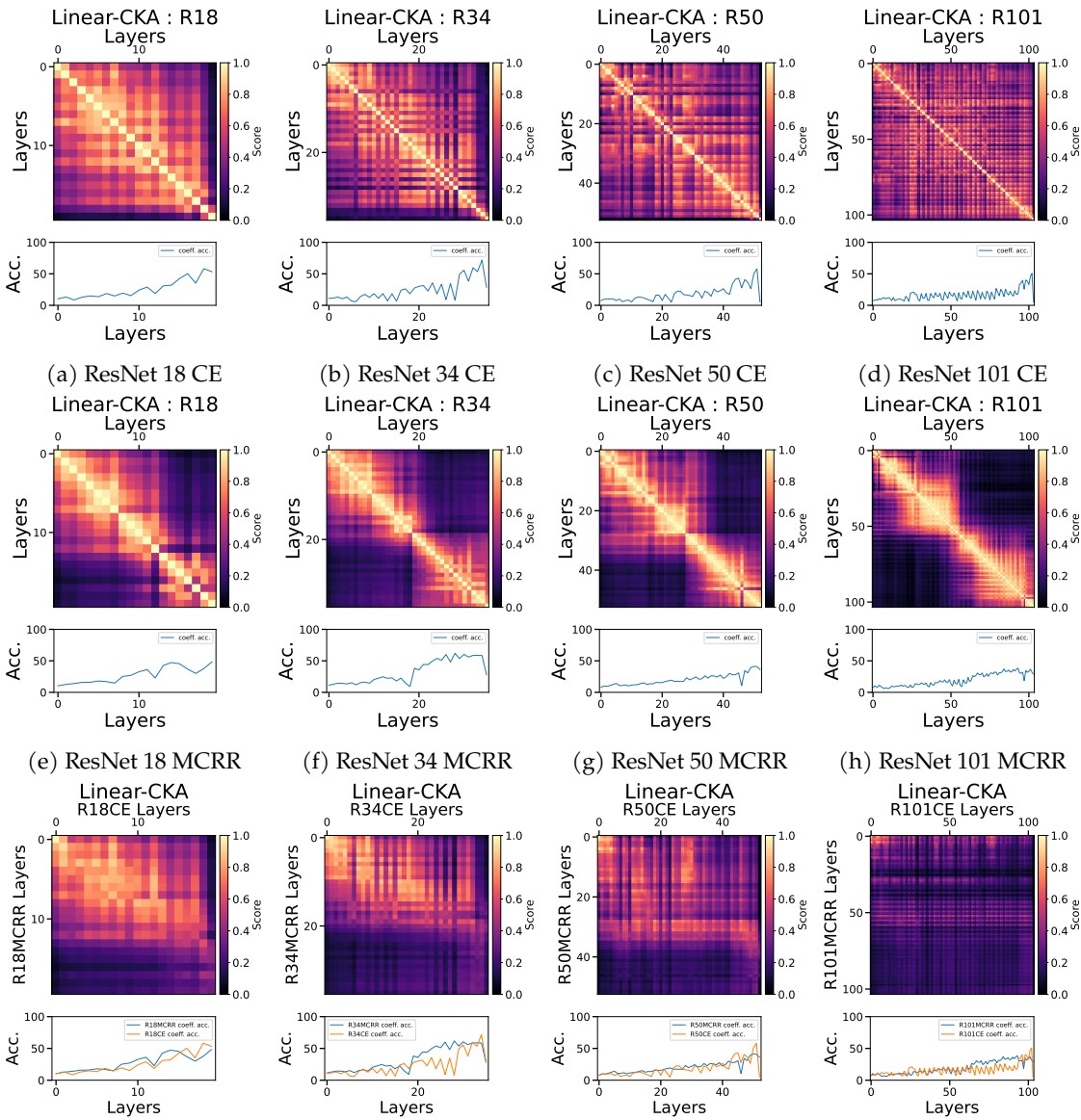

(a) ResNet 18 CE  (b) ResNet 34 CE  (c) ResNet 50 CE  (d) ResNet 101 CE

(e) ResNet 18 MCRR  (f) ResNet 34 MCRR  (g) ResNet 50 MCRR  (h) ResNet 101 MCRR

(i) ResNet 18 MCRR - CE  (j) ResNet 34 MCRR - CE  (k) ResNet 50 MCRR - CE  (l) ResNet 101 MCRR - CE

Figure 66: Linear CKA Analysis of ResNets trained with Maximal Coding Rate Reduction and Cross Entropy loss on CIFAR100.

