# OpenReview forum: "A Case Study of Low Ranked Self-Expressive Structures in Neural Network Representations"
_CPAL.cc/2025/Proceedings_Track — CPAL 2025 (Proceedings Track) Oral_

### Official Review · Reviewer_iJxk · 2025-01-11

**Rating:** 7
**Confidence:** 2

**Review:**

This paper investigates the use of Low Ranked Subspace Clustering (LRSC) and Centered Kernel Alignment to understand geometric structure in the activations of neural networks. The authors compare this method to alternative metrics for representation analysis and evaluate on different trained networks. Although the problem and methods are intriguing, the paper in its present form would benefit from making its motivations, specific insights, and significance of its results clearer. Presenting the existing experiments with improved clarity would be my main suggestion for improvement.

Here are some questions and concerns I have for the authors:

1. Motivations of LRSC and the “self-expressiveness” constraint: Under what circumstances is the assumption that X is low rank justified, given that in this setting it comes from a neural network? And why is self-expressiveness as defined a useful property to have for the representations? The main advantage claimed seems to be the sensitivity of the similarity measure, which does not seem to depend on self-expressiveness per se.

2. In what circumstances is a “uniformly weighted sum of pairwise cosine similarities” preferable to that of Linear-CKA, and why? I can see how this is helpful in some situations but could be harmful in others. This issue seems closely related to the issue of picking the hyperparameter \tau correctly, and it would be great if the authors can comment on this specific point.

3. The specific contributions of the paper are difficult to parse: for example, to understand the experiments of Section 5.2 requires reading the experimental setup of 2 referenced papers. The discussion of computing the residuals — a key step in section 3.2 — is postponed until Equation 5 in the appendix. As another example, it’s difficult as well to understand the significance of the Pearson’s Correlation coefficient results presented in Table 2.

4. The authors primarily compare LRSC with CKA to Linear-CKA, but acknowledge that other kernel methods could be used with CKA. The paper would be improved with some discussion of why the new method proposed would be preferable to these other kernel methods, to help the reader understand how they could use LRSC-CKA in other contexts.

Minor points:
309: which -> with
302: prior working -> prior work

---

### Official Review · Reviewer_sEnM · 2025-01-12
**Cool results**

**Rating:** 7
**Confidence:** 3

**Review:**

Summarize:

The author introduces LRSC-CKA. Combining standard CKA with subspace clustering. The author first shows an analytical connection between LRSC-CKA and linear-CKA, such that LRSC-CKA can be achieved by a small modification of standard linear-CKA. The author then demonstrate the benefit of “LRSC-CKA” over “linear-CKA” in constructed scenario as a proof of concept. Finally, the author use LRSC-CKA to compare different representation from neural network (different layers, different architecture, trained with different scenario and dataset) and show some insight on these representation. ***Personally, I think these insight are very cool, but not unique to the usage of "LRSC-CKA"

Strength:
1. The author done lots of experiment on the different representation from neural network (different layers, different architecture, trained with different scenario and dataset). I think these result itself could be a paper, independent of the promotion of "LRSC-CKA."
2. The author provide a great introduction on concept like LRSC and self-expressiveness, it's easy to follow.
3. The author shows an analytical connection between LRSC-CKA and linear-CKA, and shows problem of linear-CKA (in some scenario) could be mitigated when using LRSC-CKA instead.

Weakness:

1. Some of the conclusion made by the author is not unique to "LRSC-CKA:"
a. the author concludes LRSC reconstruction based accuracy is correlated with standard probing accuracy, which revealed they are a good metric for measuring semantic similarity between two representation. However, many author methods is correlated with standard probing accuracy, even the two listed by the author in table 2. Why is LRSC-CKA so special?
b. All the experiment in data memorisation part could be conducted in the exact same way but replacing LRSC-CKA with CKA or any other metric for representation. It is very likely same conclusion still holds. It seems like the memorisation and generalization phenomenon is a property of neural network, not tied to LRSC-CKA.
c. Similarly, the experiment on neural network trained on coding rate vs neural network trained on cross entropy loss can be done using other metric for representation as well.

2. Other than result from 4.1 (which is a great result), the author did not show any unique property of "LRSC-CKA" compared to other metric for representation. The author starts with toy data with ideal distribution such as data from single subspace or data from a union of low dimensional subspace, and show LRSC-CKA performs well on the later data distribution. I know the point is to use the method on activation of neural network, which has unknown data distribution, but I think such toy example is necessary and help the author to have a more clear and coherence story.


3. The writing of the experiment/result section (section 4 and 5) could be improved. It's difficult to understand what the key message the author tries to convey. It might be beneficial if the author start with concise conclusion on the benefit of LRSC-CKA, then explain each one in more detail.

4. All the heatmap shown in the paper is not that informative. If the author want to conclude on the pairwise similarity of two layer the network, then they should make a clear statement and more finegrained visualization then the heatmap.


As for rating:
I put my rating as a 5, but I would recommend the paper if the author simply changes the narrative for camera ready version of the paper. Like I said in my review, I believe the author shows a cool connection between LRSC-CKA and linear-CKA. And provides many interesting experiment on different representation from neural network (different layers, different architecture, trained with different scenario and dataset). To me, these two results seems independent to each other but are both cool.

---

### Official Review · Reviewer_sfhi · 2025-01-13
**Well motivated method with convicing results**

**Rating:** 8
**Confidence:** 3

**Review:**

This paper proposed Low Rank Subspace Clustering (LRSC) based Centered Kernel Alignment (CKA) algorithm for comparing the representation geometry of neural network activations. This combined algorithm is named LRSC-CKA.

The motivation for developing LRSC-CKA is that the previous baseline method, Linear-CKA, has the issue of being only sensitive to the largest singular values of the activation matrix. While LRSC-CKA is shown to have a uniform weighting on all the singular values. In addition to theoretical derivation, the authors also did a convincing ablation experiment that specifically perturb the largest and the lowest principle components, and showed that the proposed LRSC-CKA is much more correlated to the linear-probe accuracy changes and Linear-CKA.

When applied to analysis of NN representations across different layers, the authors showed that subspace reconstruction accuracy is strongly correlated to linear probing accuracy, and that it is also more correlated to linear probing than using PCA coefficients. This shows that NN representation has significant self-expressive subspace structure.

Using LRSC-CKA, the authors also studied the effect of training partly on noisy labels, which forces the NN to partly memorize the dataset. The conclusion is that NN trained to memorize tends to learn similar representation to regular training until the last few blocks, where the representation similarity to other blocks drops sharply along with linear probe accuracy.

The authors also included other interesting results such as studying the limitation of Linear-CKA and LRSC-CKA under non-Relu activations, and studying the effect of training with MCR loss instead of BCE loss.

Overall, it is demonstrated that LRSC-CKA is a useful and significant improvement over Linear-CKA. It would be interesting in the future to also use this set of tools to analyze the evolution of representation under self-supervised learning setup, and compare with that of supervised learning.

Small issues:
Typo on line 23: ‘that’ -> ‘has’

---

### Meta-Review · Area_Chair_reuM · 2025-02-02

**Recommendation:** Accept (Poster)
**Confidence:** 4

**Metareview:**

The authors propose a technique to measure the similarity of representations of hidden layers of a trained neural network, building on the centered kernel alignment (CKA) technique. The main innovation of this work is to use low-rank subspace clustering in conjunction with CKA ("LRSC-CKA"), in place of the more-standard "Linear-CKA", which corresponds to PCA-like model. One of the main contributions is to demonstrate theoretically and empirically that LRSC-CKA can overcome the issue that Linear-CKA has of being sensitive only to changes in the largest singular value of the activation matrix.

Reviewers are mostly positive about this work. In particular, they appreciated the view of LRSC-CKA is a spectral variant of Linear-CKA. A key concern of Reviewer sEnM was that some of the conclusions drawn from the experiments were not necessarily unique to LRSC-CKA, i.e., the conclusions of the experiments could be drawn using other existing similarity metrics. However, after a lengthy discussion, the reviewer and authors came to a consensus on this point. Finally, many reviewers felt that the experiments could be more clearly explained. The authors indicated in their rebuttal various improvements they will incorporate into the camera-ready version.

Overall, despite some minor weaknesses in terms of presentation of the experiments and the framing of the results, this work makes a nice contribution in terms of overcoming some of the deficiencies of Linear-CKA using the LRSC model, and I recommend its acceptance.

---

### Decision · Program_Chairs · 2025-02-11

Accept (Oral)